# Open fires in Greenland in summer 2017: transport, deposition and radiative effects of BC, OC and BrC emissions

**Nikolaos Evangeliou[1,*], Arve Kylling[1], Sabine Eckhardt[1], Viktor Myroniuk[2], Kerstin Stebel[1], Ronan Paugam[3], Sergiy Zibtsev[2], Andreas Stohl[1]**

[1]Norwegian Institute for Air Research (NILU), Department of Atmospheric and Climate Research (ATMOS), Kjeller, Norway.

[2]National University of Life and Environmental Sciences of Ukraine, Kiev, Ukraine.

[3]King's College London, London, United Kingdom.

**\*** Corresponding author: N. Evangeliou (Nikolaos.Evangeliou@nilu.no)

**Abstract**

Highly unusual open fires burned in Western Greenland between 31 July and 21 August 2017, after a period of warm, dry and sunny weather. The fires burned on peat lands that became vulnerable to fires by permafrost thawing. We used several satellite data sets to estimate that the total area burned was about 2345 hectares. Based on assumptions of typical burn depths and emission factors for peat fires, we estimate that the fires consumed a fuel amount of about 117 kt C and emitted about 23.5 t of black carbon (BC) and 731 t of organic carbon (OC) including 141 t of brown carbon (BrC). We used a Lagrangian particle dispersion model to simulate the atmospheric transport and deposition of these species. We find that the smoke plumes were often pushed towards the Greenland Ice Sheet by westerly winds and thus a large fraction of the emissions (30%) was deposited on snow or ice covered surfaces. The calculated deposition was small compared to the deposition from global sources, but not entirely negligible. Analysis of aerosol optical depth data from three sites in Western Greenland in August 2017 showed strong influence of forest fire plumes from Canada, but little impact of the Greenland fires. Nevertheless, CALIOP lidar data showed that our model captured the presence and structure of the plume from the Greenland fires. The albedo changes and instantaneous surface radiative forcing in Greenland due to the fire emissions were estimated with the SNICAR model and the uvspec model from the libRadtran radiative transfer software package. We estimate that the maximum albedo change due to the BC and BrC deposition was about 0.007, too small to be measured. The average instantaneous surface radiative forcing over Greenland at noon on 31 August was 0.03–0.04 W m$^{-2}$, with locally occurring maxima of 0.63–0.77 W m$^{-2}$ (depending on the studied scenario). The average value is up to an order of magnitude smaller than the radiative forcing from other sources. Overall, the fires burning in Greenland in summer of 2017 had little impact on the Greenland Ice Sheet, causing a small extra radiative forcing. This was due to the – in a global context – still rather small size of the fires. However, the very large fraction of the emissions deposited on the Greenland Ice Sheet from these fires could contribute to accelerated melting of the Greenland Ice Sheet if these fires become several orders of magnitude larger under future climate.

## 1 Introduction

In August 2017 public media reported unprecedented fire events in Western Greenland (BBC News, 2017; New Scientist Magazine, 2017). These events were documented with airborne photographs (SERMITSIAQ, 2017) and satellite images (NASA, 2017b) and raised public concerns about the effects of climate change and possible impacts of soot emissions on ice melting. Historically, wildfires have occurred infrequently on Greenland, because three-quarters of the island is covered by a permanent ice sheet and permafrost is found on most of the ice-free land (Abdalati and Steffen, 2001). Permafrost, or permanently frozen soil, lies under a several meters thick "active" soil layer that thaws seasonally. But in certain areas, where the permafrost layer starts melting, it can expose peat, a material consisting of only partially decomposed vegetation that forms in wetlands over the course of hundreds of years or longer. Peatlands, also known as bogs and moors, are the earliest stage in the formation of coal. Globally, the amount of carbon stored in peat exceeds that stored in vegetation and is similar in size to the current atmospheric carbon pool (Turetsky et al., 2014). When peatlands dry, they are often affected by fires burning into the peat layers. Peat fires are difficult to extinguish and they often burn until all the organic matter is consumed. Smoldering peat fires already are the largest fires on Earth in terms of their carbon footprint (Turetsky et al., 2014). For Greenland, it has been suggested that degradation of peat will accelerate towards 2080 (Daanen et al., 2011) and that the area affected by the fires in August 2017 is particularly vulnerable to permafrost thawing (Daanen et al., 2011).

Fires in the high northern latitudes release significant amounts of $CO_2$, $CH_4$, $N_2O$, black carbon (BC) and organic carbon (OC) and their emissions are often transported into Arctic regions (Cofer III et al., 1991; Hao et al., 2016; Hao and Ward, 1993; Shi et al., 2015). While BC is the most strongly light-absorbing component of the atmospheric aerosol (Bond et al., 2013), a portion of OC compounds has shown strong absorption towards shorter wavelengths of the electromagnetic spectrum (UV), therefore defined as brown carbon (BrC) (Andreae and Gelencsér, 2006; Chakrabarty et al., 2010). BC is formed by the incomplete combustion of fossil fuels, biofuels, and biomass (Bond et al., 2013). BrC is emitted from smoldering fires or solid fuel combustion (Bond, 2001), from pyrolysis of biomass (Mukai and Ambe, 1986) and from biogenic emissions of humic substances (Limbeck et al., 2003). Due to their particulate nature, both BC and OC are important for human health (Lelieveld et al., 2015) and climate impacts (Myhre et al., 2013). BC has an atmospheric lifetime of 3–11 days (Bond et al., 2013), while BrC lifetimes are estimated at 5–7 days (Jo et al., 2016), thus facilitating

transport over long distances (Forster et al., 2001; Stohl et al., 2006). BC, OC and BrC from
mid-latitude sources can thus reach remote areas such as the Arctic. They absorb solar
radiation in the atmosphere (Feng et al., 2013; Hansen and Nazarenko, 2004), have a
significant impact on cloud formation and also decrease surface albedo when deposited on ice
and snow and can accelerate melting processes (Hansen and Nazarenko, 2004; Wu et al.,
2016). This raises particular concerns about the effect of fires burning in the immediate
vicinity of the Greenland Ice Sheet. If a large fraction of the BC emitted by such fires is
deposited on the ice, these fires may be extremely effective in further enhancing the already
accelerating melting of the Greenland Ice Sheet (AMAP, 2017). BC, OC and BrC emissions
from such high latitude fires may also have a substantial effect on the albedo of sea ice.
Here we study transport and deposition of BC, OC and BrC over the Greenland Ice
Sheet from the fires that occurred in Western Greenland in August 2017, which likely
represent the largest fires that have occurred on Greenland in modern times (Figure S 1).
Since the fires occurred in an area entirely lacking ground-based observations, we use satellite
data and a Lagrangian atmospheric dispersion model for our study. Finally, we evaluate the
changes in the albedo of the Greenland Ice Sheet from the respective deposition of BC and
BrC and present instantaneous radiative forcing calculations for these two atmospheric
constituents released from the 2007 fires in Greenland.
**2   Methods**
**2.1   Definition of burned area**
Remote sensing has been useful for delineating fire perimeters, characterizing burn
severity and planning post-fire restoration activities in different regions. The use of satellite
imaging is particularly important for fire monitoring in remote areas due to difficult ground
access. The method that is presented in this section has been already used to calculate burned
area in the highly-contaminated radioactive forests of Chernobyl (Evangeliou et al., 2014,
2015, 2016). Coordinates of fire locations (hot spots) were downloaded from FIRMS (Fire
Information for Resource Management System) (NASA, 2017a). For the mapping of the
burned area, Sentinel 2A images were used. To delineate fire perimeters and define burn
severity precisely, we used Landsat 8 Operational Land Imager (OLI) (resolution: 30×30 m)
together with Sentinel 1A (resolution: 30×30 m) and Sentinel 2A images (resolution: 30×30
m) (see Table 1) by applying the differenced Normalized Burn Ratio (dNBR) (Key and
Benson, 2006):
$$dNBR = NBR_{pre-fire} - NBR_{post-fire} \qquad \text{(Eq. 1)}$$
Normalized burn ratios for pre- ($NBR_{pre-fire}$) and postfire ($NBR_{post-fire}$) images from
Sentinel 2A can be calculated using radiances for near- and shortwave infrared bands (bands 8
(NIR) and 12 (SWIR2) at 0.835 µm and 2.202 µm, respectively):
$$NBR = \frac{1000 \cdot (NIR - SWIR2)}{NIR + SWIR2} \qquad \text{(Eq. 2)}$$
The methodology of applying a dNBR index to assess the impact of fires has been used in
forests of the Northern and Western USA (French et al., 2008; Key and Benson, 2006) and
elsewhere (Escuin et al., 2008; Sunderman and Weisberg, 2011).

The burned severity mosaics were created using Sentinel 2A images corrected for

atmospheric scattering (see Chavez, 1988). Pre– and post–fire images were used to create
cloudless mosaics for the area where the Greenland fires burned. A Maximum Value
Composite (MVC) procedure (Holben, 1986) was used to select pixels from each band that
were not cloud covered and have a high value of Normalized Difference Vegetation Index
(NDVI). To avoid spurious burn severity values, manually delineated fire perimeters were
applied and all areas outside were classified as unburned. We have used common dNBR
severity levels (Key and Benson, 2006) that are presented in Figure 1. The occasionally dense
cloud cover was the main obstacle in reconstructing fire dynamics. As an independent source
of information, active fires from MODIS satellite product MCD14DL (Giglio et al., 2003) are
plotted in Supplemental Information (SI) Figure S 2.

**129    2.2   Injection altitudes, assumptions on biomass consumption and emissions**

**130        factors**

Injection heights into the atmosphere of the emitted smoke were simulated with version

2 of the Plume Rise Model (PRM) (Paugam et al., 2015) which is implemented in the Global
Fire Assimilation System (GFAS) emission inventory (Rémy et al., 2017). The model
(hereafter referred to as PRMv2) is a further development of PRM (Freitas et al., 2006, 2010)
and has already been used in previous studies of fire events (Evangeliou et al., 2015, 2016).
The model simulates a profile of smoke detrainment for every single fire, from which two
metrics are extracted: (i) a detrainment layer (i.e. where the detrainment rate is > 50% of its
global maximum) and (ii) an injection height (InjH, the top of the detrainment layer). Instead
of using the GFAS product, which uses the same statistics as in the PRMv2 InjH calculation,
we ran the model for every detected fire assuming a 6 h persistence and using the same
conversion factor as Kaiser et al. (2012) to estimate the biomass consumption. PRMv2 mass

detrainment profiles are then time integrated and extracted at 1°×1° spatial resolution with a 500 m vertical mesh to estimate the 3D distribution of biomass burning smoke injection into the atmosphere. Figure S 3 (SI) shows for all fires recorded in the MODIS fire product (Justice et al., 2002) during the fire period (31 July – 21 August 2017) the horizontal distribution of the median height of the emitted smoke and its integration over the longitude (right panel). Fires in Greenland showed a maximum injection height of around 2 km, but according to PRMv2 the majority of the emissions (90%) remained below 800 m. Low injection heights mostly inside the daytime planetary boundary layer are quite typical for smoldering fires including peat fires (Ferguson et al., 2003) such as those burning in Greenland (see below). For modeling the dispersion of BC, OC and BrC released from the Greenland fires, the emission profiles from PRMv2 were ingested into the Lagrangian particle dispersion model FLEXPART (see section 2.3).

Wildfires in boreal peatlands in the Canadian Arctic and in Alaska typically have (shallow) burn depths of 1–10 cm and consume 20–30 t C ha$^{-1}$ (Benscoter and Wieder, 2003; Shetler et al., 2008). The consumed carbon is often re-sequestered in 60–140 years after the fire (Turetsky et al., 2011; Wieder et al., 2009). Given that fire return intervals can be as short as 100–150 years in sub-humid continental peatlands (Wieder et al., 2009), and may exceed 2000 years in humid climates (Lavoie and Pellerin, 2007), northern peatlands are generally resilient to wildfire (Magnan et al., 2012). For example, in peatlands of Northern Russia, organic matter available for combustion has been estimated to be 121.8 t C ha$^{-1}$ for forested lands and 21.3 t C ha$^{-1}$ for non-forested lands (Smirnov et al., 2015). Accordingly, a severe wildfire that burned within an afforested peatland in the Scottish Highlands during the summer of 2006 had a mean depth of burn of 17.5±2.0 cm (range: 1–54 cm) and a carbon loss of 96±15 t C ha$^{-1}$ (Davies et al., 2013). In contrast, tropical peatlands can have deep burn depths of 40–50 cm and release an average of 300–450 t C ha$^{-1}$ (Page et al., 2015; Reddy et al., 2015). In the present study, we assume an average amount of organic fuel available for combustion for the Greenland peat fires of August 2017 of 100 t C ha$^{-1}$, guided by values suggested in Smirnov et al. (2015).

Estimation of the emissions of BC, OC and BrC, $E_{BC,OC,BrC}$ (kg), was based on the following formula (Seiler and Crutzen, 1980; Urbanski et al., 2011) using the calculated burned area $A$ (ha) and a number of assumptions:

$$E_{BC,OC,BrC} = A \times FL \times \alpha \times EF \qquad \text{Eq. 1}$$

Here, *FL* is the mass of the fuel available for combustion (kg C ha$^{-1}$); $\alpha$ is the dimensionless
combustion completeness, which was adopted from Hao et al. (2016) for litter and duff fuels
(50%). *EF* is the emission factor (kg kg$^{-1}$), which was assumed to be 0.20 g kg$^{-1}$ for BC and
6.23 g kg$^{-1}$ for OC for peatland fires (Akagi et al., 2011). Emission factors for BrC are rarely
reported, as BrC is only a fraction of OC. To our knowledge, the only reported emission
factors in the literature for BrC are from forest fires in the United States (Aurell and Gullett,
2013) estimated to be 1.0–1.4 g kg$^{-1}$ (value used here: 1.2 g kg$^{-1}$). Fuel consumption is
calculated as the product of burned area, fuel loading and combustion completeness
($A \times FL \times \alpha$).

## 183   **2.3  Atmospheric modeling**

The emissions of BC, OC and BrC obtained from Eq. 1 were fed to the Lagrangian
particle dispersion model FLEXPART version 10.2 (Stohl et al., 2005) to simulate transport
and deposition. This model was originally developed for calculating the dispersion of
radioactive material from nuclear emergencies, but since then it has been used for many other
applications (e.g., Fang et al., 2014; Stohl et al., 2011, 2013). The model has a detailed
description of particle dispersion in the boundary layer and a convection scheme to simulate
particle transport in clouds (Forster et al., 2007). The model was driven by hourly 0.5°×0.5°
operational analyses from the European Centre for Medium-Range Weather Forecasts
(ECMWF). Concentration and deposition fields were recorded in a global domain of 1°×1°
spatial resolution with three hourly outputs. To capture the spatiotemporal variability of BC,
OC and BrC over the Greenland Ice Sheet, a nested domain with 0.05°×0.05° resolution was
used. The simulations accounted for wet and dry deposition, assuming a particle density of
1500 kg m$^{-3}$ and a logarithmic size distribution with an aerodynamic mean diameter of
0.25 μm and a standard deviation of 0.3 (Hu et al., 2018; Long et al., 2013). The wet
deposition scheme considers below-cloud and in-cloud scavenging separately based on cloud
liquid water and cloud ice content, precipitation rate and cloud depth from ECMWF, as
described in Grythe et al. (2017).
To compare BC and OC concentrations in Greenland due to the emissions of the
Greenland fires to those due to emissions occurring elsewhere, we used the so-called
"retroplume" mode of FLEXPART for determining the influence of other sources. For only a
few receptor points, this mode is computationally more efficient than forward simulations.
Computational particles were tracked 30 days back in time from four receptor regions:
Northwestern (-62°E to -42°E, 72°N to 83°N), Southwestern (-62°E to -42°E, 61°N to 72°N),
Northeastern (-42°E to -17°E, 72°N to 83°N) and Southeastern Greenland (-42°E to -17°E,
61°N to 72°N). The retroplume mode allowed identification of the origin of BC and OC
through calculated footprint emission sensitivities (often also called source-receptor
relationships) that express the sensitivity of the BC and OC surface concentrations at the
receptor to emissions on the model output grid. If these emissions are known, BC and OC
concentrations at the receptor can be calculated as the product of the emission flux and the
emission sensitivity. Also, detailed source contribution maps can be calculated, showing
which regions contributed to the simulated concentration. For the anthropogenic emissions,
we used the ECLIPSE (Evaluating the CLimate and Air Quality ImPacts of ShortlivEd
Pollutants) version 5 (Klimont et al., 2017) emission data set. For the biomass burning
emissions outside Greenland, we used operational CAMS GFAS emissions (Kaiser et al.,
2012). To our knowledge, actual gridded emissions of BrC are not yet available.
**2.4   Instantaneous radiative forcing (IRF) calculations**
The IRF of the emitted substances of interest were calculated using the uvspec model
from the libRadtran radiative transfer software package (http://www.libradtran.org/doku.php)
(Emde et al., 2016; Mayer and Kylling, 2005). The radiative transfer equation was solved in
the independent pixel approximation using the DISORT model in pseudo-spherical geometry
with improved treatment of peaked phase functions (Buras et al., 2011; Dahlback and
Stamnes, 1991; Stamnes et al., 1988). Radiation absorption by gases was taken from the Kato
et al. (1999) parameterization modified as described in the libRadtran documentation and
Wandji Nyamsi et al. (2015). External mixture of aerosols was assumed, i.e. BC and BrC
were treated in isolation of other aerosol types that may also have been present in the plume.
This assumption likely leads to underestimates of the radiative impacts, at least for BC
(Jacobson, 2001), in the atmosphere as coating, for example, can enhance its radiative effects.
However, these assumptions should have little impact on the more important albedo
calculations (see below). For snow-covered surfaces, deposited BC and BrC were assumed to
reside in the uppermost 5 mm. Below 5 mm the snow was assumed to be without any
impurities. The albedo of the snow was calculated with the SNICAR model
(http://snow.engin.umich.edu/info.html) in a two-layer configuration (Flanner et al., 2007,

2009).

The IRF was calculated for three scenarios: (a) BC only, (b) BC and BrC and (c) BC and
BrC, where all OC is considered to be BrC. The BC only scenario demonstrates the impact of
BC alone, while the two other scenarios provide an estimate of the additional impact of BrC
in the plume, with the last scenario considered to be a maximum estimate. We calculated both
the bottom of the atmosphere (BOA) and top of atmosphere (TOA) instantaneous radiative
forcing (IRF) due to the Greenland fires at 1°×1° resolution. The IRF includes both the effects
of BC and BrC in the atmosphere and deposited in snow. Note that the IRF does not include
any semi-direct nor indirect effects. We show IRF for cloudy conditions, which represents the
possible radiative effects of BC and BrC due to the 2017 fires with respect to the actual
meteorological situation. Liquid and ice water clouds were adopted from ECMWF.
**2.5   Remote sensing of the smoke plume**
To confirm the presence of the emitted substances from the Greenland fires and
elsewhere in the atmosphere over Greenland, we used the AERONET (AErosol RObotic
NETwork) data (Holben et al., 1998). AERONET provides globally distributed observations
of spectral aerosol optical depth (AOD), inversion products, and precipitable water in diverse
aerosol regimes. We chose data from three stations that were close to the 2017 fires and for
which cloud-free data exist for most of the simulated period, namely Kangerlussuaq
(50.62°W–66.99°N), Narsarsuaq (45.52°W–61.16°N) and Thule (68.77°W–76.51°N). Their
locations are shown in Figure S 2. We used Level 2.0 AOD data (fine and coarse mode AOD
at 500 nm and total AOD at 400 nm) from the AERONET version 3 direct-sun spectral
deconvolution algorithm (SDA version 4.1) product (downloaded on 20 July 2018) for the
simulated period (31 July to 31 August 2017).
To examine in particular the vertical depth of the smoke, we used data from the
CALIOP (Cloud-Aerosol Lidar with Orthogonal Polarization) lidar on the CALIPSO (Cloud-
Aerosol Lidar and Infrared Pathfinder Satellite Observations) platform (Winker et al., 2009).
CALIOP provides profiles of backscatter at 532 nm and 1064 nm, as well as the degree of the
linear polarization of the 532 nm signal. For altitudes below 8.3 km lidar profiles at 532 nm
are available with a vertical resolution of 30 m. We have utilized the level 1 data products
(version 3.40) of total attenuated backscatter at 532 nm. This signal responds to aerosols (like
BC, OC and BrC) as well as water and ice clouds, which in most cases can be distinguished
based on their differences in optical properties. The data were downloaded from the ICARE
Data and Services Center (http://www.icare.univ-lille1.fr/).

## 3  Results

### 3.1  Indications of early permafrost degradation and fuel availability

Table 1 reports burned areas in August 2017 calculated for Greenland. In total, 2345 hectares burned between 31 July and 21 August 2017 (Figure 1). We estimate that about 117 kt of carbon were consumed by these fires. The area burned is not large compared to the global area burned each year (464 million hectares), or the areas burned in boreal North America (2.6 million hectares) or boreal Asia (9.8 million hectares) (Randerson et al., 2012), but still highly unusual for Greenland.

It is not yet known how these fires started. Fires on carbon-rich soils can be initiated by an external source, e.g. lightning, flaming wildfire and firebrand, or self-heating. The fires burned relatively close to the town of Sisimut, so it is quite possible that humans started the fires. Self-heating is another possibility as porous solid fuels can undergo spontaneous exothermic reactions in oxidative atmospheres at low temperatures (Drysdale, 2011; Restuccia et al., 2017b). This process starts by slow exothermic oxidation at ambient temperature, causing a temperature increase, which is determined by the imbalance between the rate of heat generation and the rate of heat losses (Drysdale, 2011). Fire initiated by self-heating ignition is a well-known hazard for many natural materials (Fernandez Anez et al., 2015; Restuccia et al., 2017a; Wu et al., 2015) and can also occur in natural soils (Restuccia et al., 2017b). Southwestern Greenland was under anticyclonic influence during the last week of July and according to the MODIS ESDIS worldview tool, direct sunshine occurred for eight consecutive days before the fires started at the end of July 2017. It might be possible that this long period of almost continuous insolation at these latitudes in July heated the soil enough to self-ignite. In any case, the continuous sunshine had dried the soil, making it susceptible to fire.

The fact that these fires were burning for about three weeks but spread relatively slowly compared to above-ground vegetation fires indicates that the main fuel was probably peat. The predominant vegetation in Western Greenland varies from carbon-rich Salix glauca low shrubs (mean canopy height: 95 cm), mainly at low altitude south-facing slopes with deep soils and ample moisture, to dwarf-shrubs and thermophilous graminoid vegetation (Arctic steppe) at higher altitudes (Jedrzejek et al., 2013). In addition, the observed smoke was nearly white, indicating damp fuel, such as freshly thawed permafrost, which produces smoke rich in OC aerosol (Stockwell et al., 2016).

Literally no fires should be expected in Greenland, since there is little available fuel as

it has been suggested by global models and validated by observations (Daanen et al., 2011;
Stendel et al., 2008); the only way to provide substantial amounts of fuel in Greenland is
permafrost degradation. However, it has been suggested that significant permafrost loss in
Greenland may occur only by the end of the 21[st] century (Daanen et al., 2011; Stendel et al.,
2008). The fires in 2017 might indicate that significant permafrost degradation has occurred
sooner than expected.

## 3.2   Transport and deposition of BC in Greenland

We estimate that about 23 t of BC and 731 t of OC, including 141 t of BrC, were

released from the Greenland fires in August 2017 (Table 1). According to the FLEXPART
model simulations, these emissions were transported and deposited as shown in Figure 2. Due
to the low injection altitude of the releases within the boundary layer, transport was relatively
slow and thus the emitted substances initially remained quite close to their source. Slow
transport was also favored by mostly anticyclonic influence during the first half of August. It
seems that even though katabatic winds from the Greenland Ice Sheet occasionally
transported the plume westwards, most of the time the large-scale circulation pushed the
plume back towards Greenland (see SI animations). Consequently, a large fraction of the
emitted substances were deposited in Southwestern Greenland. On 3 August a small portion
of the emitted BC, OC and BrC (0.5 t, 16.1 t and 3.1 t, respectively) were lifted higher into
the atmosphere and were transported to the east and deposited in the middle of the Ice Sheet
over the course of the following two days (4 and 5 August). From 5 to 8 August, when the
fires were particularly intense, the emitted aerosols were transported to the south, where they
were deposited at the southern part of the Ice Sheet and close to the coastline. At the same
time, another branch of the plume was moving to the north depositing BC, OC and BrC over
Greenland's western coastline up to 80°N. Around 10 August, the plume circulated north- and
then eastwards in the northwestern sector of the anti-cyclone and the emitted aerosols were
deposited to the northern part of the Ice Sheet until 13 August. From around 16 August, a
cyclone approached from the northwest and the smoke was briefly transported directly
eastwards along the southern edge of the cyclone (see SI animations). Strong rain associated
with the cyclone's frontal system appears to have largely extinguished the fire by 17 or 18
August, although smaller patches may have continued smoldering for a few more days before
they also died out. The exact fire behavior after 16 August is difficult to determine because of
frequent dense cloud cover. However, satellite imagery on 21 August shows no smoke
anymore in the area where the fires had burned.
The total deposition of BC, OC and BrC from the fires in Greenland was estimated to
be 9 t, 280 t and 54 t, respectively, or about 39% of the total emissions. About 7 t of BC, 218 t
of OC and 42 t of BrC were deposited on snow or ice covered surfaces, which is equivalent to
30% of the total emissions. Most of the rest was deposited in the Baffin Bay between
Greenland and Canada and in the Atlantic Ocean. With 30% of the emissions deposited on
snow or ice surfaces, Greenland fires may have a relatively large efficiency for causing
albedo changes on the Greenland Ice Sheet.
By comparison, the respective BC deposition on snow and ice surfaces over Greenland
from global emissions of BC (from ECLIPSEv5) was only 0.4% (39 kt) of the total emissions.
Even the total deposition of BC in the Arctic (>67°N) was only about 3% (215 kt). This
indicates the high relative potential of Greenland fires to pollute the cryosphere (on a per unit
emission basis), likely also giving them a particularly high radiative forcing efficiency.
Considering that the projected rise of Greenland temperatures is expected to result in further
degradation of the permafrost (Daanen et al., 2011) and, hence, likely resulting in more and
larger peat fires on Greenland, this constitutes a potentially important climate feedback which
could accelerate melting of the glaciers and ice sheet of Greenland and enhance Arctic
warming.
We also calculated the concentration of the deposited carbon aerosols in Greenland
snow (Figure 3) by taking the ratio of deposited quantities and the amount of water deposited
by rain or snowfall during the same time period (31 July to 31 August 2017). As expected,
snow concentrations show the same general patterns as the simulated deposition with the
highest concentrations obtained close to the source (western side of Greenland). High snow
concentrations were also computed in some regions of the Ice Sheet due to relatively intense
precipitation events. By contrast, dry deposition (example for BC) over the Ice Sheets was
low (Figure S 4). Dry deposition was responsible for a major fraction of the deposition only in
regions where the plume was transported during dry weather, and in most of these regions
total deposition was low. A notable exception is the region close to the fires, where dry
deposition was relatively important due to the generally dry weather when the fires were
burning. It can be also ascribed to the fact that dry deposition occurs in the quasi-laminar sub–
layer close to the surface. A fraction of the aerosols can be quickly deposited close to the
sources before they are transported to higher altitudes and away from the sources (Bellouin

and Haywood, 2014). The average calculated snow concentration of BC on the Ice Sheet was estimated to be <1 ng g$^{-1}$, but in some areas snow concentrations reached up to 3 ng g$^{-1}$. These higher values are substantial considering that measured concentrations of BC in snow typically range up to 16 ng g$^{-1}$ in most of Greenland (Doherty et al., 2010) or from 1 – 17 ng g$^{-1}$ in summer 2012 and 3–43 ng g$^{-1}$ in summer 2013 (Polashenski et al., 2015) and up to 15 ppb C (ng g$^{-1}$) during preindustrial times (from 1740 to 1870) on average (Legrand et al., 2016). OC concentrations in snow were 2 ng g$^{-1}$ (ppb C), on average, with local maxima of 10 ng g$^{-1}$. They are lower than those measured in snow over several places in Antarctica (23–928 ppb C) (Antony et al., 2011; Grannas et al., 2004; Legrand et al., 2013; Lyons et al., 2007), in Greenland (400–580 ppb C) (Grannas et al., 2004) or in the Alps (70–304 ppb C) (Legrand et al., 2013). Snow BrC was estimated to be even less; though, to our knowledge, no available measurements exist in the relevant literature so far.

It has been reported that the size of rapidly coagulated aerosol particles produced by different types of fires ranges between 0.1 to 10 μm, but more than 90% of the mass lies between 0.1 and 1 μm (e.g., Conny and Slater, 2002; Long et al., 2013; Zhuravleva et al., 2017 and many others). Therefore, we simulated the Greenland fires with an aerodynamic mean diameter of 0.25 μm for BC, OC and BrC and a logarithmic standard deviation of 0.3 (see section 2.3), because all these substances have more or less the same lifetimes (Bond et al., 2013; Jo et al., 2016; Lim et al., 2003). To examine the sensitivity of deposition in the Greenland Ice Sheet from the Greenland fires of 2017 to the particle size distribution used in the model, we simulated the same event for particles with aerodynamic mean diameters of 0.1, 0.25, 0.5, 1, 2, 4 and 8 μm and calculated the relative standard deviation of deposition normalized against the aerodynamic mean diameter of 0.25 μm that was our basic assumption. The results are shown in Figure S 5 for BC. The use of different size distributions for the BC particles produced from the 2017 fires created a relative uncertainty on the deposited mass of BC in the Greenland Ice Sheet, which ranges from 10%–30% in 86% of the Sheet's surface to up to 50% in the rest of the Sheet's surface. As expected, the calculated uncertainty is sensitive to the use of larger particles for BC; though BC particles larger than 1 μm are rather rare in peat fires (Hosseini et al., 2010; Leino et al., 2014).

**3.3  Impact from other emissions in the Northern Hemisphere**

In summertime 2017, intense wildfires were reported in British Columbia, Western Canada (NASA, 2017c), and fires also burned at mid latitudes in Eurasia, as is typical during spring and summer (Hao et al., 2016). Previous studies of wildfires have shown that the

produced energy can be sufficient to loft smoke above the boundary layer by supercell
convection (Fromm et al., 2005) even up to stratospheric altitudes (Leung et al., 2007). As a
result, emitted aerosols can become subject to long-range transport over long distances
(Forster et al., 2001; Stohl et al., 2007). To examine the impact of these fires in Greenland,
average footprint emission sensitivies were calculated for four compartments of Greenland
(Northwestern, Southwestern, Northeastern and Southeastern Greenland) for the period 31
July to 31 August 2017 and the results are shown in Figure S 6 together with the active fires
in the Northern Hemisphere from 10 July to 31 August 2017 adopted from the MODIS
satellite product (MCD14DL) (Giglio et al., 2003). As can be seen in Figure S 6, fires in
Alaska and in Western Canada might have affected BC, OC and BrC concentrations in
Greenland, as the corresponding emission sensitivies are the highest in North America. On
the contrary, emissions from fires in Eurasia seem to have affected Greenland less.
Using gridded emissions for BC and OC, the contribution of both biomass burning and
anthropogenic sources to surface concentrations in the four different regions over Greenland
(Northwestern, Northeastern, Southwestern and Southeastern Greenland, Figure S 7) was
calculated (see section 2.3). Fires affected the northern part of Greenland more than the
southern part with an average BC concentration of about 30 ng m$^{-3}$, almost twice the
respective average for Southern Greenland ($\approx$16 ng m$^{-3}$). OC simulated concentrations were
much higher that those of BC with an average concentration of 945 ng m$^{-3}$ in North
Greenland, while the respective concentrations in the southern part were about 490 ng m$^{-3}$.
About one third of BC and OC originated from wildfires in Eurasia and the rest from North
America where the year 2017 appears to have been a particularly high fire year. The
anthropogenic contribution to surface concentrations of BC and OC over Greenland was
between 14% to 50% of the total contribution from all biomass burning sources (Figure S 7),
similar to what has been suggested previously for the Arctic in summer (Winiger et al., 2017).
The anthropogenic contribution is larger in Southern Greenland than in Northern Greenland,
due to the shorter distance from the main emission areas of North America and Western
Europe, but it remains much lower than the biomass burning contribution. The concentrations
of BC and OC that are calculated for the studied fire period (31 July to 31 August 2017) are
relatively high compared to those reported previously. For instance, von Schneidemesser et al.
(2009) observed an annual average BC concentration of 20 ng m$^{-3}$ at Summit (Greenland) in
2006, while Massling et al. (2015) reported a summer average BC concentration of 11 ng m$^{-3}$
at station Nord (Greenland) between May 2011 and August 2013. As regards to OC, average
concentrations of its water soluble part were measured in 2006 between 194 and 730 ng m$^{-3}$ in
Summit, Greenland (Anderson et al., 2008) showing a large decreasing trend compared to
previous years (Dibb et al., 2002). We attribute this difference in the calculated concentrations
to more active fires during 2017 in Greenland than in previous years (see Figure S 1).
As an example of the importance of Northern Hemispheric biomass burning emissions
for the air over Greenland, we present time-series of surface BC concentrations in
Northwestern, Northeastern, Southwestern and Southeastern Greenland from the fires in
Greenland and from all the other wildfire emission sources occurring outside Greenland
(North Hemisphere) for the same period of time (Figure 4). The calculated dosages
(concentrations summed over a specific time period) for the same time period were also
computed. The fires in Greenland affected mainly its western part with concentrations that
reached up to 4.8 ng m$^{-3}$ (Southwestern Greenland on 10 August) and 4.4 ng m$^{-3}$
(Northwestern Greenland on 12 August), while BC concentrations in the eastern part
remained significantly lower (Figure 4). These concentrations are substantial considering that
the observed surface BC concentrations in Greenland in summer are usually below 20 ng m$^{-3}$
(Massling et al., 2015). Surface BC due to wildfires occurring outside Greenland was also low
most of the time in the studied period (up to 10 ng m$^{-3}$ at maximum) except for a large peak
between 19 and 23 August that mainly affected Northern Greenland (Figure 4). The
concentrations during this episodic peak were as high as 27 ng m$^{-3}$. During the same period,
the contribution from anthropogenic emissions was also a few ng m$^{-3}$ (Figure 4). BC dosages
for the simulation period (31 July – 10 August 2017) in Western Greenland due to the
Greenland fires were about one order of magnitude smaller than dosages from fires elsewhere
but of the same order of magnitude as BC originating from anthropogenic emissions.
**4   Discussion**
**4.1   An evaluation attempt**
There are few observations available that can be used to evaluate our model results. We
use the AERONET and CALIOP data for some qualitative comparisons. We present only BC
here, but similar plots can be generated for OC, considering that we used the same scavenging
coefficients as for BC to represent the similar lifetimes of BC and OC (Bond et al., 2013; Jo
et al., 2016; Lim et al., 2003). Contours of simulated vertical distribution of BC and column-
integrated simulated BC from fires inside and outside Greenland are plotted together with
time-series of measured AOD (fine and coarse mode AOD at 500 nm and total AOD at 400

nm) for the AERONET stations Kangerlussuaq, Narsarsuaq and Thule (Figure 5). It can be seen that observed AOD variations were in very good agreement with the variation of simulated column-integrated BC from fires outside Greenland (mainly in Canada), confirming that the transport of these fire plumes was well captured by FLEXPART. Good examples are the peaks at Kangerlussuaq on 24 August, at Narsarsuaq on 19 August and at Thule on 21 August (Figure 5) that are attributed to the Canadian fires. The simulated contribution of the Greenland fires to simulated BC burdens was negligible by comparison, except at Kangerlussuaq in the beginning of August when the Greenland fire emissions were the highest. This station is less than 100 km away from where the fires burned, but not in the main direction of the BC plume transport. It seems the period of simulated fire influence corresponds to a small increase of the observed AOD values of up to 20% (Figure 5).

To evaluate the smoke plume's vertical extent, we used the CALIOP data. These data were only available from 5 August 2017 onward and frequent dense cloud cover inhibited lidar observations at the altitudes below the clouds. High aerosol backscatter was only found in the close vicinity of the fires. Figure 6a shows NASA's ESDIS view of the plume on 14 August 2017 at 6 UTC (available: https://worldview.earthdata.nasa.gov/?p=ge ographic&l=MODIS_Aqua_CorrectedReflectance_TrueColor(hidden),MODIS_Terra_Correc tedReflectance_TrueColor,MODIS_Fires_Terra,MODIS_Fires_Aqua,Reference_Labels(hidd en),Reference_Features,Coastlines&t=2017-08-14&z=3&v=- 54.13349998138993,66.35888052399868,-50.32103113049877,69.08420005412792), where a clear smoke signal was recorded. A CALIOP overpass through the edge of the plume allows studying its vertical structure. Increased attenuated backscatter is found below ~1.5 km above sea level between 52°E and 51°E (Figure 6b; black line denotes the orography). Figure 6c (red line), shows that the CALIOP overpass transects directly the simulated plume of the Greenland fires. Notice that the simulated plume also agrees very well with the smoke as seen in NASA's ESDIS picture (Figure 6a). The vertical distribution of simulated BC as a function of longitude is illustrated in Figure 6d. It corresponds very well to the vertical distribution of aerosols observed by CALIOP (Figure 6b). In particular, the smoke resides at altitudes below 1.5 km and at exactly the same location both in the simulations and observations.

## 4.2 Instantaneous radiative forcing and albedo effects

BOA IRF due to (a) BC only, (b) BC and BrC and (c) BC and BrC when all OC was assumed to be BrC (extreme scenario) for noon on 31 August 2017 is depicted in Figure 7a–c. This day is shown because almost all the aerosols emitted by the fires had been deposited,

thus giving a high IRF via albedo reduction due to snow contamination. The IRF is the largest
over ice close to the fire site and at locations where relatively large amounts of BC and BrC
were deposited. For BC only, the maximum BOA (TOA) IRF is 0.63 W m$^{-2}$ (0.59 W m$^{-2}$),
and the average 0.03 W m$^{-2}$ (0.03 W m$^{-2}$). Including BrC slightly increases the maximum
BOA (TOA) IRF to 0.65 W m$^{-2}$ (0.61 W m$^{-2}$), while the change in the average IRF values is
negligible. For the extreme BrC scenario, the maximum BOA (TOA) IRF is 0.77 W m$^{-2}$ (0.71
W m$^{-2}$) and the average 0.04 W m$^{-2}$ (0.06 W m$^{-2}$). So, including BrC in our analysis increases
BOA IRF by only 20% even for the extreme scenario.

The IRF depends on the optical properties of the smoke from the fire, which are not

known. Hence, a sensitivity analysis was performed where the single scattering albedo (SSA)
was perturbed in contrast to a "medium case" (Figure S 8a) that was adopted from the
SNICAR model (Flanner et al., 2007, 2009) and has been used for the discussion in the
previous paragraph. To estimate the uncertainty due to the choice of BC optical properties,
additional calculations were made by scaling the SSA (red solid lines in Figure S 8a). The
choices of these scaled SSA values were based on the SSA reported for various modified
combustion efficiencies (MCE) by Pokhrel et al. (2016). Pokhrel et al. (2016) reported an
MCE of 0.9 for peat land. As such, our adopted SSA may be considered low (compare black
solid line and red line with upward triangles). Figure S 8b shows the IRF as BC is deposited
for the three cases. It suggests that the IRF ranges between 40% and 130% of our above-
assumed medium-case values for realistic variation of the aerosol optical properties.

Figure 7d depicts the temporal behaviour of the cloudy TOA IRF averaged over

Greenland (daily averages) for BC only (red line), for BC and BrC (blue line) and for BC and
BrC, when all OC is assumed to be BrC (black line, extreme case scenario). The daily
averaged IRF is seen to increase as the plume from the fires spreads out and starts to decline
after the fires were extinguished at the end of the month. The fact that the reduction towards
end of August is relatively slow is caused by the effect of the albedo reduction, which persists
until clean snow covers the polluted snow. Overall, albedo reduction dominates the total IRF
averaged over Greenland for the period of study contributing between 85% (in the beginning
of the study period) to 99% (at the end of the study period) and increasing in relative
importance with time as atmospheric BC and BrC are removed. The largest IRF differences
between the BC only case IRF and the two BC+BrC cases occur when there is still smoke in
the air and the lowest IRF differences occur after August 15[th]. This indicates that BrC is most
important for the IRF when it is airborne, even in the extreme scenario. However, for the
latter, the impact is also large after August 15[th] due to a further albedo decrease of about
0.001 compared to the case where only BC was considered.

According to Hansen et al. (2005) the TOA IRF of BC approximates the adjusted RF as

reported by Myhre et al. (2013). In their Table 8.4, Myhre et al. (2013) estimated the global
averaged RF due to BC between the years 1750 and 2011 to be +0.40 (+0.05 to +0.80) W m$^{-2}$.
Skeie et al. (2011) estimated a global mean radiative forcing of 0.35 W m$^{-2}$ due to fossil fuel
and biofuel increases between 1750 and 2000. For Greenland, Skeie et al. (2011) found the
RF to be less than about 0.2 W m$^{-2}$. This number may be compared to our area averaged IRF
estimate due to the Greenland fire. For cloudy conditions the TOA IRF over Greenland due to
the Greenland fires is about a factor 4 to 10 smaller compared with the RF over Greenland
due to BC from all global anthropogenic sources reported in Skeie et al. (2011).

The albedo reduction at 550 nm for the three scenarios (BC only, BC+BrC and BC+BrC

extreme) is shown in Figure 7e–g. The maximum albedo change is about 0.006 when only BC
was considered. Adding BrC from the most extreme scenario, the maximum albedo change
was calculated as 0.007 This albedo change has an impact on IRF, but it is too small to be
measured by satellites. For example, MODIS albedo estimates have been compared to in situ
albedo measurements in Greenland by Stroeve et al. (2005). They found that the root mean
square error between MODIS and in situ albedo values was ±0.04 for high quality flagged
MODIS albedo retrievals. Unmanned Aerial Vehicle (UAV) measurements over Greenland
made by Burkhart et al. (2017) have uncertainties of similar magnitude. Also, Polashenski et
al. (2015) reported that the albedo reduction due to aerosol impurities on the Greenland Ice
Sheet in 2012–2014 period is relatively small (mean 0.003), though episodic aerosol
deposition events can reduce albedo by 0.01–0.02. The albedo changes due to BC and BrC
from the Greenland fires are generally an order of magnitude smaller (Figure 7e–g) and thus
too small to be detected by present UAV and satellite instruments and retrieval methods
(Warren, 2013).
**5    Conclusions**

We studied atmospheric transport, deposition and impact of BC, BrC and OC emitted as

a result of unusual open fires burning in Greenland between 31 July and 21 August 2017. Our
conclusions can be summarized below:
• The fires burned on peat lands that became vulnerable by permafrost thawing. The region

where the fires burned was identified previously as being susceptible to permafrost

melting; however, large-scale melting was expected to occur only towards the end of the
21st century. The 2017 fires show that at least in some locations substantial permafrost
thawing is already occurring now.
• The total area burned was about 2345 hectares. We estimate that the fires consumed a fuel
amount of about 117 kt C and emitted about 23.5 t of BC and 731 t of OC including 141 t
of BrC.
• The Greenland fires were small compared to fires burning at the same time in North
America and Eurasia, but a large fraction of BC, OC and BrC emissions (30%) was
deposited on the Greenland Ice Sheet.
• Measurements of aerosol optical depth at three sites in Western Greenland in August 2017
were strongly influenced by forest fires in Canada burning at the same time, but the
Greenland fires had an observable impact doubling the column-integrated BC
concentrations at the closest station.
• A comparison of the simulated BC releases in FLEXPART with the vertical cross-section
of total attenuated backscatter (at 532 nm) from CALIOP lidar showed that the
spatiotemporal evolution and particularly the top height of the plume was captured by the
model.
• We estimate that the maximum albedo change due to the BC deposition from the
Greenland fires was about 0.006, whereas adding deposited BrC increases albedo to 0.007
at maximum, which is too small to be measured. The average instantaneous BOA radiative
forcing over Greenland at noon on 31 August was between 0.03–0.04 W m$^{-2}$ for the three
scenarios (BC only, BC+BrC and BC+BrC extreme), with locally occurring maxima of
0.63 W m$^{-2}$, 0.65 W m$^{-2}$ and 0.77 W m$^{-2}$, respectively. The average value when only BC
was considered is up to an order of magnitude smaller than the radiative forcing due to BC
from other sources.
• We conclude that the fires burning in Greenland in summer of 2017 had small impact on
the Greenland Ice Sheet, causing almost negligible extra radiative forcing. This was due to
the – in a global context – still rather small size of the fires.
The very large fraction of the emissions deposited on the Greenland Ice Sheet from
these fires (30% of the emissions) could contribute to accelerated melting in Greenland if such
fires become more severe under future climate.
The very large fraction of the emissions deposited on the Greenland Ice Sheet makes
these fires very efficient climate forcers on a per unit emission basis. Thus, while the fires in
2017 were still relatively small on a global scale, if the expected future warming of the Arctic
(IPCC, 2013) produces more and larger fires in Greenland (Keegan et al., 2014), this could
indeed cause substantial albedo changes and thus contribute to accelerated melting of the
Greenland Ice Sheet.

*Data availability.* All data used for the present publication can be obtained from the
corresponding author upon request.

*Competing financial interests.* The authors declare no competing financial interests.

*Acknowledgements*. This study was partly supported by the Arctic Monitoring and
Assessment Programme (AMAP) and was conducted as part of the Nordic Centre of
Excellence eSTICC (Nordforsk 57001). We acknowledge the use of imagery from the NASA
Worldview application (https://worldview.earthdata.nasa.gov/) operated by the
NASA/Goddard Space Flight Center Earth Science Data and Information System (ESDIS)
project. We thank Brent Holben and local site managers for their effort in establishing and
maintaining the AERONET sites used in this investigation. We thank NASA/CNES engineers
and scientists for making CALIOP data available. The lidar data were downloaded from the
ICARE Data and Service Center.

*Author contributions*. NE performed the simulations, analyses, wrote and coordinated the
paper. AK performed the radiation calculations and wrote parts of the paper. VM and SZ
performed GIS analysis for the burned area calculations. RP made all the runs for the
injection height calculations using the PRMv2 model. KS analysed satellite data for AOD and
CALIOP, SE and AS commented and coordinated the manuscript. All authors contributed to
the final version of the manuscript.

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

 **FIGURE LEGENDS**

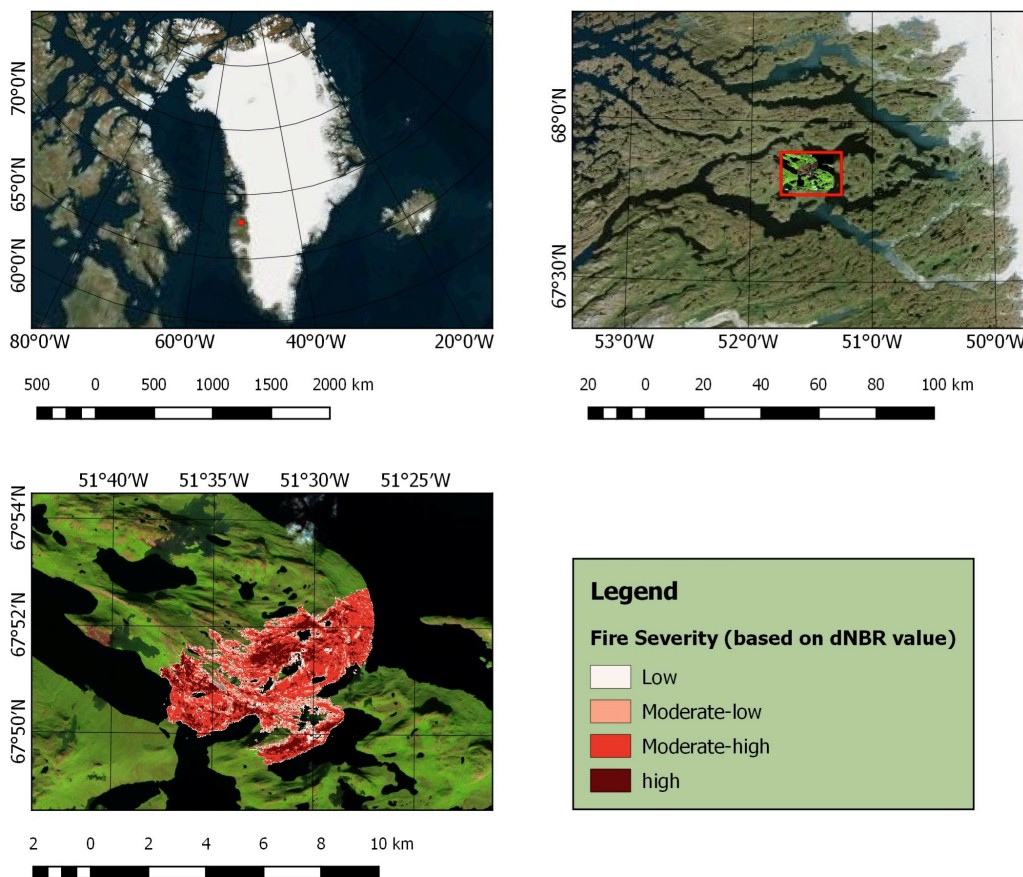

**Figure 1.** Map of Greenland (upper left) and zoomed map marked with fire location (upper
right and burned area classification (bottom) in terms of fire severity according to Sentinel 2A
images for fires burning in Greenland in August 2017. To delineate fire perimeters, both
Landsat 8 OLI and Sentinel 1A – 2A data were used (**Table 1**).

1037

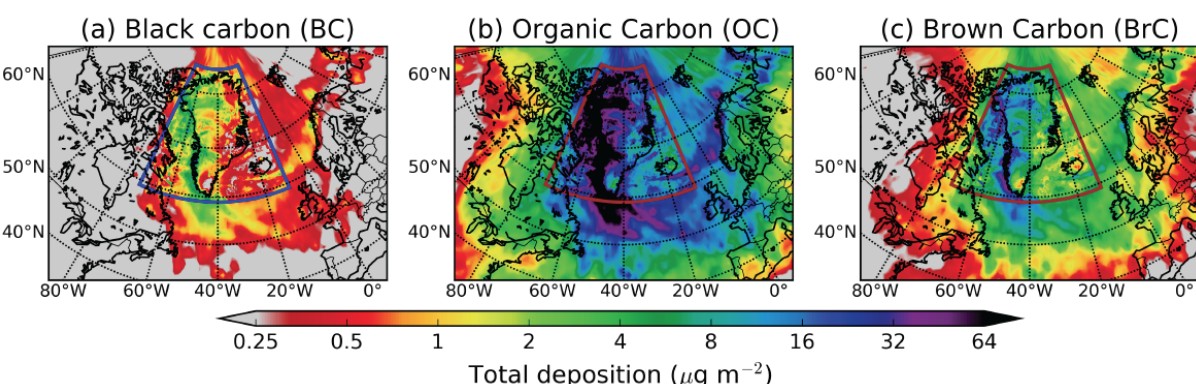

**Figure 2.** Total (wet and dry) deposition of (a) BC, (b) OC and (c) BrC (in μg m$^{-2}$) from the Greenland fires until 31 August 2017. The colored rectangle depicts the nested high-resolution domain.

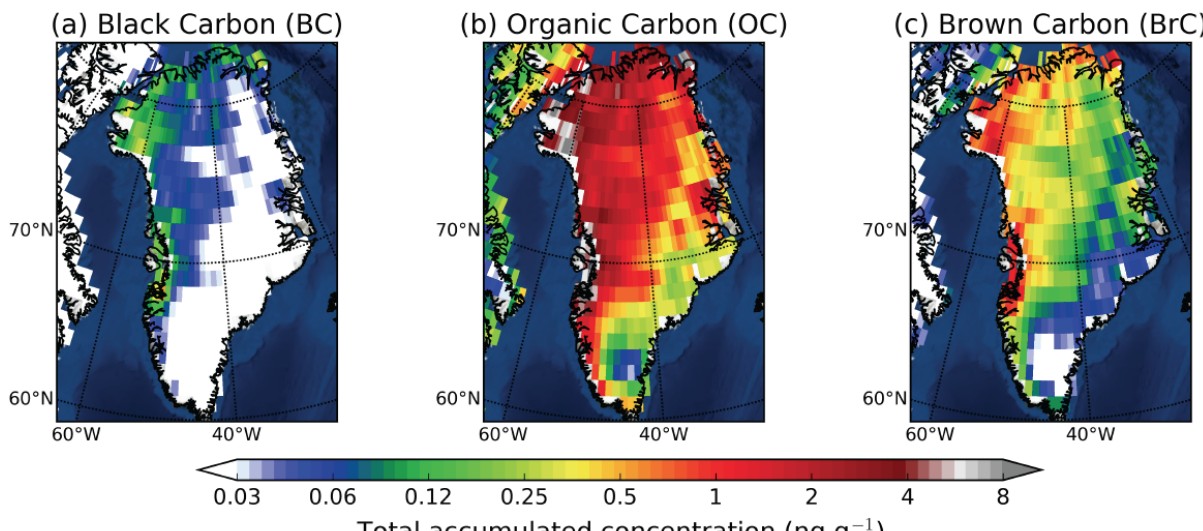

**Figure 3.** Calculated snow concentrations of (a) BC, (b) OC and (c) BrC over Greenland based on the modeled deposition and the snow precipitation (large scale and convective) adopted from the operational ECMWF data that were used in our simulation (see section 2.3).


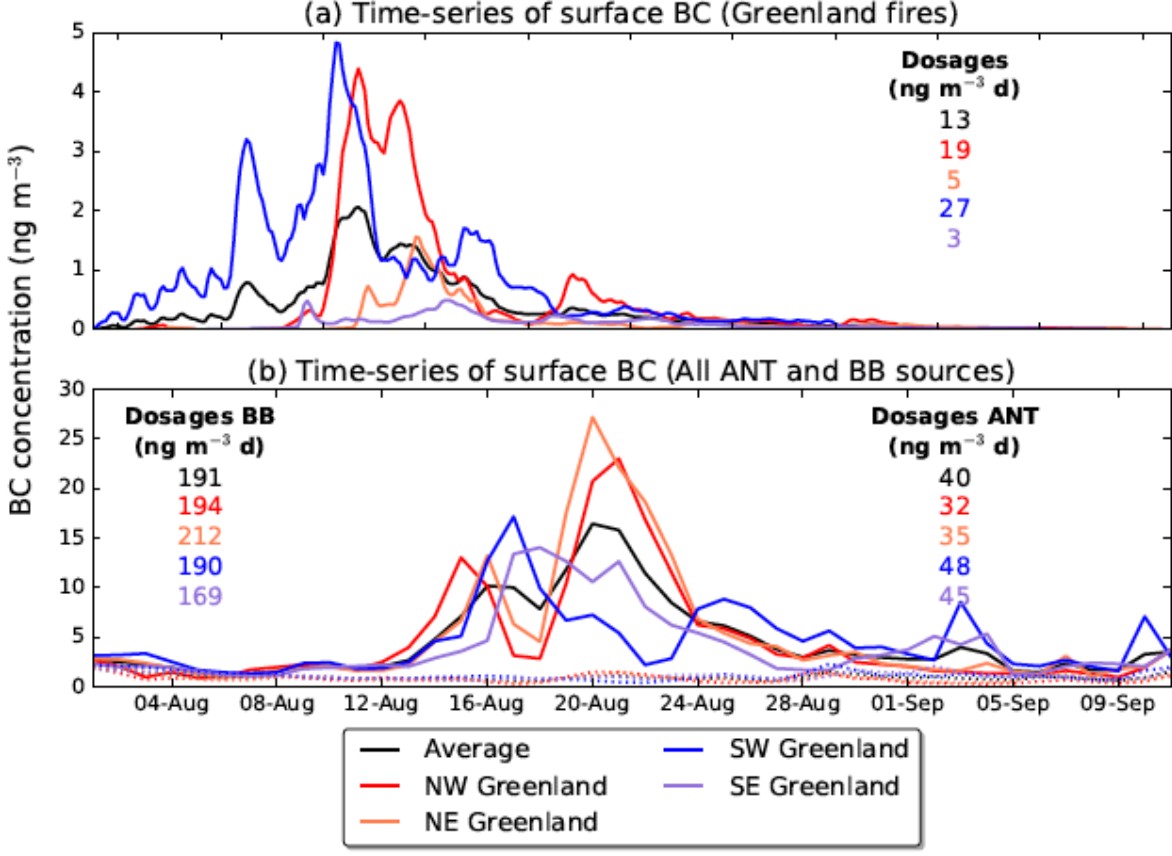


**Figure 4.** (a) Time-series of surface BC concentrations in Northwestern, Northeastern,
Southwestern and Southeastern Greenland from the summer 2017 fires in Western Greenland.
(b) Time-series of surface BC concentrations in Northwestern, Northeastern, Southwestern
and Southeastern Greenland from global anthropogenic (ANT, dashed lines) and biomass
burning (BB, solid lines) emissions for the same period. The numbers represent the respective
dosages (time-integrated concentrations) for the time period shown. The color codes are
reported in the legend.

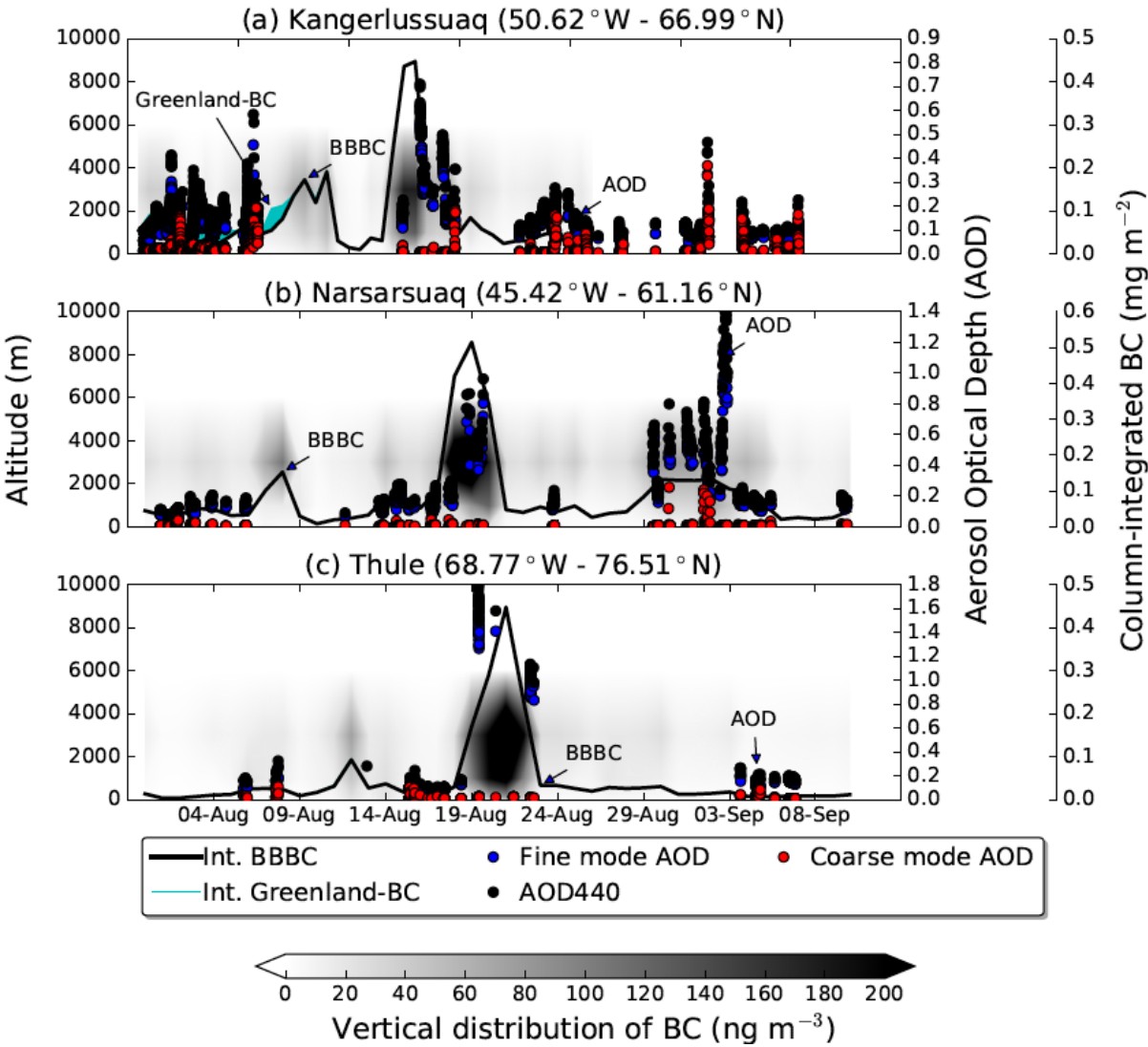

**Figure 5.** Contour plot of the vertical distribution of simulated BC (altitude a.g.l. shown on left y-axis) as a function of time (x-axis) and time-series of column-integrated simulated BC (extended right axis) from fires burning outside Greenland (black line) and Greenland fires (cyan stacked area). Column-integrated BC from anthropogenic sources was extremely small and it is not plotted here. Time-series for fine mode (blue) and coarse (red) AOD at 500 nm and total AOD at 400 nm (black) correspond to the right y-axis. The three panels show results for stations (a) Kangerlussuaq, (b) Narsarsuaq and (c) Thule (sorted from the closest to the farthest station).

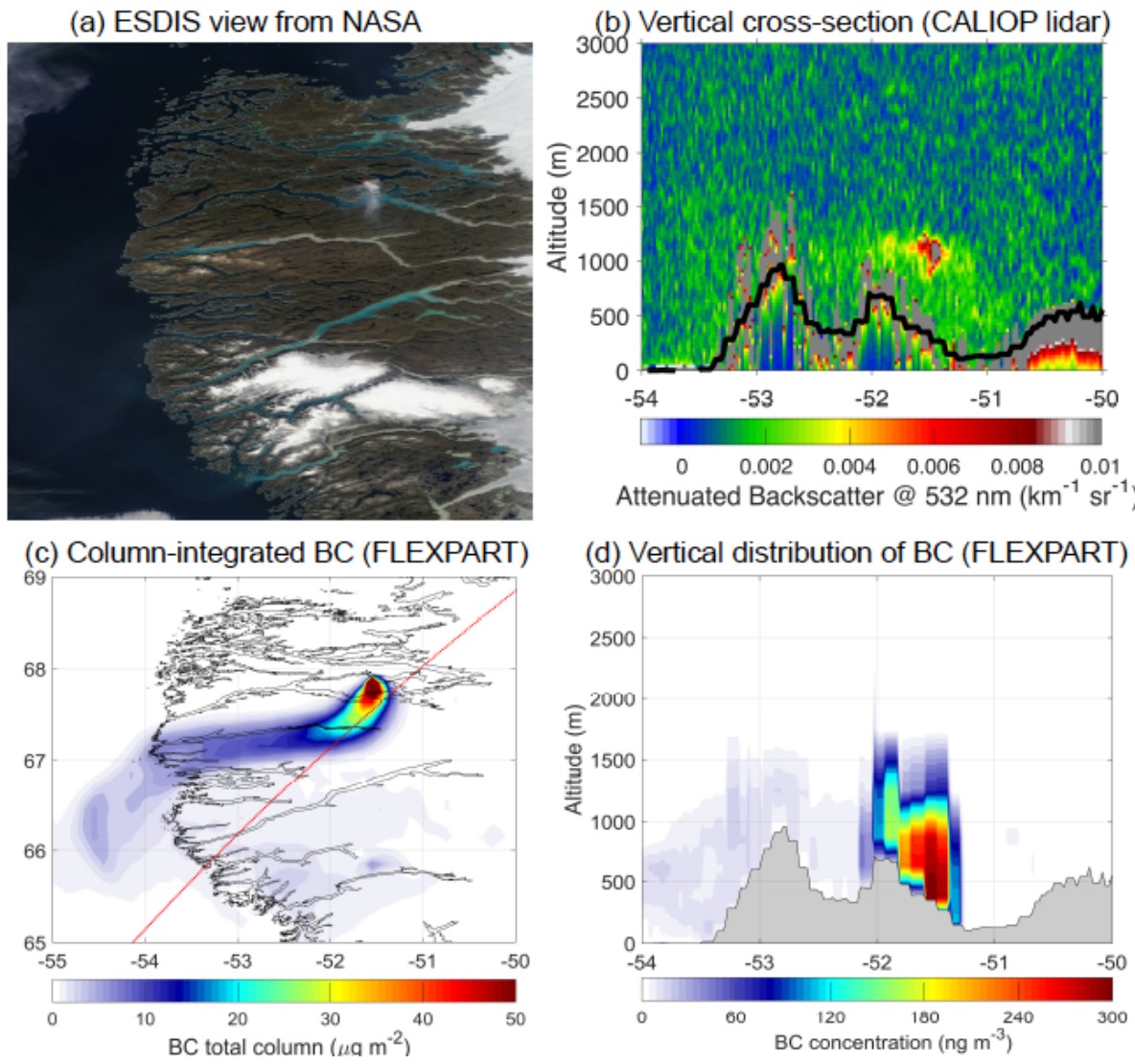

**Figure 6.** (a) Worldview application from the NASA/Goddard Space Flight Center Earth Science Data and Information System (ESDIS) project on 14 August 2017. (b) Vertical cross-section along satellite's route (red line in c) of total attenuated backscatter at a wavelength of 532 nm obtained from the CALIOP lidar on 14 August 2017 at 6 UTC (black line denotes the orography of the area). (c) Column-integrated BC concentration simulated with FLEXPART (read line shows the path of the satellite). (d) Vertical distribution of BC concentrations with longitude as seen with FLEXPART (grey area denotes the orography of the area).

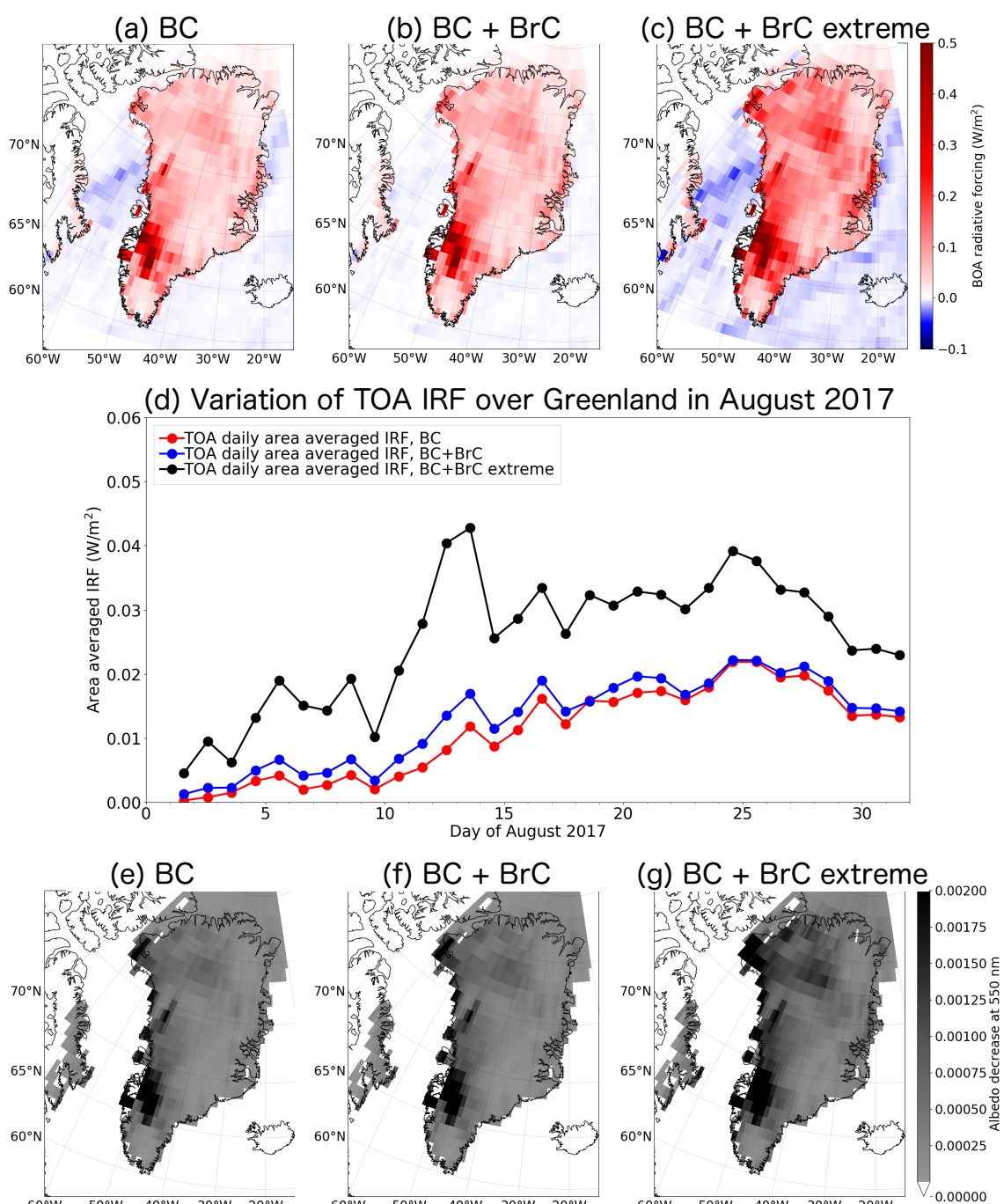


**Figure 7.** The instantaneous direct BOA RF due to (a) BC only, (b) BC and BrC, and (c) BC
and BrC when OC was assumed to be all BrC (extreme case) from the Greenland fire for
cloudy conditions on 31 August, 2017. (d) Daily variation of the TOA IRF over Greenland in
August 2017 for the three studied scenarios. Albedo reduction at 550 nm due to (e) BC only,
(f) BC and BrC, and (g) BC and BrC when OC was assumed to be all BrC (extreme case).
Note that the maximum albedo change due to deposited smoke is 0.00585 (BC only), 0.00590
(BC+BrC) and 0.00670 (BC+BrC extreme).

    **SUPPLEMENTARY FIGURE LEGENDS**


**Figure S 1.** Annual number of active fires over Greenland during the last 17 years as seen
from NASA's MODIS satellite (product MSC14DL).

**Figure S 2.** Fire dynamics in Greenland for the August 2017 fires according to MODIS
(magenta dots show active fire hot spots from the MODIS MCD14DL product). Locations of
stations with AOD measurements from AERONET are also shown.

**Figure S 3.** Median injection heights (km above sea level – ASL; left panel) and distribution
of longitudinally integrated burned biomass (Tg) as a function of injection altitude (right
panel) calculated by PRMv2 for the period between 31 July and 21 August 2017.
**Figure S 4.** Dry to total deposition ratio (example for BC) from the 2017 peat fires over
Greenland.

**Figure S 5.** Relative standard deviation of deposited mass (example for BC) for different
assumed size distributions normalized against the results from our reference size distribution
with a logarithmic mean diameter of 0.25 μm. Particle size distributions with aerodynamic
mean diameters of 0.1, 0.25, 0.5, 1, 2, 4, 8 μm and a logarithmic standard deviation of 0.3
were simulated.

**Figure S 6.** Footprint emissions sensitivities for Northwestern, Northeastern, Southwestern
and Southeastern Greenland for the period 31 July to 31 August 2017. Active fires from
NASA's MODIS MCD14DL product are shown with red dots.

**Figure S 7.** Average contribution of biomass burning (upper panels) and anthropogenic
emissions (lower panels) to surface concentrations of (a) BC and (b) OC in Northwestern,
Northeastern, Southwestern and Southeastern Greenland (in ng m$^{-3}$ per grid cell). Numbers (in
red) represent total concentrations in the studied domain, obtained by spatial integration over
all source grid cells. Receptor areas in Greenland are highlighted by pink boxes.

**Figure S 8.** (a) The single scattering albedo (SSA) of BC as a function of wavelength for
various modified combustion efficiencies (MCE). The star and dot marked lines are from the
parameterization of Pokhrel et al. (2016). (b) The IRF as a function of BC deposited on the
Ice Sheet. The calculations were made for cloudless conditions with a snow-covered surface
for noon on 31 August 2017 at 65°N.

**Table 1.** Start and end date of releases, source of data, type of sensor, burned area and daily increment of burned area, fuel consumption and calculated BC emissions from Eq. 1 during the Greenland fires in 2017. Total numbers for burned area, fuel consumption and BC emissions are highlighted in bold.

| Start | End | Source of RS data | Type of sensor | Burned area (ha) | Increment of burned area (ha) | Fuel consumption (t C) | BC emissions (kg) | OC emissions (kg) | BrC emissions (kg) |
|-------|-----|-------------------|----------------|------------------|-------------------------------|------------------------|-------------------|-------------------|--------------------|
| 31/07/17 | 02/08/17 | Sentinel 2A | MSI | 304 | 304 | 15176 | 3035 | 94543 | 18211 |
| 02/08/17 | 03/08/17 | Landsat 8 OLI | MSI | 428 | 125 | 6247 | 1249 | 38916 | 7496 |
| 03/08/17 | 04/08/17 | Sentinel 1A | SAR | 588 | 160 | 7980 | 1596 | 49712 | 9575 |
| 04/08/17 | 05/08/17 | Sentinel 1A | SAR | 740 | 152 | 7621 | 1524 | 47479 | 9145 |
| 05/08/17 | 07/08/17 | Sentinel 2A | MSI | 1100 | 359 | 17966 | 3593 | 111925 | 21559 |
| 07/08/17 | 08/08/17 | Sentinel 2A | MSI | 1314 | 214 | 10706 | 2141 | 66698 | 12847 |
| 08/08/17 | 12/08/17 | Landsat 8 OLI | MSI | 1868 | 554 | 27714 | 5543 | 172658 | 33257 |
| 12/08/17 | 14/08/17 | Sentinel 1A | SAR | 2005 | 136 | 6817 | 1363 | 42470 | 8180 |
| 14/08/17 | 15/08/17 | Sentinel 1A | SAR | 2169 | 165 | 8244 | 1649 | 51363 | 9893 |
| 15/08/17 | 16/08/17 | Sentinel 1A | SAR | 2209 | 40 | 1998 | 400 | 12444 | 2397 |
| 16/08/17 | 19/08/17 | Sentinel 1A | SAR | 2254 | 44 | 2213 | 443 | 13784 | 2655 |
| 19/08/17 | 21/08/17 | Sentinel 2A | MSI | 2345 | 92 | 4579 | 916 | 28530 | 5495 |
| **TOTAL** | | | | | **2345** | **117259** | **23452** | **730524** | **140711** |

RS - Remote Sensing
MSI - Multispectral Images
SAR - Synthetic Aperture RADAR