# Peer review of "Open fires in Greenland in summer 2017: transport,"

_Atmospheric Chemistry and Physics, 2018_

## Referee Comment (RC1) · Anonymous Referee #1 · 17 Apr 2018

Review of "Open fires in Greenland: an unusual event and its impact on the albedo of the Greenland Ice Sheet" by N. Evangeliou and co-authors.

General comments.

N. Evangeliou and co-authors present a paper dealing with the atmospheric emission of black carbon by peat fires in Greenland during an extreme event in August 2017. They estimate the total amount of BC released in the atmosphere and its impact on the atmospheric radiative balance and snow albedo. The authors conclude that none of those impact are really significant. I found the paper lacking a focused scientific objective and finally it will have a limited interest for the scientific community. The

methodology is sound but many of the assumptions must be clarify. The validation exercise is too qualitative while the dataset can be used for quantitative assessment. The conclusion that peat fire in Greenland could be of a significant importance for climate is not really supported by the findings of this paper.

Specific comments.

Abstract

Line 43. Your conclusion doesn't support this fact and it's not scientifically based.

Introduction.

The introduction is missing a comprehensive literature review on Arctic peat ecosystem and fire occurrence to better understand why those particular fires have been studied.

Line 83-84. Provide evidence of the significance of this event compared to other events.

Method

L89-118. This section is very important as it is the starting point for the estimation of the BC amount released by fires. However the methodology used (eg. which sensor, when, spatial resolution, who and how has done the estimation, . . .) is unclear. On Line 241, we can read that the burnt surface area comes from GlobeCover 2009. So finally, what is your point?

You should rewrite this section with a detailed comment of Table 1 and explain how it compares to active fire mapping. Line 118 needs clarification based on quantitative information.

L155 Explain how you get this number and provide a range of possible values

L180. Provide reference for BC density and size distribution. Peat fires emits large amount of organic carbon. The possible impact of the mixing state of BC and POM on aerosol size distribution, optical properties and residence time should be discuss in

this paper.

L200 and discussion section 3.3

The apportionment between emission from peat fires and other sources remains unclear for me. The methodology is not same as the one use for assessing impact of peat fire. The figure 4 is not really useful while other figures are in the supplement material.

L204 and section 4.2

The methodology and the discussion section on RF computation must be improved and clearly states how you deal with both surface albedo and atmospheric effect of BC on the radiative balance. Figure 7 is confusing as it deals with both BOA, TOA, time series and geographical distribution as the same time.

L218 and section 4.1 along with Figures 5 and 6. The validation exercise is really too qualitative and based on visual inspection of satellite data that are not really used scientifically. AERONET data can provide detailed information on aerosol optical properties and radiative forcing. CALIOP data products give aerosol extinction profiles which can be used in the RF computations.

L466 Your last bullet point is rather speculative and not supported by the findings of the paper.

---

## Referee Comment (RC2) · Anonymous Referee #2 · 25 Apr 2018

General comments :

This work investigates the quantification of emissions of black carbon (BC) from intense fires on peat lands in Western Greenland during summer 2017 and their impacts on albedo reduction and radiative forcing. The authors conclude that those impacts of BC deposition of the Greenland Ice Sheet are almost negligible, which turns out to be a scientific result for the community. This study is interesting and sound for ACP. I have nevertheless several criticisms requiring a careful and revision and in-depth improvements both in the methodology, often unclear, and in the discussion of the results before the paper is suitable for publication in ACP.

[Figure]

Specific comments :

L1-2 : The title seems to indicate that the main focus of the paper is the quantification of the reduction in albedo due to open fires in Greenland. Only ten lines in the paper really focus on the modification of the albedo due to BC deposition. The title should reflect the main findings of the paper : quantification of BC emissions of this unusual event, transport of the plume, deposition.

L41-44 and L496-500 : I find a bit strange to conclude both abstract and conclusion by something purely speculative and that does not match the main results of the paper.

L83-84 : "the largest fires". Give maybe statistics or cite a climatological study to support this assertion.

L111 : The authors should give more details about the procedure applied on the data. "Additional classification" is too vague.

L130 : "assuming a 6h persistence". How is this hypothesis justified ? Is it confirmed b observations or by other studies ?

L161 : Say clearly that the only variable computed in this study from measurements is the burned area A. The other factors are based on assumptions or provided by previous studies.

L181 : Those values suggest that aerosols are not only composed of BC (which is a reasonable assumption). How do the authors justify this size distribution ? It has indeed a huge influence on the deposition efficiencies (both sedimentation and wet removal) and on the calculation of aerosol optical properties. Both the radiative forcings and reduction of albedo on snow surfaces will be sensitive to this assumption on the size distribution. I suggest that the authors perform a sensitivity study on the influence of those parameters.

L200 : "a simple emission scheme". What does it mean ? Why don't the authors use the same methodology for all fires ?

L200-201 : Those emission factors should depend on the type of soil and vegetation. Which maps have been used here ? Which values for emission factors have been finally chosen ? The reader should be able to reproduce the results of this study ; without such assumptions, it is impossible.

Sect. 2.4 : Do the authors calculate radiative forcing assuming refractive index of BC only ? The choice of the refractive index should be done in accordance with the size distribution (L181), which probably reflects an internal mixture of aerosols.

L226 : "we display" : where ?

L292 : "a small portion of the emitted BC". Please quantify it.

L334 : "due to the generally dry weather when the fires were burning". It can be also ascribed to the fact that dry deposition mostly occurs in the quasi-laminar sublayer close to the surface. Aerosols are quickly deposited close to the sources before being injected at higher altitudes and being transported away from sources.

L365 : "the anthropogenic contribution is larger". For the sake of clarity, the authors might write that the anthropogenic is relatively larger in Southern Greenland in contrast to Northern Greenland but remains lower than the biomass burning contribution.

L367 : "the BC concentrations that are calculated here for the studied fire period are relatively high compared to those reported previously". I am not sure this is always true. The authors should also quote more recent studies, e.g. Polashenski et al. (2015), Legrand et al. (2016) or Thomas et al. (2017), who have reported higher events of biomass burning BC deposition over Greenland. If the BC deposited on snow/ice surfaces is much larger in those studies, it also suggests higher surface BC concentrations.

L378 and L389 : "dosages". Do you mean concentrations / mixing ratios ?

L397-398 : BC particles are probably not the main contributors to AOD in this region for two reasons : the BC loadings are rather low in comparison to other aerosol compounds and the diameter of BC-containing particles is much smaller than the wavelength (0.5 um). A better proxy of the temporal evolution of the integrated BC would be the absorbing AOD (AAOD), which is also often provided at AERONET stations. The AAOD/AOD would be also a good indicator of the contribution of BC to the total AOD (even if BC is not the only absorbing component). This should be shown on Fig. 5.

L401-407 : How do the authors explain the significant AOD enhancement at the beginning of September observed at Narsarsuaq station ?

L422 : "was not studied". Does it mean that the transport of those North American fire plumes was not correctly captured by FLEXPART ? It is indeed impossible to see on Fig. 6d as the vertical scale is not appropriate.

Sect. 4.2 : The authors should remind that they calculated only the forcing due to the Greenland fires, which is itself small compared to the North American or Eurasian fires. It should also be said explicitly that the calculated radiative forcing values does not include semi-direct nor indirect effects, which may be dominant here.

L436 : "cloudless conditions". I do not understand the purpose of this. It is only an ideal simulation, which is not commented in the paper afterwards. What does it bring to the discussion ?

L440-442 : It is not clear if the given values refer to the total radiative forcing of BC. What are the relative contributions of the direct radiative forcing of BC and of the radiative forcing of BC deposited on snow surfaces ? The authors also give the values without any uncertainty, but a lot of assumptions have been done to retrieve the BC emissions, the BC size distribution, the BC optical properties. Each of those hypothesis would lead to a range of values of IRF.

L 442: "Fig 7c depicts the temporal behaviour..." Does it represent calculations in cloudy conditions ?

L443-444 : I don't see how this information (blue line) can be useful. The location of

the pixel where the maximum IRF is found likely varies with time. Besides the analysis of this figure is not done in text. I recommend to remove it.

L448-455 : If the authors want to be able to compare their results to global studies, as it is done here, they need to multiply the value of RF by the area of the simulation domain to obtain a forcing value in watts, and then divide it by the surface area of the Earth to obtain an equivalent global radiative effect in mW/m2 that could be compared to results for global studies.

L453-455 : What about the impact of North American and Eurasian fires, whose plumes reach Greenland during the studied period ?

L456-457 : What is the albedo reduction due to BC deposition that can be ascribed to Greenland fires / to fires outside Greenland / to anthropogenic sources ? If the goal of the paper is indeed to focus on the impact of the Greenland fires, quantifying this effect and comparing it to the relative contribution of the different sources would be really valuable for the paper. The authors should also compare their albedo reduction values to previous studies, e.g. Polashenski et al. (2015).

Sect. 5 : The conclusions may be more quantitative. For example : L478-479 : the ratio of BC deposition from the different sources can be given L481-483 : the AOD enhancement can be precised L488 : "albedo change due to the BC deposition". Which sources have been considered ?

L496-500 : Remove this purely speculative sentence. The opposite could also be said, given the findings of the paper.

The choice of the figures kept in the manuscript is rather strange. Most useful figures relevant for the discussion have been displaced to the Supplementary Material. I recommend to move them to the main paper.

Fig. 2a: Are those values averaged over the simulation domain ? over Greenland ? I had hard time to figure out how those values could be realistic. I think there is either

a issue with the unit or a mistake in the calculation. Shouldn't it be ng/m3 or ng/kg instead of ug/m3 ? The total concentrations of BC in the domain should be calculated as the volume average of the grid cell concentrations, not the sum over all grid cells in the domain...

Fig 2b : Here again, there is an issue with the unit. The color bar indicates ug/m2 (which is probably right), but the caption says ng/m2. Which one is correct ?

Fig. 4 : It is extremely difficult to see the colored grid cells an read their values. Please improve the quality of this figure.

Fig. 5 : Does the altitude represent agl or amsl ? The orography in Greenland is not flat.

Fig 5 : Why do you keep the contribution of fires burning outside Greenland but exclude the BC contribution of anthropogenic sources ? According to Fig. 4, their contribution is absolutely not negligible and they might modify the time series of column-integrated BC in Greenland.

Fig. 6 : it would be better to use the same scale for longitude and altitude on panels (b) and (d).

Fig. 7c : Is the snow albedo reduction plotted for 31 August or for the full period ?

Table 1 : This table is not commented nor anlyzed in text. We can notice changes in the sources of RS data at different periods, which should be detailed in the methodology section.

Legrand, M., et al. (2016), Boreal fire records in Northern Hemisphere ice cores: A review, Clim. Past, 12(10), 2033–2059.

Polashenski, C. M., J. E. Dibb, M. G. Flanner, J. Y. Chen, Z. R. Courville, A. M. Lai, J. J. Schauer, M. M. Shafer, and M. Bergin (2015), Neither dust nor black carbon causing apparent albedo decline in Greenland's dry snow zone: Implications for MODIS C5 surface reflectance, Geophys. Res. Lett., 42, 9319–9327, doi:10.1002/2015GL065912.

Thomas, J. L., et al. (2017), Quantifying black carbon deposition over the Greenland ice sheet from forest fires in Canada, Geophys. Res. Lett., 44, 7965–7974, doi:10.1002/2017GL073701.

Technical comments :

L350 : "adopted". Do you mean "adapted" ?

L394 : Replace "for validating" by "to validate".

L485 : Replace "attenuation" by "attenuated"

L512 : Please write "Brent Holben" in two words.
* * *

---

## Author Comment (AC1) · 21 Jun 2018

Review of "Open fires in Greenland: an unusual event and its impact on the albedo of the Greenland Ice Sheet" by N. Evangeliou and co-authors.

General comments. N. Evangeliou and co-authors present a paper dealing with the atmospheric emission of black carbon by peat fires in Greenland during an extreme event in August 2017. They estimate the total amount of BC released in the atmosphere and its impact on the atmospheric radiative balance and snow albedo. The

authors conclude that none of those impact are really significant. I found the paper lacking a focused scientific objective and finally it will have a limited interest for the scientific community. The methodology is sound but many of the assumptions must be clarify. The validation exercise is too qualitative while the dataset can be used for quantitative assessment. The conclusion that peat fire in Greenland could be of a significant importance for climate is not really supported by the findings of this paper.

–Response: We agree that the methodology and parts of the discussion needed lots of improvement and we have made substantial effort with numerous changes in all parts of the manuscript according to both reviewers' suggestions (please see manuscript with Track Changes). However, we do not agree with this description of our work. The validation is qualitative because no direct measurements of BC concentrations exist from this event occurring in a particularly data-sparse region, and also few satellite data document the event. The only data we found are Lidar data from CALIPSO that confirmed the presence of the plume where our model predicted it. Could the reviewer suggest, in concrete terms, which dataset could be used for quantitative assessment? The reviewer says, "The conclusion that peat fire in Greenland could be of a significant importance for climate is not really supported by the findings of this paper." This is NOT a conclusion, but a logical probability, considering that 25% of Greenland's surface is permafrost that is rich in peat. We now show this more clearly in the updated version of our manuscript. In addition, NASA's satellites show an increasing trend of fires in thawed permafrost over Greenland (see new supplementary figure S1 or attached Fig. 1) and our simulations showed that 30% of the emissions were deposited in the Greenland Ice Sheet (Lines 388-391). We disagree with the comment of the reviewer that this paper will have a limited interest for the scientific community. We present some statistics from the ACP Discussions website. In the Discussions page of the journal (https://www.atmos-chem-phys-discuss.net/discussion_papers.html), at the time that we started writing this response (22-05-2018), there are 30 papers in open discussion (ACPD) that were published the same time as ours (March 2018). If we calculate the average views and downloads we get 302±100 (min: 199 – max: 570),

while our paper's visibility is 293. Furthermore, although media coverage does not converge with scientific quality, the present study was selected for a press conference on "Shape of things to come? The 2017 wildfire season" during the EGU 2018 conference (https://client.cntv.at/egu2018/pc5).

Specific comments. Abstract Line 43. Your conclusion doesn't support this fact and it's not scientifically based.

–Response: Line 43 states "If the expected further warming of Greenland produces much larger fires in the future, this could indeed cause substantial albedo changes and thus lead to accelerated melting of the Greenland Ice Sheet." This sentence is NOT a conclusion, but a logical hypothesis (if). We have slightly rephrased the sentence, so it now reads: "If the expected future warming of the Arctic produces more fires in Greenland, this could indeed cause albedo changes and thus contribute to accelerated melting of the Greenland Ice Sheet." Finally, in order to prove that this is not pure speculation but a solid hypothesis, we support it with references (see last paragraph in conclusions).

Introduction. The introduction is missing a comprehensive literature review on Arctic peat ecosystem and fire occurrence to better understand why those particular fires have been studied.

–Response: We have focused our introduction on peatlands and fires in Greenland and think that a more comprehensive literature review on Arctic peat ecosystems in general is out of scope of this paper. After all, this paper studies the impact of fires in Greenland on BC concentration and deposition in Greenland, not on future scenarios of fire occurrence, permafrost melt or such.

Line 83-84. Provide evidence of the significance of this event compared to other events.

–Response: Our statement that ". . . the fires . . . , probably represent the largest fires that have occurred on Greenland in modern times.", is now supported by a new plot of

the number of MODIS active fire detections (MODIS MCD14DL) over Greenland (see new supplementary Figure S1 or attached Fig.1).

Method L89-118. This section is very important as it is the starting point for the estimation of the BC amount released by fires. However the methodology used (eg. which sensor, when, spatial resolution, who and how has done the estimation, . . .) is unclear. On Line 241, we can read that the burnt surface area comes from GlobeCover 2009. So finally, what is your point?

–Response: Line 241 has been corrected. We appreciate the reviewer for this constructive comment. As regards to the methodology, we have done a few corrections to explain better what has been done, also giving specifications of the products we used (lines 97-99). In our opinion, detailed explanations on the calculations are not needed, since the method has been already published in the relevant literature and used in many other previous cases. As we explain in the manuscript, the burned area was mapped using severity levels of dNBR index. The methodology of its application is described in details in Lutes et al. (2006) (pp. 201-270), which is attached. There is another paper describing how dNBR was calibrated in field - Escuin et all, 2008 (see reference in the manuscript). Since the index is sensitive to any disturbances, we applied a manually delineated fire perimeter to increase the accuracy of mapping.

You should rewrite this section with a detailed comment of Table 1 and explain how it compares to active fire mapping. Line 118 needs clarification based on quantitative information.

–Response: Line 94 explicitly says that the location of the active fires were downloaded from NASA's website. So, what is shown in Table 1 has been confirmed with NASA's active fires (also shown in supplements' Figure S1 and attached Fig.1). Regarding to the severity levels (Line 118), qualitative information is given in Key and Benson (2006) together with all the details of the methodology used. The same methodology has been used to map the Chernobyl fires (see: Evangeliou et al., 2014; 2015; 2016) The

comment to confirm Line 118 based on quantitative information is too generic and we do not really understand what the reviewer wants us to do.

L155 Explain how you get this number and provide a range of possible values

–Response: Line 155 says "In contrast, tropical peatlands can have deep burn depths of 40–50 cm and release an average of 300–450 t C ha−1 (Page et al., 2015; Reddy et al., 2015)." It should be obvious that this range of values was reported by Page et al. (2015) and Reddy et al. (2015). If the reviewer means the average amount of organic fuel available for combustion that we used for the Greenland fires (100 t C ha-1), it has been taken from Smirnov et al. (2015). In this paper, it was assumed that for peat-bog fires, the average amount of fuel available for combustion (including the soil organic matter) is up to 120 t/ha supported from measurements from IPCC (2006).

L180. Provide reference for BC density and size distribution. Peat fires emits large amount of organic carbon. The possible impact of the mixing state of BC and POM on aerosol size distribution, optical properties and residence time should be discuss in this paper.

–Response: We agree that fires also emit large quantities of organic carbon (OC). However, the impact of OC on the albedo of the ice sheet is probably small, although it probably enhances the BC effect, since OC can also be slightly absorbing (e.g., brown carbon). But given the lack of information on the optical properties of the emitted OC, we think an additional analysis of OC would not be very meaningful. With respect to BC density and size distribution, a reference was added in Line 214 of the updated manuscript. We have now also performed a sensitivity study on the impact of different particle size distributions on the deposition of BC over Greenland's Ice Sheet and discuss it in section 3.2. A detailed analysis of residence times of BC has been already presented by Grythe et al. (2017) [reference in the manuscript] and in Evangeliou et al. (2018) [reference under editorial check in ACP Discussions].

L200 and discussion section 3.3 The apportionment between emission from peat fires

and other sources remains un- clear for me. The methodology is not same as the one use for assessing impact of peat fire. The figure 4 is not really useful while other figures are in the supplement material.

–Response: Lagrangian models such as the one used in our work (FLEX-PARTv10) can run forward in time (like CTMs or climate models) using specific emissions that can be taken from an existing inventory (for example ECLIPSE, see: http://www.iiasa.ac.at/web/home/research/researchPrograms/air/Global_emissions.html). Moreover, Lagrangian models have the advantage that they can also run backward in time, from a specific point or region for which the user wants to calculate concentrations. What is produced then is the footprint emission sensitivity (or footprint), which is simply the residence time of the computational particles (in sec) in each grid-cell of the model. Then, by multiplying this footprint with a given emission inventory (e.g. ECLIPSE) given in kg/m2/s and dividing with the altitude of the lowest vertical level in the model, one obtains surface concentrations again. Notice that forward and backward calculations are equivalent, so the methodology is not different. However, depending on the setup, the computational efficiency can be much higher in backward mode, and that is also the reason we used it to assess the impact of emissions outside Greenland. For FLEXPART that we used in this study, a comparison between forward and backward simulations can be found in Seibert and Frank (2004). We calculate average concentrations of surface BC in four compartments of Greenland based on ECLIPSE emissions. ECLIPSE includes all anthropogenic sources, while we calculate biomass burning emissions using global MODIS-satellite hot spot data (Giglio et al., 2016) and GFAS (references in the manuscript). Everything is well documented in the associated references. Figure 4 has been replaced by Figure S4 as suggested.

L204 and section 4.2 The methodology and the discussion section on RF computation must be improved and clearly states how you deal with both surface albedo and atmospheric effect of BC on the radiative balance. Figure 7 is confusing as it deals with both BOA, TOA, time series and geographical distribution as the same time.

–Response: We have re-written and re-structured the whole chapter, both in the Methodology and analysis of the Results (see manuscript with Truck Changes). Our perception is not to present in detail methods that have been documented in previous publications. For RF calculations we used the uvspec model from the libRadtran radiative transfer software package (http://www.libradtran.org/doku.php) (see references in the manuscript: Emde et al., 2016; Mayer and Kylling, 2005). Snow albedo was calculated with the SNICAR model (http://snow.engin.umich.edu/info.html) in a two-layer configuration (see references in the manuscript: Flanner et al., 2007, 2009). These are open source codes that have been used by many groups worldwide. Figure 7 has been improved as suggested.

L218 and section 4.1 along with Figures 5 and 6. The validation exercise is really too qualitative and based on visual inspection of satellite data that are not really used scientifically. AERONET data can provide detailed information on aerosol optical properties and radiative forcing. CALIOP data products give aerosol extinction profiles which can be used in the RF computations.

–Response: The reason the validation exercise is so qualitative is that we have no clear observations of the Greenland fire plume. The AERONET data show impacts of the forest fires burning outside Greenland. Only at one site, the AERONET data show an AOD increase that is partly (but not exclusively) due to the Greenland fires.

L466 Your last bullet point is rather speculative and not supported by the findings of the paper.

–Response: This is true and we have now corrected it. The last bullet is NOT ALL OF IT a conclusion, but rather a comment and therefore, we now show it as a comment (not bulleted) below the bullet. We further support what we say in the sentence with references.

REFERENCES

Lutes, Duncan C.; Keane, Robert E.; Caratti, John F.; Key, Carl H.; Benson, Nathan C.; Sutherland, Steve; Gangi, Larry J. 2006. FIREMON: Fire effects monitoring and inventory system. Gen. Tech. Rep. RMRS-GTR-164-CD. Fort Collins, CO: U.S. Department of Agriculture, Forest Service, Rocky Mountain Research Station. (ATTACHED)

Key, C. H. and Benson, N. C.: Landscape assessment: Sampling and analysis methods, USDA For. Serv. Gen. Tech. Rep. RMRS-GTR-164-CD, (June), 1–55, doi:10.1002/app.1994.070541203, 2006.

Evangeliou, N., Balkanski, Y., Cozic, A., Hao, W. M. and Møller, A. P.: Wildfires in Chernobyl-contaminated forests and risks to the population and the environment: A new nuclear disaster about to happen?, Environ. Int., 73, 346–358, doi:10.1016/j.envint.2014.08.012, 2014.

Evangeliou, N., Balkanski, Y., Cozic, A., Hao, W. M., Mouillot, F., Thonicke, K., Paugam, R., Zibtsev, S., Mousseau, T. A., Wang, R., Poulter, B., Petkov, A., Yue, C., Cadule, P., Koffi, B., Kaiser, J. W., Møller, A. P. and Classen, A. T.: Fire evolution in the radioactive forests of Ukraine and Belarus: Future risks for the population and the environment, Ecol. Monogr., 85(1), 49–72, doi:10.1890/14-1227.1, 2015.

Evangeliou, N., Zibtsev, S., Myroniuk, V., Zhurba, M., Hamburger, T., Stohl, A., Balkanski, Y., Paugam, R., Mousseau, T. A., Møller, A. P. and Kireev, S. I.: Resuspension and atmospheric transport of radionuclides due to wildfires near the Chernobyl Nuclear Power Plant in 2015: An impact assessment., Sci. Rep., 6, 26062 [online] Available from: http://www.nature.com/srep/2016/160517/srep26062/full/srep26062.html, 2016.

Smirnov, N. S., Korotkov, V. N. and Romanovskaya, A. A.: Black carbon emissions from wildfires on forest lands of the Russian Federation in 2007–2012, Russ. Meteorol. Hydrol., 40(7), 435–442, doi:10.3103/S1068373915070018, 2015 2006 IPCC Guidelines for National Greenhouse Gas Inventories, Vol. 4: Agriculture, Forestry and Other Land Use (IPCC, 2006) [in Russian].

Seibert, P. and Frank, A.: Source-receptor matrix calculation with a Lagrangian particle dispersion model in backward mode, Atmos. Chem. Phys., 4(1), 51–63, doi:10.5194/acp-4-51-2004, 2004.

Giglio, L., Descloitres, J., Justice, C. O. and Kaufman, Y. J.: An enhanced contextual fire detection algorithm for MODIS, Remote Sens. Environ., 87(2–3), 273–282, doi:10.1016/S0034-4257(03)00184-6, 2003.

[Figure]

[Figure]

**Fig. 1.** Annual number of active fires over Greenland during the last 17 years as seen from NASA's MODIS satellite (product MSC14DL).

---

## Author Comment (AC2) · 21 Jun 2018

General comments : This work investigates the quantification of emissions of black carbon (BC) from intense fires on peat lands in Western Greenland during summer 2017 and their impacts on albedo reduction and radiative forcing. The authors conclude that those impacts of BC deposition of the Greenland Ice Sheet are almost negligible, which turns out to be a scientific result for the community. This study is interesting and sound for ACP. I have nevertheless several criticisms requiring a careful and revision and in-depth improvements both in the methodology, often unclear, and in the discussion of

the results before the paper is suitable for publication in ACP.

–Response: We acknowledge the reviewer's comments and his effort to improve this manuscript. We have tried to follow his suggestions to correct the manuscript and have basically re-written parts of the manuscript (please see manuscript with Track Changes).

Specific comments : L1-2 : The title seems to indicate that the main focus of the paper is the quantification of the reduction in albedo due to open fires in Greenland. Only ten lines in the paper really focus on the modification of the albedo due to BC deposition. The title should reflect the main findings of the paper : quantification of BC emissions of this unusual event, transport of the plume, deposition.

–Response: We agree; we have changed the title to "Open fires in Greenland in summer 2017: transport and deposition of BC and impact on the Greenland Ice Sheet"

L41-44 and L496-500 : I find a bit strange to conclude both abstract and conclusion by something purely speculative and that does not match the main results of the paper.

–Response: We admit that this is probably an extreme formulation and we have changed it to a weaker statement. We would like to draw attention that this statement is not a conclusion, but a logical hypothesis. To further show that it's not a conclusion, we now support the paragraph with references (see last paragraph in conclusions).

L83-84 : "the largest fires". Give maybe statistics or cite a climatological study to support this assertion.

–Response: We have plotted the annual number of active fires from NASA's MODIS product in supplements' Figure S1 (or Fig.R1) starting from first year that satellite data were available (2001).

L111 : The authors should give more details about the procedure applied on the data. "Additional classification" is too vague.

–Response: The statement has been removed!

L130 : "assuming a 6h persistence". How is this hypothesis justified ? Is it confirmed b observations or by other studies ?

–Response: Well, this is confirmed by previous studies (Kaiser et al., 2012 – reference in the paper). We chose a persistence model similar to what is done in Kaiser et al. (2012) and used a time of the same order of magnitude with the mean return time of MODIS in the afternoon (peak time of fire) $\sim$ 4h. For a description of the persistence model that was used, please see line 8 - page 9852 of Paugam et al. (2015).

L161 : Say clearly that the only variable computed in this study from measurements is the burned area A. The other factors are based on assumptions or provided by previous studies.

–Response: Corrected. This is now explicitly mentioned in Line 184.

L181 : Those values suggest that aerosols are not only composed of BC (which is a reasonable assumption). How do the authors justify this size distribution ? It has indeed a huge influence on the deposition efficiencies (both sedimentation and wet removal) and on the calculation of aerosol optical properties. Both the radiative forcings and reduction of albedo on snow surfaces will be sensitive to this assumption on the size distribution. I suggest that the authors perform a sensitivity study on the influence of those parameters.

–Response: After rapid coagulation, more than 90% of the mass of BC after fires is present in sizes between $0.1 - 1$ $\mu$m in the atmosphere. This has been highlighted by many experiments/measurements and is now well justified in section 3.2. However, we have followed the suggestion of the current reviewer and performed a sensitivity study using different size distribution of the BC particles produced from the 2017 fires in Greenland and we calculated the uncertainty on the deposited mass of BC due to different size distribution. We present and discuss the results at the end of section 3.2.

The effect of different size distribution on residence times has been already studied by Grythe et al. (2017) [reference in the manuscript] and the different deposition coefficients in Evangeliou et al. (2018) [reference under editorial check in ACP Discussions]. The calculated uncertainty from this sensitivity test ranges from 10%–30% in 86% of the Sheet's surface to up to 50% in the rest of the Sheet's surface.

L200 : "a simple emission scheme". What does it mean? Why don't the authors use the same methodology for all fires ?

–Response: We appreciate reviewer's comment here. This was a typo error and we have updated this part of the methodology.

L200-201 : Those emission factors should depend on the type of soil and vegetation. Which maps have been used here ? Which values for emission factors have been finally chosen ? The reader should be able to reproduce the results of this study ; without such assumptions, it is impossible.

–Response: Corrected; See previous comment.

Sect. 2.4 : Do the authors calculate radiative forcing assuming refractive index of BC only? The choice of the refractive index should be done in accordance with the size distribution (L181), which probably reflects an internal mixture of aerosols.

–Response: The radiative forcing was calculated using the refractive index of BC only. We agree with the reviewer that BC was likely present as an internal mixture with other aerosol components (especially OC). However, we did not simulate OC and therefore used only the refractive index of BC. This will lead to an underestimation of the atmospheric effects of BC, since internal mixing with OC will likely enhance the BC absorption, and there may also be other absorbing components in the aerosol. However, we think that as an order of magnitude estimation of the atmospheric effects, our assumption should be sufficient. Furthermore, the more important impact of BC on the albedo is less (or not) sensitive to the mixing state of the aerosols.

L226 : "we display" : where ?

–Response: We substituted 'display' with 'used'.

L292 : "a small portion of the emitted BC". Please quantify it.

–Response: We have quantified the portion that lifted up in this particular day (≈516 kg).

L334 : "due to the generally dry weather when the fires were burning". It can be also ascribed to the fact that dry deposition mostly occurs in the quasi-laminar sublayer close to the surface. Aerosols are quickly deposited close to the sources before being injected at higher altitudes and being transported away from sources.

–Response: Thanks for this comment. We have included it in the manuscript.

L365 : "the anthropogenic contribution is larger". For the sake of clarity, the authors might write that the anthropogenic is relatively larger in Southern Greenland in contrast to Northern Greenland but remains lower than the biomass burning contribution.

–Response: Comment was added to the manuscript.

L367 : "the BC concentrations that are calculated here for the studied fire period are relatively high compared to those reported previously". I am not sure this is always true. The authors should also quote more recent studies, e.g. Polashenski et al. (2015), Legrand et al. (2016) or Thomas et al. (2017), who have reported higher events of biomass burning BC deposition over Greenland. If the BC deposited on snow/ice surfaces is much larger in those studies, it also suggests higher surface BC concentrations.

–Response: We thank the reviewer for providing the references and have added them to the previous section. Please see Line 425-426 for Polashenski et al (2015) and Legrand et al. papers. However, the Thomas et al. paper is using another unit (g/m2) and without knowing the density of the samples no conversion to ng/g (units used in

the present) can be applied.

L378 and L389 : "dosages". Do you mean concentrations / mixing ratios ?

–Response: They are dosages of concentrations. It is now explained in the last paragraph of section 3.3 and in the caption of the respective Figure.

L397-398 : BC particles are probably not the main contributors to AOD in this region for two reasons : the BC loadings are rather low in comparison to other aerosol compounds and the diameter of BC-containing particles is much smaller than the wavelength (0.5 um). A better proxy of the temporal evolution of the integrated BC would be the absorbing AOD (AAOD), which is also often provided at AERONET stations. The AAOD/AOD would be also a good indicator of the contribution of BC to the total AOD (even if BC is not the only absorbing component). This should be shown on Fig. 5.

–Response: The reviewer has a very good point here and we tried to retrieve AAOD data as he suggested. Though in Kangerlussuaq and Thule no AAOD Level 2 data are available for July-September 2017, while in Narsarsuaq AAOD Level 2 data are available for 2 September 2017 (when the fires had been already extinguished). In Andrews et al. (2017) paper is stated that "One obvious limitation of the AERONET inversion retrievals is that the uncertainty of the derived SSA becomes very large at low values of AOD (Dubovik et al., 2000). To minimize the effects of this uncertainty, the AERONET Level 2 data invalidate all absorption-related values if the AOD at wavelength 440 nm (AOD440) is below 0.4 (Dubovik et al., 2000, 2002; Holben et al., 2006)." In page 6043 of the same paper it is stated that "It should be noted that AERONET does not recommend the use of absorption-related parameters (e.g., SSA, AAOD and complex index of refraction) at AOD440 below 0.4." In our case, except for the characteristic peak of AOD that is attributed to the N. American fires, all the other AOD values were below 0.4. Sometimes researchers use AAOD LEV 1.5 data, but these get high uncertainty (see Andrews et al, 2017). In page 6051 of the Andrews et al. (2017) paper is also stated that "Using the sum-of-squares propagation of errors to calculate the uncertainty in AAOD for both high and low AAOD cases results in an AAOD uncertainty of approximately 0.015 for both high- and low-AOD cases . . . An AAOD uncertainty value of 0.015 suggests an uncertainty of about 60% in AAOD for AOD440 D0.5 and more than 140% uncertainty in AAOD for AOD440 < 0.2." Therefore, we do not think that these uncertain LEV1.5 AAOD measurements should be plotted instead of AOD here. However, if the reviewer or the editor disagree, we have retrieved them and we could use them in a next step. Besides, we only used AOD as an indicator for the presence of the plume, and for that purpose it should be sufficient.

L401-407 : How do the authors explain the significant AOD enhancement at the beginning of September observed at Narsarsuaq station ?

–Response: As the reviewer can see, in the attached Fig.R2 we present the biomass burning BC from GFAS (upper panel) in the beginning of September and the footprint emission sensitivity from the Narsarsuaq station (bottom panel) on September 3rd. We observe that the highest footprint emission sensitivity is located exactly at the place where GFAS emissions are the highest (Canada). Therefore, we have a clear indication that the increase that the reviewer mentioned is due to the Canadian wildfires.

L422 : "was not studied". Does it mean that the transport of those North American fire plumes was not correctly captured by FLEXPART ? It is indeed impossible to see on Fig. 6d as the vertical scale is not appropriate.

–Response: Here, we wanted to state that the existence of the N. American fires in the attenuated backscatter measurements that we get from CALIOP was not further studied. The study of the N. American fires is beyond the scope of this paper. We have used a better formulation in the manuscript now.

Sect. 4.2 : The authors should remind that they calculated only the forcing due to the Greenland fires, which is itself small compared to the North American or Eurasian fires. It should also be said explicitly that the calculated radiative forcing values does not include semi-direct nor indirect effects, which may be dominant here.

–Response: We have rewritten the first part of the section to include the information requested by the referee.

L436 : "cloudless conditions". I do not understand the purpose of this. It is only an ideal simulation, which is not commented in the paper afterwards. What does it bring to the discussion ?

–Response: The IRF for cloudless conditions is compared against IRF including clouds in the subsequent lines. IRF for cloudless conditions was included, as they show the potential maximum effect of the forcing. The results presented show that the clouds reduce both the TOA and BOA IRF.

L440-442 : It is not clear if the given values refer to the total radiative forcing of BC. What are the relative contributions of the direct radiative forcing of BC and of the radiative forcing of BC deposited on snow surfaces ? The authors also give the values without any uncertainty, but a lot of assumptions have been done to retrieve the BC emissions, the BC size distribution, the BC optical properties. Each of those hypothesis would lead to a range of values of IRF.

–Response: The given values refer to the total instantaneous radiative forcing, that is including both the effect of atmospheric BC and BC deposited on the snow. The latter dominates the IRF contributing between 85 to 99 % depending on BC amount. This has been clarified in the manuscript. The composition of the BC from the fire is not known. Hence average BC optical properties were adopted. We have subsequently performed an uncertainty analysis using realistic variations in BC optical properties. This uncertainty analysis is included in the supplementary material and referred to in the manuscript. We have also performed a sensitivity study and estimated the uncertainty of the BC deposition over Greenland due to the use of different size distribution of BC particles (see answer to previous comments).

L 442: "Fig 7c depicts the temporal behaviour..." Does it represent calculations in cloudy conditions ?

–Response: It is the cloudy conditions that are shown. This information has been added both in the text and the figure caption. The temporal behaviour is shown in Fig 7d. This typo has been corrected as well.

L443-444 : I don't see how this information (blue line) can be useful. The location of the pixel where the maximum IRF is found likely varies with time. Besides the analysis of this figure is not done in text. I recommend to remove it.

–Response: We have removed the blue line from the plot. The idea of plotting the TOA max IRF was to show that the single pixel maximum and area averaged RFs peak at different times. However, we agree with the reviewer that this information was perhaps not so useful.

L448-455 : If the authors want to be able to compare their results to global studies, as it is done here, they need to multiply the value of RF by the area of the simulation domain to obtain a forcing value in watts, and then divide it by the surface area of the Earth to obtain an equivalent global radiative effect in mW/m2 that could be compared to results for global studies.

–Response: The cited value from Skeie et al. (2011) is not a global value, but a value representative for the Greenland ice sheet (Fig 17 of Skeie et al., 2011). It is this value we are comparing against. The values from Myhre et al (2013) are included in order to give the reader a global value to compare against. This has been clarified in the text. It is clear that, on a global scale, the obtained RF values are negligible.

L453-455 : What about the impact of North American and Eurasian fires, whose plumes reach Greenland during the studied period ?

–Response: These plumes are not the focus of the present study. Similar plumes have been studied before, so we don't think focusing on these plumes would provide a lot of new information beyond what has been published before. More technically, we only estimate the impact of these plumes using backward calculations, whereas

RF calculations would require forward calculations. We think this is out of scope of the present paper. What we have done, instead, is to calculate the impact of the N. American fires in the surface concentrations of BC over Greenland (see section 3.3). This proved that the BC concentrations from the N. American fires in August 2017 are more than 1 order of magnitude higher compared with those produced from the Greenland fires of August 2017 (see updated Figure 4).

L456-457 : What is the albedo reduction due to BC deposition that can be ascribed to Greenland fires / to fires outside Greenland / to anthropogenic sources ? If the goal of the paper is indeed to focus on the impact of the Greenland fires, quantifying this effect and comparing it to the relative contribution of the different sources would be really valuable for the paper. The authors should also compare their albedo reduction values to previous studies, e.g. Polashenski et al. (2015).

–Response: We now compare our results to those of Polashenski et al. (2015) in section 4.2. The detailed study of the fires outside Greenland and from anthropogenic sources is beyond the scope of this paper. Notice that the albedo effects can't be done on the basis of the backward calculations done for the other sources and would require totally new forward simulations. However, giving the range of surface BC concentrations over Greenland (section 3.3 and Figure 4) is already enough to conclude that the event that we studied in the current paper has minor effects on the albedo or RF compared to BC from the N. American fires or from anthropogenic sources simply because of the different magnitude and duration of these fires.

Sect. 5 : The conclusions may be more quantitative. For example : L478-479 : the ratio of BC deposition from the different sources can be given

–Response: We have not quantified how big the deposition from anthropogenic and biomass burning sources is, and we have removed this sentence from the manuscript.

L481-483 : the AOD enhancement can be precised

–Response: Corrected.

L488 : "albedo change due to the BC deposition". Which sources have been considered ?

–Response: Corrected. It is the albedo change due to BC deposition from the Greenland fire of 2017.

L496-500 : Remove this purely speculative sentence. The opposite could also be said, given the findings of the paper.

–Response: These lines state that "The very large fraction of the BC emissions deposited on the Greenland Ice Sheet (30% of the emissions) makes these fires very efficient climate forcers on a per unit emission basis. If the expected future warming of the Arctic (IPCC, 2013) produces more fires in Greenland in the future (Keegan et al., 2014), this could indeed cause substantial albedo changes and thus contribute to accelerated melting of the Greenland Ice Sheet." We do not understand why this is speculative. A fraction of 30% deposition on the Greenland ice sheet is substantial, much higher than from any other source type and source region, so – on a per unit mass basis – the forcing due to albedo change is efficient, even if it is small overall. The second sentence can perhaps be considered somewhat speculative, but we have now reformulated it and, moreover, we support it with references. Furthermore, it is not presented as a conclusion, so it should be very clear to the reader to what extent this sentence is speculative. We nevertheless consider it important enough to keep it.

The choice of the figures kept in the manuscript is rather strange. Most useful figures relevant for the discussion have been displaced to the Supplementary Material. I recommend to move them to the main paper.

–Response: We have moved the figure with the calculated dosages to the manuscript (Fig. 4), as the dosages are discussed more in the text. We have now placed back to the supplements (Fig. S5) the figure of the footprint emission sensitivities that are

not the main focus of the paper. We are willing to put more figures in the manuscript in a next step, if the reviewer point into this direction. The only reason for using limited number of figures was that this paper was intended to be short.

Fig. 2a: Are those values averaged over the simulation domain ? over Greenland ? I had hard time to figure out how those values could be realistic. I think there is either a issue with the unit or a mistake in the calculation. Shouldn't it be ng/m3 or ng/kg instead of ug/m3 ? The total concentrations of BC in the domain should be calculated as the volume average of the grid cell concentrations, not the sum over all grid cells in the domain...

–Response: We thank the reviewer for this comment that we have now corrected. We now present the average vertical concentrations over Greenland from the 2017 fires in pg/m3 in the updated Figure 2.

Fig 2b : Here again, there is an issue with the unit. The color bar indicates ug/m2 (which is probably right), but the caption says ng/m2. Which one is correct ?

–Response: We also appreciate reviewer's help to correct this mistake. The error was in the legend and it has now been updated.

Fig. 4 : It is extremely difficult to see the colored grid cells an read their values. Please improve the quality of this figure.

–Response: Quality of the figures has been set to 300 dpi. This should solve the problem.

Fig. 5 : Does the altitude represent agl or amsl ? The orography in Greenland is not flat. Response: It is agl altitude and we now clarify it in the legend.

Fig 5 : Why do you keep the contribution of fires burning outside Greenland but exclude the BC contribution of anthropogenic sources ? According to Fig. 4, their contribution is absolutely not negligible and they might modify the time series of column-integrated BC in Greenland.

[Figure]

–Response: We thank the reviewer for this comment. We tried to put also anthropogenic BC in these time-series. Column-ntegrated anthropogenic BC is very low and a stacked line does not show anything in the time-series and that's the reason that we decided not to present it. We have added a small comment in the legend.

Fig. 6 : it would be better to use the same scale for longitude and altitude on panels (b) and (d).

–Response: The reason that we did not use the same scale for longitude and altitude in these two figures is due to the small aerosol structure at high altitudes seen in the CALIOP data. We thought that this is likely due to the N. American fires that were burning at the same time with the Greenland ones. This is visible from the AOD measurements at many of the Greenland stations where large increases in AOD were observed. In a previous comment for the AOD increase in the Narsarsuaq station at the beginning of September, we provided relevant footprint emission sensitivities and biomass burning emissions from CAMS_GFAS (see Figure R2). They explicitly show that the largest footprint was found in Canada in areas with large biomass burning emissions. However, since we do not study the impact of the N. American fires in detail, the sentences about the presence of N. American fire plumes at high altitudes in section 4.1 are rather speculative and we have removed them. We have also corrected Figure 6, as the reviewer suggested and for this, we acknowledge him.

Fig. 7c : Is the snow albedo reduction plotted for 31 August or for the full period ?

–Response: The snow albedo reduction due to BC deposition from the beginning of the fires until 31 August is plotted in Figure 7c (please see last paragraph of section 4). Legend has also been updated.

Table 1 : This table is not commented nor anlyzed in text. We can notice changes in the sources of RS data at different periods, which should be detailed in the methodology section.

–Response: In line 105 we state that we used different RS data to better delineate fire perimeters and define burn severity. Which day each RS tool was used is shown by pointing to Table 1. In addition, discussion of the results presented in Table 1 is presented in section 3.1 and 3.2.

Legrand, M., et al. (2016), Boreal ïnЁĞA ÌĬre records in Northern Hemisphere ice cores: A review, Clim. Past, 12(10), 2033–2059. Polashenski, C. M., J. E. Dibb, M. G. Flanner, J. Y. Chen, Z. R. Courville, A. M. Lai, J. J. Schauer, M. M. Shafer, and M. Bergin (2015), Neither dust nor black carbon causing apparent albedo decline in Greenland's dry snow zone: Implications for MODIS C5 sur-facereïnЁĞC ÌĄectance,Geophys.Res.Lett.,42,9319–9327,doi:10.1002/2015GL065912. Thomas, J. L., et al. (2017), Quantifying black carbon deposition over the Green- land ice sheet from forest fires in Canada, Geophys. Res. Lett., 44, 7965–7974, doi:10.1002/2017GL073701.

Technical comments : L350 : "adopted". Do you mean "adapted" ?

–Response: In this sentence we think that "adopted" fits better. We just used active fires from MODIS; we did not adapt anything.

L394 : Replace "for validating" by "to validate".

–Response: Corrected.

L485 : Replace "attenuation" by "attenuated"

–Response: Corrected.

L512 : Please write "Brent Holben" in two words.

–Response: Corrected.

REFERENCES

Holben, B. N., Eck, T. F., Slutsker, I., Smirnov, A., Sinyuk, A.,Schafer, J.,

Giles, D., and Dubovik O.: AERONET's Version 2.0 quality assurance criteria, http://aeronet.gsfc.nasa.gov/new_web/Documents/AERONETcriteria_final1.pdf, 2006.

Andrews, E., Ogren, J. A., Kinne, S., and Samset, B.: Comparison of AOD, AAOD and column single scattering albedo from AERONET retrievals and in situ profiling measurements, Atmos. Chem. Phys., 17, 6041-6072, https://doi.org/10.5194/acp-17-6041-2017, 2017.

Dubovik, O., Smirnov, A., Holben, B. N., King, M. D., Kaufman, Y. J., Eck, T. F., and Slutsker, I.: Accuracy assessment of aerosol optical properties retrieval from AERONET sun and sky radiance measurements, J. Geophys. Res., 105, 9791–9806, 2000.

Paugam, R., Wooster, M., Atherton, J., Freitas, S. R., Schultz, M. G., and Kaiser, J. W.: Development and optimization of a wildfire plume rise model based on remote sensing data inputs – Part 2, Atmos. Chem. Phys. Discuss., 15, 9815-9895, https://doi.org/10.5194/acpd-15-9815-2015, 2015.

[Figure]

**Fig. 1.** Annual number of active fires over Greenland during the last 17 years as seen from NASA's MODIS satellite (product MSC14DL).

[Figure]

**Fig. 2.** Biomass burning emissions of BC from GFAS (upper panel) in the beginning of September and the footprint emission sensitivity from the Narsarsuaq station (bottom panel)

---

## Author Comment (AC3) · 21 Jun 2018

For the convenience of the reviewing process, we attach the reference that is mentioned in the response to the referee's #1 comments:

Lutes, Duncan C.; Keane, Robert E.; Caratti, John F.; Key, Carl H.; Benson, Nathan C.; Sutherland, Steve; Gangi, Larry J. 2006. FIREMON: Fire effects monitoring and inventory system. Gen. Tech. Rep. RMRS-GTR-164-CD. Fort Collins, CO: U.S. Department of Agriculture, Forest Service, Rocky Mountain Research Station.

Please also note the supplement to this comment:

[Figure]

https://www.atmos-chem-phys-discuss.net/acp-2018-94/acp-2018-94-AC3-supplement.pdf

[Figure]

**Supplement:**

[Figure]

United States
Department
of Agriculture

Forest Service

**Rocky Mountain
Research Station**

General Technical
Report RMRS-GTR-164-CD

June 2006

[Figure]

**FIREMON: Fire Effects Monitoring and Inventory System**

**Technical editor:
Duncan C. Lutes**

[Figure]

**Abstract**

Lutes, Duncan C.; Keane, Robert E.; Caratti, John F.; Key, Carl H.; Benson, Nathan C.; Sutherland, Steve; Gangi, Larry J. 2006. **FIREMON: Fire effects monitoring and inventory system**. Gen. Tech. Rep. RMRS-GTR-164-CD. Fort Collins, CO: U.S. Department of Agriculture, Forest Service, Rocky Mountain Research Station. 1 CD.

Monitoring and inventory to assess the effects of wildland fire is critical for 1) documenting fire effects, 2) assessing ecosystem damage and benefit, 3) evaluating the success or failure of a burn, and 4) appraising the potential for future treatments. However, monitoring fire effects is often difficult because data collection requires abundant funds, resources, and sampling experience. Often, the reason fire monitoring projects are not implemented is because fire management agencies do not have scientifically based, standardized protocols for inventorying pre- and postfire conditions that satisfy their monitoring and management objectives. We have developed a comprehensive system, called the **Fire Effects Monitoring and Inventory System (FIREMON)**, which is designed to satisfy fire management agencies' monitoring and inventory requirements for most ecosystems, fuel types, and geographic areas in the United States. FIREMON consists of standardized sampling methods and manuals, field forms, database, analysis program, and an image analysis guide so that fire managers can 1) design a fire effects monitoring project, 2) collect and store the sampled data, 3) statistically analyze and summarize the data, 4) link the data with satellite imagery, and 5) map the sampled data across the landscape using image processing. FIREMON allows flexible but comprehensive sampling of fire effects so data can be evaluated for significant impacts, shared across agencies, and used to update and refine fire management plans and prescriptions. FIREMON has a flexible structure that allows the modification of sampling methods and local code fields to allow the sampling of locally important fire effects evaluation criteria.

**Keywords**: monitoring, fire effects, inventory, sample design, sampling methods, fuels, burn severity, analysis

**The Authors**

**Duncan C. Lutes**—Systems for Environmental Management, Fire Sciences Laboratory, P.O. Box 8089, Missoula, MT 59807; phone: 406.329.4761; FAX: 406.329.4877; e-mail: dlutes@fs.fed.us.

**Robert E. Keane**—USDA Forest Service, Rocky Mountain Research Station, Fire Sciences Laboratory, P.O. Box 8089, Missoula, MT 59807; phone: 406.329.4846, FAX: 406.329.4877; e-mail: rkeane@fs.fed.us.

**John F. Caratti**—Systems for Environmental Management, P.O. Box 8868, Missoula, MT 59807; phone: 406.549.7478.

**Carl H. Key**—USDI U.S. Geological Survey, Northern Rocky Mountain Science Center, West Glacier, MT 59936; phone: 406.888.7991; FAX: 406.888.7990; e-mail: carl_key@usgs.gov.

**Nathan C. Benson**—USDI National Park Service, Natural Resource Program Center, 1201 Oak Ridge Dr., Suite 200, Fort Collins, CO 80525-5596; phone: 970.267.2121; FAX: 970.225.3585; e-mail: nate_benson@nps.gov.

**Steve Sutherland**—USDA Forest Service, Rocky Mountain Research Station, Fire Sciences Laboratory, P.O. Box 8089, Missoula, MT 59807; phone: 406.329.4813; FAX: 406.329.4877; e-mail: ssutherland@fs.fed.us.

**Larry J. Gangi**—Systems for Environmental Management, Box 8868, Missoula, MT 59807; phone: 406.549.7478.

Rocky Mountain Research Station
Natural Resources Research Center
2150 Centre Avenue, Building A
Fort Collins, CO  80526

**FIREMON**

**Fire Effects Monitoring and Inventory System**

[Figure]

**Integration of Standardized Field Data Collection Techniques and Sampling Design With Remote Sensing to Assess Fire Effects**

This project was funded by the USDA and USDI Joint Fire Sciences Plan.

Agreement number 98-IA-189

Joint Fire Science Program, National Interagency Fire Center,

3833 S. Development Ave., Boise, ID 83705

[Figure]

[Figure]

**Contents**

**FIREMON Introduction**

[Figure]

**THE AUTHORS**

**Duncan C. Lutes** is a Research Forester with Systems for Environmental Management, stationed at the USDA Forest Service, Rocky Mountain Research Station, Fire Sciences Lab, P.O. Box 8089, Missoula, MT 59807; phone: 406.329.4761; FAX: 406.329.4877; e-mail: dlutes@fs.fed.us. He has a B.S. and M.S. degree in forestry from the University of Montana, Missoula. His background is in fuels, principally coarse woody debris, primarily studying spatial and temporal distributions. He has been involved in the development of the First Order Fire Effects Model and several Fire and Fuels Extension variants to the Forest Vegetation Simulator. He is the Technical Editor and contributing author for this publication.

**Robert E. Keane** is a Research Ecologist with the USDA Forest Service, Rocky Mountain Research Station at the Fire Sciences Laboratory, P.O. Box 8089, Missoula, MT 59807; phone: 406.329.4846; FAX: 406.329.4877; e-mail: rkeane@fs.fed.us. Since 1985, Bob has conducted ecological research into fuel dynamics, ecosystem simulation, ecosystem restoration, and spatial modeling for the Fire Effects project. His most recent research includes 1) developing ecological computer models for exploring landscape, fire, and climate dynamics, 2) the mapping of fuel characteristics for spatially explicit fire growth and fire effects evaluation, 3) synthesis of a First Order Fire Effects Model to predict the direct consequences of a fire, 4) exploring the ecology and restoration of whitebark pine in the Northern Rocky Mountains, and 5) classification and simulation of vegetation communities on the landscape using GIS and satellite imagery. He received his B.S. degree in forest engineering from the University of Maine, Orono; his M.S. degree in forest ecology from the University of Montana, Missoula; and his Ph.D. degree in forest ecology from the University of Idaho, Moscow.

**John F. Caratti** is a Systems Ecologist with Systems for Environmental Management, P.O. Box 8868, Missoula, MT 59807; phone: 406.549.7478. Since 1988, John has developed computer software for ecological data analysis and wildlife population modeling. His most recent work includes developing database applications for georeferenced field data, vegetation classification and mapping, and fire effects monitoring. He received his B.A. degree in ecology from the University of California, San Diego, in 1988, and his M.S. degree in wildlife biology from the University of Montana in 1993. John has worked as an ecologist with the USDA Forest Service, Northern Region, a quantitative ecologist with The Nature Conservancy, and as an independent contractor.

**Carl H. Key** is a Geographer with the USDI U.S. Geological Survey at the Northern Rocky Mountain Science Center, Glacier Field Station, West Glacier, MT 59936; phone: 406.888.7991; FAX: 406.888.7990; e-mail: carl_key@usgs.gov. Carl developed the Geographic Information System currently used by Glacier National Park, and continues to support its application and advancement. Since 1976 he has coordinated and conducted several studies in fire ecology, land type classification, exotic plants, biological monitoring, and satellite telemetry. He also contributed to various other wildlife studies, has advanced software development, and technology transfer, and has consulted on management issues for the National Park Service and other agencies. He was instrumental in establishing the global change research program at Glacier in the early 1990s. Since 1996, he has been researching the characteristics of fire severity using remote sensing, and helping to implement burn assessment on a national level.

**Nathan C. Benson** is a Fire Ecologist with the USDI National Park Service at the Natural Resource Program Center, 1201 Oak Ridge Drive, Suite 200, Fort Collins, CO 80525-5596; phone: 970.267.2121; FAX: 970.225.3585; e-mail: nate_benson@nps.gov. Nate has worked for the National Park Service for more than 15 years in a variety of positions. He started his NPS fire career as a fire effects monitor at Glacier National Park, and then moved to Yellowstone and Great Smoky Mountains National Parks as a Fire Use Module Leader. Most recently he was the Prescribed Fire Specialist at Everglades National Park and is currently working for the NPS Fire Management Program Center as a Fire Ecologist. Since 1996, he has worked with Carl Key on researching characteristics of fire severity using remote sensing and has been the lead for the National Park Service in developing the National Burn Severity Mapping Project, a cooperative project between the National Park Service and U.S. Geological Survey. Nate has an M.S. degree in land resources from the University of Wisconsin-Madison's Institute for Environmental Studies.

**Steve Sutherland** is a Research Ecologist with the Fire Effects Project at the USDA Forest Service, Rocky Mountain Research Station, Fire Sciences Laboratory, P.O. Box 8089, Missoula, MT 59807; phone: 406.329.4813; FAX: 406.329.4877; e-mail: ssutherland@fs.fed.us. Since joining the lab, he has been involved with the Southern Utah Fuels Management Demonstration Project, expanding weed summaries for the Fire Effects Information System, and researching Postfire Weed Response in western Montana. Before joining the Fire Lab, Steve was the State Ecologist for the Ohio Chapter of The Nature Conservancy where he oversaw the plant community and rare species monitoring program. Prior to that, he was an Assistant Research Professor in the Department of Biology at the University of Utah. He received B.S. and M.S. degrees in forestry from Utah State University, a Ph.D. degree in evolutionary ecology from the University of Arizona, and a postdoctoral fellowship in population genetics from the University of Utah.

**Larry H. Gangi** is a Programmer with Systems for Environmental Management, P.O. 8868, Missoula, MT 59807; phone: 406.549.7478. Larry has developed software for a variety of natural resource applications including fire modeling, ecological survey data analysis and conversions, timber stand, and water resources. He received his B.S. and M.S. degrees in computer science from the University of Montana, Missoula.

**ACKNOWLEDGMENTS**

Funding for FIREMON was provided by the Joint Fire Science Program. Additional support was provided by the USDA Forest Service, Rocky Mountain Research Station; USDI United States Geological Survey, and Systems for Environmental Management. USDA Forest Service Statisticians Rudy King and Dave Turner provided critical input regarding the Integrated Sampling Strategy (ISS) and the analysis tools software. Courtney Couch worked diligently to take the authors' stick-figure pictures and create many of the illustrations for FIREMON. Further valuable support came from Tim Sexton, Dick Bahr, Brian Sorbel, and Tom Zimmerman. Many at the USGS EROS Data Center, in particular, Don Ohlen, Zhi Liang Zhu and Stephen Howard, have been instrumental in the national implementation of the burn severity mapping for the NPS. Axiom IT solutions, especially Jeffrey Heng, Douglas Wissenbach, and Marc Dousset, used their expertise to build a robust data structure, data entry system, and enhanced documentation. Virginia Arensberg searched numerous publications to build a thorough glossary of terms. Finally, we thank the fire monitors, fire ecologists, fire GIS specialists, and others, from many agencies, who have enthusiastically supported development of FIREMON.

**EXECUTIVE SUMMARY**

Monitoring and inventory to assess the effects of wildland fire is critical for 1) documenting fire effects, 2) assessing ecosystem damage and benefit, 3) evaluating the success or failure of a burn, and 4) appraising the potential for future treatments. However, monitoring fire effects is often difficult because data collection requires abundant funds, resources, and sampling experience. Often, the reason fire

monitoring projects are not implemented is because fire management agencies do not have scientifically based, standardized protocols for inventorying pre- and postfire conditions that satisfy their monitoring and management objectives. We have developed a comprehensive system, called the *Fire Effects Monitoring and Inventory System (FIREMON)*, which is designed to satisfy fire management agencies' monitoring and inventory requirements for most ecosystems, fuel types, and geographic areas in the United States. FIREMON consists of standardized sampling methods and manuals, field forms, database, analysis program, and an image analysis guide so that fire managers can 1) design a fire effects monitoring project, 2) collect and store the sampled data, 3) statistically analyze and summarize the data, 4) link the data with satellite imagery, and 5) map the sampled data across the landscape using image processing. FIREMON allows flexible but comprehensive sampling of fire effects so data can be evaluated for significant impacts, shared across agencies, and used to update and refine fire management plans and prescriptions.

The key to successful implementation of FIREMON requires the fire manager to succinctly state the objectives of the proposed fire monitoring project and accurately determine the available monitoring or inventory project resources. Using this information, the manager uses a series of FIREMON keys to decide the sampling strategy, methods, and intensity needed to accomplish the objectives with the resources on hand. Next, the necessary sampling equipment is gathered and dispersed to sampling crews. Field crews then collect FIREMON data using the detailed methods described in this FIREMON documentation. Collected data are then entered into a Microsoft® Access database. These data can be summarized, analyzed, and evaluated using the set of integrated programs developed specifically for FIREMON.

FIREMON has a flexible structure that allows the modification of sampling methods and local code fields to allow the sampling of locally important fire effects evaluation criteria.

**INTRODUCTION**

We have developed a comprehensive *Fire Effects Monitoring and Inventory System*, called *FIREMON*, that integrates new and current ecological field sampling methods with remote sensing of satellite imagery to assess the effects of fire on important ecosystem components. The primary objective of FIREMON is to measure the immediate and long-term effects of a planned or unplanned fire on critical ecosystem characteristics so that fire managers can evaluate the impact of that fire on ecosystem health and integrity. This information can be used to refine fire management plans and prescriptions. This system is NOT used to document the behavior of the fire, but rather it is used to record the consequences of the fire on the landscape.

We used the National Park Service Fire Monitoring Handbook (FMH) (USDI NPS 2001) and the ECODATA Handbook (Hann and others 1988) as the framework for designing FIREMON sampling methods. However, we extended the utility of these protocols by providing nested levels of sampling intensity coupled with sampling flexibility. We designed FIREMON so that most of the data collected with FIREMON procedures will be compatible with other monitoring and inventory systems such as FMH and Natural Resource Information System (NRIS) databases. Additional sampling methods can be easily added to FIREMON as fire managers recognize their relevance in regard to inventorying and monitoring fire effects. A method to monitor water quality, for instance, would be a useful addition to the group of FIREMON sampling protocols.

*Monitoring* is the critical feedback loop that allows fire management to constantly improve prescriptions and fire plans based on the new knowledge gained from field measurements. *Inventory* is the description and quantification of important ecosystem and landscape elements and is critical to fire management activities for planning, prioritizing, and designing prescribed fire activities.

Monitoring the effects of wildland fire is critical for 1) documenting extent of fire effects, 2) assessing ecosystem damage and benefit, 3) evaluating the success or failure of a prescribed burn, 4) appraising the potential for future treatments, and 5) prioritizing stands for fire treatment. Objectives for

monitoring depend on the type of fire. Wildfire monitoring is necessary to evaluate the possible need for rehabilitation or to assess a fire's potential impact to the ecosystem, while monitoring prescribed fires is invaluable for assessing the efficacy of the treatment. Monitoring data can have far-reaching applications in fire management because they provide the scientific basis for planning and implementing future burn treatments. Moreover, this information documents important fire effects, which can be used by other districts, agencies, and countries for their projects. Measuring postfire ecosystem response also allows us to understand the consequences of fire on important ecosystem components and share this knowledge in a scientifically based language.

Despite its importance, it is often a challenge for fire managers to install effective monitoring programs due to resource limitations inherent in time, money, people, and expertise. Also, often fire managers find themselves too busy with other essential duties to design and implement monitoring projects. And the perceived complexity of monitoring sampling designs has often overwhelmed or intimidated some fire managers. The issue of complexity is especially true when the fires to be monitored are large (greater than 1,000 acres), occur on diverse landscapes, and have complex severity patterns. Moreover, it is difficult to design a cost-efficient sampling strategy that will quantify stand- and landscape-level fire effects across an entire landscape using scientifically credible methods. But perhaps the main reason most fire monitoring projects never become implemented is the lack of standardized and comprehensive sampling methods and tools easily available to fire managers. Most fire management agencies do not have the scientifically based sampling protocols for inventorying pre- and postfire conditions to satisfy monitoring objectives. (The USDA Forest Service Monitoring and Evaluation Working Paper dedicates only one paragraph to data collection methods.) The major exception is the USDI National Park Service, which has extensive guidelines and protocols for sampling ecosystem characteristics that are important to monitoring fire effects (National Park Service 2001, http://www.nps.gov/fire/fire/fir_eco_firemonitoring.html). Collecting field data is easily the most expensive part of any monitoring and evaluation project, requiring extensive expertise in field sampling, fire and landscape ecology, and sampling methods design. Perhaps the single greatest challenge of designing a fire monitoring project is matching existing funding, personnel, and equipment with monitoring objectives to achieve scientifically credible evaluation data.

Monitoring is an extremely complex task that requires an extensive assessment of many ecosystem characteristics across multiple time and space scales. Fire effects monitoring, in this approach, does not include documentation of the behavioral characteristics of the fire, but rather the sampling of the ecosystem characteristics that are directly affected by the fire. These fire effects can be described at the plant level (mortality), at the stand level (fuel composition, species composition), and at the landscape level (patch dynamics, burn severity mosaic). Moreover, fire effects can be described over many timeframes including immediate (directly after fire), short (1 to 5 years postfire), or long (10 to 100 years postfire) term measurements. A valid sampling strategy for monitoring fire effects must provide for the integration and linkage of ecosystem response across these multiple time and space scales to provide meaningful data to fire management. Our intent in developing FIREMON is not to replace current systems of fire severity assessment, but rather to augment these efforts with a comprehensive and flexible set of recognized field and office methods.

It would be impossible, and probably inefficient, to design a fire monitoring program to include the measurement of all possible information a fire manager in any part of the United States would want to monitor. For instance, fire managers in the Western United States may not need a measurement, such as depth to water table, but this measurement might be absolutely critical to Eastern United States managers. Therefore, we have included local code fields in FIREMON that allow the manager to include other measurements that describe the macroplot. For example, hiding cover (horizontally projected plant cover) may be an important criterion in setting the objectives for a prescribed burn, so the manager could develop a coding system and use one of the FIREMON local code fields to assess hiding cover.

As managers attempt to oversee broader and broader areas for fire, fire effects information is increasingly difficult to obtain. Direct observation may be largely impeded by fire size, remoteness, and

rugged terrain, and there may be little chance for sufficient reconnaissance on the ground. In some cases, the sheer number of areas to evaluate in one fire season is overwhelming. In others, managers with regional responsibilities may need to aggregate information from many districts to report their burn results, or to develop integrated plans. For circumstances such as these, FIREMON offers a section on Landscape Assessment (LA), which primarily addresses the need to identify and quantify fire effects over large areas, involving potentially many burns and covering tens of thousands of acres at a time. It incorporates remote sensing and GIS technologies that can produce a variety of derived products such as maps, images, and statistical summaries. The ability to compare results is emphasized, along with capacity to aggregate information across broad regions over time.

Landscape Assessment shows the spatial heterogeneity of burns, and how fire interacts with vegetation and topography, providing a *quantitative* picture of the whole burn as if viewed from the air. The quantity measured and mapped is "burn severity," defined here as a scaled index gauging the magnitude of ecological change caused by fire. In the process, two methodologies are integrated. One, the Normalized Burn Ratio (NBR), involves remote sensing using Landsat 30-m data and a derived radiometric value. The NBR is temporally differenced between pre- and postfire datasets to spatially determine the degree of change detected from burning. The other methodology, the Composite Burn Index (CBI), adds a complimentary field sampling approach. It entails a relatively large plot with independent severity ratings for individual strata within the community and a synoptic rating for the whole plot area. Plot sampling may be used to calibrate and validate remote sensing results, or it may be implemented as a stand-alone field survey for individual site assessment

**GENERAL DESCRIPTION**

**What FIREMON Is…**

FIREMON consists of a standardized set of sampling manuals, databases, field forms, analysis programs, and image analysis tools that will allow the manager to design and implement a fire effects monitoring project. To use FIREMON, a fire manager must first succinctly state the objectives of the proposed fire monitoring project. Then the manager must decide the amount of resources available to successfully conduct the project. Using this information, the manager goes to a series of FIREMON keys to decide which methods to use to accomplish the objectives, and the sampling strategy to employ to implement these methods across the landscape. Results from these keys are then used to design the fire monitoring project using FIREMON guidelines and procedures. Sampling equipment and plot forms are gathered and dispersed to sampling crews. The field crews then collect FIREMON data using the detailed methods described in this FIREMON publication. Collected data are then entered into a standardized database using Microsoft® Access software. These data are then summarized, analyzed, and evaluated using the set of FIREMON programs provided by this publication.

FIREMON is designed to be robust by being flexible. It allows fire managers to design a sampling strategy where only those ecosystem measurements of the greatest concern are measured. But FIREMON will still provide a myriad of comprehensive and detailed sampling schemes to measure the many important fire-related ecosystem elements. Sampling design focuses on wildland fire use objectives, rather than a shotgun approach where all ecosystem characteristics are measured to quantify ecosystem change. FIREMON is designed to be applicable for most land areas or ecosystems in the United States.

FIREMON is structured so that it can be easily learned. First, FIREMON resides on an Internet Web site so that it will be easily accessible to all. Second, the entire FIREMON system, including sampling methods, field forms, and databases, are available on CD so that it can be accessed from any computer with Microsoft Word and Access installed (versions 2000 and later). Finally, training courses have been developed to teach FIREMON to fire personnel with limited sampling experience.

**What FIREMON Is NOT…**

To fully understand FIREMON, it is important to emphasize what FIREMON is NOT:

FIREMON is NOT intended to be a corporate database, although it surely could be at some point in the future.

FIREMON is NOT a replacement for FMH in the National Park Service or the NRIS protocols developed by the Forest Service. FIREMON can complement these systems and provide additional help with monitoring tasks.

FIREMON does NOT contain software for extensive data analysis. FIREMON software will provide a general report and statistical summary, but not extensive statistical analyses. More extensive analysis can be accomplished by exporting the data from FIREMON and using them in a statistical package. Also, additional statistical analysis can be added at a later date.

FIREMON is NOT used to document fire behavior; it is used to record the consequences of the fire on the landscape.

FIREMON is NOT just a fire monitoring package. Many procedures and the database within FIREMON are useful for other ecosystem inventory and monitoring. One inventory need we especially included in FIREMON is fuels. FIREMON contains the necessary components for sampling surface fuels for inventory, fuels mapping reference (ground-truth), and fuels summary for input to fire behavior and effects programs.

FIREMON does NOT include sampling methods for all important fire effects. For example, changes in water quality may be an important fire effects issue, but there is no water quality sampling protocol in FIREMON. The sampling methods in FIREMON were written using existing, recognized sampling methods. We were unable to find a standardized protocol for water quality sampling, so we did not include one. However, new sampling methods can be readily added into FIREMON in the future.

**The Four FIREMON Components**

There are four major components to FIREMON:

1) **Integrated Sampling Strategy**—This is a set of step-by-step procedures for designing fire effects sampling projects. This component is composed of design keys, strategy descriptions, and guidelines for designing a successful fire monitoring project.

2) **Field Methods**—These are methods for sampling important ecosystem characteristics used to assess fire effects. There are currently 10 methods implemented into FIREMON: Plot Description (PD), Tree Data (TD), Fuel Load (FL), Species Cover (SC), Cover/Frequency (CF), Line Intercept (LI), Density (DE), Point Intercept (PO), Rare Species (RS), and Composite Burn Index (BI). These sampling methods provide a complete set of field sampling protocols to quantify changes in ecosystem characteristics due to fire to describe stand-level fire effects. Additionally, there are two database tables to record metadata (MD) information and fire behavior (FB).

   The Landscape Assessment component details how remotely sensed imagery can be used to design a spatially explicit strategy to locate, collect, and summarize field data across a burned landscape. These methods require extensive expertise in the processing of remotely sensed imagery.

3) **FIREMON Database**—Field data are stored in the Microsoft® Access-based FIREMON database. Data entry forms look like field forms, and drop down lists limit data entry errors.

4) **Analysis Tools**—These include queries in the FIREMON database for producing plot-level data summaries, and the FIREMON Analysis Tools (FMAT) software for analyzing collected field data. The FMAT program provides data summaries for either plot-level or grouped plots and statistical inference of grouped plots using Dunnett's procedure for multiple comparisons with a control. This test is designed to statistically compare pre- and posttreatment data.

The fire manager can choose to perform all or part of one or more components, but the real power of FIREMON is in the integration of all components to describe fire effects at multiple scales.

**Integrated Sampling Strategy**

The Integrated Sampling Strategy (ISS) component provides the manager with step-by-step instructions on how to design a comprehensive, statistically valid field sampling effort for the purpose of quantifying fire effects over long periods across burned landscapes. This component describes how the detailed sampling procedures are selected, and how to place sample plots across project area. This will allow the fire manager to design a sampling procedure to implement on preburn or postburn areas for describing the effect of the wildfire or prescribed fire.

As in any project, there are three ways to get things done: good, fast, and cheap. But a fact of nature says we cannot accomplish these three goals simultaneously; one can only effectively manage for one and compromise on the remaining two. Therefore, the ISS has a three-level, hierarchically nested strategy for implementing each sampling method in the field assessment. This three-level strategy is geared toward a number of important sampling considerations that attempt to provide a compromise between good, fast, and cheap:

1. **Level I—Simple sampling scheme**. Fastest and cheapest while still collecting useful data in the context of the management objective. Use this scheme if little time, money, or personnel are available to complete the monitoring tasks.

2. **Level II—Recommended sampling scheme**. Somewhat fast, somewhat cheap, and somewhat good. Statistically valid data collected as efficiently as possible but with high levels of variability. Use this scheme if defensible numbers are needed from the monitoring effort, but there is limited time and/or resources.

3. **Level III—Detailed sampling scheme**. Real good but slow and somewhat costly. Statistically valid data with minimized levels of variation but with high collection costs. Use this scheme if the most statistically valid estimates are needed and time and money are not limiting.

These three sampling levels can be used at two spatial levels. The fire manager must pick the sampling level to assign to monitor the landscape and the sampling level to monitor the stands. For example, the land manager may not care about fire effects across the landscape, such as in a prescribed burn, but cares more about stand level changes across the burn unit. In this case, the fire manager would decide on Landscape Level I with Stand Level III. However, another fire manager may not care how a wildfire burned at the stand level, but wants to know general characteristics of how the fire burned across the landscape. In this case, Landscape Level II or III would be selected while Stand Level I or II might be selected, depending on time and resources.

**Field Assessment**

The field assessment portion of FIREMON contains an extensive set of procedures for sampling important ecosystem characteristics before and after a prescribed or natural fire for ecosystems in the United States, including forests, grasslands, and shrublands. The design of FIREMON is such that the fire manager can tailor the field measurement procedures to match burn objectives or wildland fire use concerns. Moreover, the fire manager can scale the intensity of measurement to match resource and funding constraints. For example, to document tree mortality, the fire manager might choose one of three hierarchically nested sampling procedures, where the first procedure might provide general descriptions of tree mortality quickly at low cost (photopoints, walk-through), while the third procedure would document, in detail, individual tree health and vigor, to generate comprehensive data applicable to many analyses but costly to collect. A key has been developed to help fire managers decide the appropriate methods and sampling intensity for each.

The field assessment procedures are written into a handbook that can be taken into the field. The assessment is composed of 1) field methods, 2) plot forms, 3) cheat sheets, and 4) equipment lists. This assessment does not include details on how certain sampling procedures are selected; those details are in the ISS section.

FIREMON contains the following sampling procedures for monitoring ecosystem characteristics:

**Plot Description (PD)**—A generalized sampling scheme used to describe site characteristics on the FIREMON macroplot with biophysically based measurements.

**Tree Data (TD)**—Trees and large shrubs are sampled on a fixed-area plot. Trees and shrubs less than 4.5 ft tall are counted on a subplot. Live and dead trees greater than 4.5 ft tall are measured on a larger plot.

**Fuel Load (FL)**—The planar intercept (or line transect) technique is used to sample dead and down woody debris in the 1-hour, 10-hour, 100-hour, and 1,000-hour and greater size classes. Litter and duff depths are measured at two points along the along the base of each sampling plane. Cover and height of live and dead, woody and nonwoody vegetation is estimated at two points along each sampling plane.

**Species Composition (SC)**—Used for making ocular estimates of vertically projected canopy cover for all or a subset of vascular and nonvascular species by diameter at breast height (DBH) and height classes using a wide variety of sampling frames and intensities. This procedure is more appropriate for inventory than monitoring.

**Cover/Frequency (CF)**—A microplot sampling scheme to estimate vertically projected canopy cover and nested rooted frequency for all or a subset of vascular and nonvascular species.

**Point Intercept (PO)**—A microplot sampling scheme to estimate vertically projected canopy cover for all or a subset of vascular and nonvascular species. Allows more precise estimation of cover than the CF methods because it removes sampler error.

**Density (DE)**—Primarily used when the fire manager wants to monitor changes in plant species numbers. This method is best suited for grasses, forbs, shrubs, and small trees that are easily separated into individual plants or counting units, such as stems. For trees and shrubs over 6 ft tall the TD method may be more appropriate.

**Line Intercept (LI)**—Primarily used when the fire manager wants to monitor changes in plant species cover and height of plant species with solid crowns or large basal areas where the plants are about 3 ft tall or taller.

**Rare Species (RS)**—Used specifically for monitoring rare plants such as threatened and endangered species.

**Landscape Assessment (LA)**—Useful for mapping fire severity over large areas. Combines a ground-based burn severity assessment, the **Composite Burn Index (BI)** and a satellite derived remote sensing analysis method, the **Normalized Burn Ratio (BR)**. The LA methodology will assist in determining landscape level management actions where fire severity is a determining factor. See below for more information.

Each sampling method is discussed in detail in their respective sections. Additional sampling methods can be easily added to FIREMON as fire managers recognize their relevance.

**Landscape Assessment**

The remote sensing of severity is captured by a new Landsat TM radiometric index we call the Normalized Burn Ratio, or NBR. The NBR evolved through sampling of TM band reflectance over burned surfaces, and was tested against three other TM measures appearing in the literature. Multitemporal differencing was employed to enhance contrast and detection of changes from before to after fire. Seasonal effects also were tested to determine the best time of year for TM data acquisition. Based on statistical and visual characteristics, NBR difference from early growing season dates was judged to be optimal, compared to other measures. Results clearly showed the extent of burning that represented a wide range of severity magnitude that was easily interpreted for each burn. Further, the full range of differenced NBR can be stratified into a finite number of ordinal severity levels, to facilitate summation of burns through mapping and tabular statistics. Those data provide a basis for monitoring burn impacts over large regions, and for comparing burns spatially and temporally.

Sensor characteristics make this approach suitable for moderate resolution (30-m) applications that require more extensive and precise information than rapid assessment techniques, and can be completed within a 1-year timeframe of the subject fire.

**FIREMON DOCUMENTATION STRUCTURE**

FIREMON is presented using a series of sections to document the entire fire effects monitoring system. This set of documents is not necessarily designed to be read from front to back like a book, but rather it is designed for FIREMON users to read only those sections that are important to their sampling requirements. Every FIREMON user should read the Integrated Sampling Strategy (ISS) because it contains absolutely essential FIREMON sampling concepts and terminology that are used throughout all documents.

There is an obvious lack of citations in the bulk of FIREMON documentation. This was done on purpose to reduce clutter and improve readability. This does not mean that we didn't consult numerous sampling and monitoring texts during the development of FIREMON. The References sections contain citations for the journal articles, textbooks, reference books, and symposium proceedings used designing and developing FIREMON.

FIREMON also includes a glossary that defines common FIREMON terminology, and a How To… section that describes sampling techniques used in more than one of the FIREMON sampling methods.

We attempted to design FIREMON document structure so that major and minor headings describe critical monitoring tasks. This way, the FIREMON user can easily jump to a particular method or procedure instead of having to read the entire document. For this to work, each heading section must effectively stand on its own so the user does not have to read other sections to understand the section of interest. A side effect of this independent section treatment is that there is often redundant text across sections that may be annoying to those reading each section sequentially. We apologize for this repetition and hope you will recognize its purpose.

[Figure]

**Integrated Sampling Strategy (ISS) Guide**

[Figure]

**Robert E. Keane**
**Duncan C. Lutes**

**SUMMARY**

What is an Integrated Sampling Strategy? Simply put, it is the strategy that guides how plots are put on the landscape. FIREMON's Integrated Sampling Strategy assists fire managers as they design their fire monitoring project by answering questions such as:

- What statistical approach is appropriate for my sample design?
- How many plots can I afford?
- How many plots do I need?
- Where should I put my plots?
- What sampling methods should I use on my plots?

The Integrated Sampling Strategy (ISS) is used to design fire monitoring sampling projects by selecting the most appropriate sampling approach and the most efficient sampling strategy, then choosing the best sampling methods for a fire monitoring project. The first section of the ISS Guide introduces the FIREMON user to the terminology and inherent properties of sampling design in the FIREMON monitoring approach. The second section presents the preliminary information that must be collected or compiled for designing a monitoring project. The third section documents how a monitoring project is implemented. And the last section provides users with guides and keys to assist in developing the monitoring project. New users, especially those responsible for the design of monitoring programs, should read the third section in detail in order to gain the knowledge and understanding needed to implement an appropriate and successful FIREMON monitoring projects.

The ISS in FIREMON is critical to fire monitoring for several reasons. First, many fire managers do not have the background in ecosystem inventorying and sampling to design a statistically credible and efficient sampling strategy. Second, fire managers rarely have the time to learn sampling theory and concepts. Last, integrated sampling requires extensive experience in statistical sampling design and field implementation. FIREMON condenses this detailed information on sampling strategy into the ISS to guide the fire manager in planning and implementing an appropriate fire monitoring project.

**INTRODUCTION**

The FIREMON Integrated Sampling Strategy (ISS) uses the best estimate of resources that the manager can provide to help design the plot level and landscape level sampling strategy of a fire effects monitoring project. A sampling strategy is different from a sampling method in that a sampling strategy

describes where, when, and how the sampling methods (procedures for measuring things) are implemented across the landscape. This section allows the fire manager to match the appropriate sampling strategies with the scope and context of the project objectives.

The quickest way to design a fire effects monitoring project is to complete the set of sampling strategy and method keys provided in the FIREMON ISS. These keys provide guidance in the selection of various criteria needed to design a statistically credible and defensible monitoring sampling strategy. FIREMON provides methods for measuring fire effects at most levels of intensity and most any scale, and then provides guidance for data analysis that is appropriate for the data that have been collected. For example, a coarse sampling design that specifies pictures as the only data collected cannot be used to determine tree mortality, fuel consumption, or any other fire effect. Likewise, broad visual estimates of plant species canopy cover for a large area cannot be used to describe changes in plant composition.

Implementation of a FIREMON monitoring program is based on two components: objective(s) and sampling resources. The sampling objective or objectives provide the fundamental criteria for determining the sampling methods and, to a lesser extent, the sampling intensity that will be integrated into a FIREMON monitoring program. It is critical that the fire manager succinctly articulate the actual purpose of the sampling effort in the FIREMON sampling strategy and design process. Without an expression of the sampling purpose, the fire monitoring project is doomed to fail. The fire manager must explicitly state the reasons why a fire effects monitoring project is needed. These reasons provide the critical context to form the project objectives, which in turn drive the sampling methods. Sampling resources are less easily assessed as they are related to funds, time, personnel, and equipment, all of which can be somewhat dynamic throughout the course of the field season.

**Advanced Alternative to the FIREMON Integrated Sampling Strategy**

The FIREMON ISS provides general guides and keys for you to use to determine the sampling strategy that best fits with the objectives and resources available for monitoring fire effects. Recently, new technology has been developed by Spatial Dynamics in cooperation with the USDI National Park Service Fire Monitoring Program that is an advanced alternative to the ISS presented in FIREMON. This new software is called FEAT or Fire Ecology Assessment Tool and it allows the user to interactively design sampling strategies with Geographic Information Systems (GIS) and integrated databases, and then implement the strategy on the landscape using GIS techniques and plot-level databases similar to FIREMON. FEAT is a complete fire monitoring software package that integrates the entire monitoring effort into one system. Users can use the FIREMON plot methods or they can use the FEAT plot methods for collecting data.

The system allows the user to examine a range of monitoring design applications and alternatives, such as:

- Random location of plots within an area using GIS techniques
- Identification of the sampling area and strata using any number of GIS layers
- Plot sampling methods linked to relational databases inside the GIS structure that allows plots to be shown on a GIS map and sampled attributes of the plot to be spatial displayed.
- Digital photo integration with plot and a GIS to allow point-and-click real-time information for each plot or sampling strata using photos or data.
- Ability to easily define new sampling protocols and modify existing protocols.
- Ability to manage tabular data using GIS.
- Designed to support efficient data entry into Personal Data Assistants (PDA).

FEAT is a comprehensive system that combines a number of software platforms to form an integrated fire effects monitoring package. The all-inclusive nature is the benefit of FEAT; however, some monitoring programs may find it difficult to meet the associate resource needs. For instance, there is

a substantial initial financial commitment and ongoing maintenance overhead for the software needed to run FEAT (ARCMap/Spatial Analyst, Microsoft XP). To use the full capability and understand the underlying analysis within FEAT, specialized training is required. Also, there is a workload associated with updating and maintaining the GIS layers that FEAT requires. FEAT was developed to facilitate flexibility in sampling procedures and methods, so field methods can be extremely adaptable if required by your monitoring project. This is especially true if you want to modify a sampling procedure to measure a new entity or ecosystem characteristic; FEAT will allow one to easily modify or develop a new sampling protocol.

FIREMON users are encouraged to consider using FEAT for their monitoring system if they feel comfortable using the advanced features offered in FEAT, and have the financial commitment to obtain and keep the resources necessary to effectively apply the system. More information about FEAT can be obtained from the National Park Service, Fire Monitoring Program Web site: http://www.nps.gov/fire/fire/fir_ecology.html.

**Resampling Existing FIREMON Plots**

If you are revisiting plots that have already been sampled then you do not necessarily need to read through the ISS at this time. Instead, carefully read through the FIREMON metadata (MD) information and/or FIREMON notebook to determine the methods and sampling intensity that were incorporated during the first sampling visit, and identify any optional fields or data variables that were developed at that time. Return to the FIREMON plots and sample using the same methods, intensity, and so forth, used during the original sampling. When reading the MD information you may also note any shortcomings identified by the previous sampling visit and modify the methods to make the sampling more effective. Use care when doing this so that the initial measurements can be used for analysis. For instance, changing the vegetation sampling method from cover to frequency would mean that the cover values could not be used in the analysis. Instead, the frequency method should be *added* to the list of methods applied at the plot and not used to replace the cover method.

Many studies examine change in vegetation attributes after a treatment. Generally, these attributes are related to the change in species numbers, the number of individual plants or vegetation cover as a result of the treatment. For instance, a manager might be interested in noting the difference in density of undesirable weed species after a prescribed fire. Or, a manager may want to study the difference in that same weed species in areas burned in the spring versus the fall in an attempt to identify an effective way to control its numbers. Whatever your reason for sampling it is important to recognize how plant attributes change during the season and take them into account with your sampling. Generally, this will mean sampling at the same time or times every year. It would be difficult to observe the effectiveness of treatment if, say, the first season the vegetation was sampled in late summer and the next year in early spring because plant growth during the year would influence plant attributes such as cover, density, and height. There is no hard and fast rule for timing the vegetation sampling, however, so it is up to the fire manager to determine the annual sampling schedule. The schedule will probably be set by date but could be set by some other attribute, such as the phenological stage of some species of interest. Recognize that rigid sampling schedules may make it difficult to finish all the sampling tasks each year. For example, if you decide that late season sampling is the most appropriate time for estimating species cover, some years you may not be able to sample because extreme fire danger keeps the monitoring crews out of the field or an early snowfall may make it impossible to sample fine and coarse woody debris.

**Terminology**

There are a number of terms used in the FIREMON documentation that are either unique to FIREMON or imply a meaning that is specific to the FIREMON documentation. In general, these terms are used as shortcuts to reduce text and focus discussion. The more important FIREMON terms are stratified by subject area and put in context below. Complete definitions are located in the **Glossary**.

**The Project**—A *fire monitoring project* is a fire management activity used to evaluate the effects of a fire using field sampling and statistical analysis. A fire monitoring project that installs field plots AFTER a planned or unplanned fire (or other treatment or disturbance) is called a *postevent monitoring project*, whereas a project that establishes plots BEFORE and AFTER the burn is called a *complete fire monitoring project*. We recognize that disturbance occurs at many intensities and scales, so conceivably every monitoring project is both a postevent and complete monitoring project. However, we make the distinction based on the disturbance event that initiates the sampling program. *Sampling resources* are those assets available to the fire manager to accomplish the monitoring project, most frequently, funds, time, personnel, and equipment.

**The People**—A *FIREMON team* is the group of people involved in the planning and implementation of a fire monitoring project. This team is usually composed of a *FIREMON Project Leader* who oversees the project; a *FIREMON Architect* who plans and designs the appropriate fire monitoring methods and sampling strategies; and *field crews* who implement FIREMON methods in the field. The field crew is composed of the *crew leader* responsible for all logistics in the field and crew training; a *data recorder* who fills out the FIREMON plot sheets and *sampler* that does the actual collection of field data. There can be more than one sampler, and the crew leader and data recorder can also perform sampling duties. Be sure to let different members of the field crew try their hand at different tasks. In other words, if a person is a sampler on one plot let him or her switch jobs with the data recorder on the next. This will keep the field work from getting too monotonous and will let everyone become familiar with a number of field sampling procedures.

There may not always be a large number of people involved in a FIREMON project. For instance, in a small FIREMON sampling project, one person can be the FIREMON project leader, architect, crew leader, and data recorder. There should always be at least two people on the field crew, for safety sake. In the interest of good quality data it is useful to have one field crew member that has the expertise to overlook the sampler's observations, checking both the accuracy and precision of the recorded data. For instance, cover estimation can be quite difficult, especially for someone who is just starting out. It is important to have someone on the crew who is able to accurately estimate cover and to have that person check the cover estimations made by other crew members.

**The Sampling Procedure**—*Sampling strategies* are how, where, and why sampling methods are implemented on the landscape. *Sampling methods* are a set of procedures for measuring specific ecosystem attributes. The difference between strategies and methods can be somewhat vague. Think of measuring a tree's diameter with a diameter tape—that is a sampling method; then think of measuring tree diameters on all trees above 4.5 ft on a 0.25 acre circular plot randomly across a landscape—that is a sampling strategy. Finally, the *sampling approach* is the scheme used to drive the sampling strategy design process. Simply put, there are two sampling approaches in FIREMON, statistical and relevé. Each is discussed later in the ISS.

**The Sampling Unit**—The FIREMON *macroplot* defines the greater sampling area in which all of the sampling methods are nested. The size and shape of the macroplot is determined by sampling objectives and resources, but most macroplots will be rectangular or circular encompassing about 0.1 to 0.25 acres (0.04 to 0.1 ha).

Depending on the methods used, the FIREMON plot may be divided into *microplots*, also known as *quadrats*, *belts,* or *subplots*. Each one is a much smaller area used for measuring small-scale phenomena, such as ground cover or individual plant or species attributes. Microplots are usually located in a grid pattern within the macroplot. The size of the microplot depends on the size of the plant or species being measured, but typically it is about 3 ft square (1 m$^2$). Some studies have found that certain types of vegetation are more effectively measured using belt transects. These belts are essentially elongated quadrats. In FIREMON we only associate subplots with the Tree Data (TD) methods where saplings and seedlings are sampled on a smaller plot—the microplot—nested within the larger plot used for sampling mature trees.

A *transect* is a one-dimensional line that is located within the macroplot. Ecological attributes that intersect or cross the transect are tallied or measured.

The vegetation sampling methods, in particular, use a macroplot to define the potential sampling area, with microplots located within, where data are actually collected. Microplot sampling allows macroplot scale attribute estimation using subplot sampling, and this can simplify sampling. For instance, determining plant density across a macroplot would be quite time consuming. However, by using microplots located within the macroplot, density can be sampled more quickly and with sufficient accuracy and precision. All site attributes such as slope, aspect, and elevation are recorded at the macroplot level.

Lastly, the FIREMON documentation uses some terms to describe spatial elements that need to be defined. *Stratifying factors* are defined by the project objectives and are the characteristics used to divide the treatment area or landscape into strata. *Polygons* are areas that exhibit unique characteristics in relation to the adjacent polygons and are usually defined by overlaying the different strata. Polygons can be defined by hand-drawn maps or electronically mapped in a Geographical Information System (GIS). A *Sampling stratum* is made up of the polygons that have similar attributes, as defined by all of the stratification factors. For example, if tree density and fuel load stratum were overlaid, a number of polygons would be defined; some polygons would have low tree density and low fuel load, some with high tree density and low fuel load, some with high tree density and high fuel load, and so on. All of the polygons that had low tree density and low fuel would be in the same sampling stratum; all of the polygons with low tree density and high fuel load would be in another sampling stratum, and so forth. Each polygon will belong to one of the sampling strata. A *landscape* is a large area that can be any size and shape but spatially defines stands and is composed of continuous polygons. The *sample landscape* is the area to be sampled in a FIREMON project and is often described by the prescribed burn map or wildfire map. In statistical terms, the sample landscape defines the population about which inferences will be made.

**Sampling Intensities**

There are three ways to get things done: good, fast, and cheap. Unfortunately, we can only manage for one and compromise on the remaining two. The FIREMON ISS allows users to choose between three levels of sampling intensity based on the project objective(s) and sampling resources. This three-level strategy provides a context for striking a compromise between good, fast, and cheap:

Simple sampling intensity (Level I): Fastest and cheapest while still collecting useful data in the context of the management objectives. This scheme is used if there is limited time, money, or personnel available to complete the monitoring tasks. The data collected in this effort are usually qualitative and not suitable for statistical comparisons.

Alternative sampling intensity (Level II): Somewhat fast, somewhat cheap, and somewhat good. Statistically valid data collected as efficiently as possible but with poor estimates of variability. This scheme is used if defensible numbers are needed from the monitoring effort, but there is limited time and/or resources. Caution must be used in statistical inference due to the low number of samples that can be collected.

Detailed sampling intensity (Level III): Provides the most statistically defensible data, but most methods are slow and costly to implement. Data are statistically valid with appropriate estimates of variation but with high collection costs. Use this scheme if the most statistically valid estimates are needed, and time and money are not limiting.

These three sampling levels are implemented at two spatial levels—landscape and polygon. The fire manager must pick a sampling level to monitor landscape conditions and one level to monitor polygon-level conditions. This decision is based on the *sampling objectives* and the *sampling resources*. The sampling levels for each spatial scale may or may not be the same. For example, a land manager may not care about fire effects across the landscape, such as with a prescribed burn, but is more concerned

with the polygon level changes across the burn unit. In this case, a fire manager may decide on Level I landscape sampling intensity and Level III polygon intensity. Another fire manager may not care how a wildfire burned at the polygon scale but wants to know general characteristics of how the fire burned across the landscape. In this case, landscape Level II or III would be selected while polygon Level I or II might be selected, depending on time and resources.

We refer to the sampling intensity levels frequently throughout the FIREMON documentation. However, they are intended as guidelines, not as rigid criteria. FIREMON allows the user to design sampling strategies at any level of intensity or complexity because the FIREMON procedures and methods have been constructed to be flexible and robust. For example, the Alternative Sampling Intensity, LEVEL II, may suggest that all trees above 4 inches DBH be measured individually using the TD method. However, the FIREMON architect can select any threshold DBH to accommodate the sampling objectives and the resources that are available. As long as the change is documented in the project records there would be no problem with dropping the diameter threshold from 4 to 2 inches, for instance. We have provided a metadata (MD) table in the FIREMON database so that changes to the sampling methods can be recorded and recovered easily.

**Sampling Approaches**

There are two basic sampling approaches used in the FIREMON sampling strategy. The first is the *relevé approach*, used extensively in many ecological vegetation studies during the past 50 to 70 years. The relevé approach is used when documentation of important ecological characteristics is more important than statistically valid estimates of change. When using the relevé approach, one plot is placed in a representative portion of the stand or polygon "without preconceived bias," that is, the plots are not located to make the sampling results look good but, instead, are located with bias in order to represent the general conditions of the polygon or sampling stratum. Representativeness is based on stand history, vegetation composition, stand structure, and a host of other ecological attributes. The advantage of the relevé method is that the fire manager can choose where to locate plots based on past experience, management objectives, and crew safety. For example, the manager may wish to use a relevé approach if the restoration of an important plant community is the objective and the manager wants to make sure that the plots land inside this community. The disadvantage is that this approach is somewhat biased, and plot locations can be manipulated to influence monitoring results, making subsequent statistics highly suspect.

The next approach is the familiar *statistical approach* utilized in most natural resource inventories using systematic, random, or cluster plot establishment. Systematically established plots are distributed following a preset pattern, usually on a grid. Randomly established plots are located using some sort of random number routine. They are not regularly distributed across the sample site and will have some level of clumping. In FIREMON we describe how to purposely cluster plots in adjacent polygons or sampling stratum to allow less travel time between sampling locations. This is not the same as the traditional statistical method of *cluster sampling*, which can be quite complex and is outside the scope of FIREMON. The use of cluster plots can be problematic, statistically, because plots may not be distributed well enough to quantify variance, may not be independent, and samplers have an opportunity to place plots with bias. Despite these potential shortcomings, cluster plots have the advantage of allowing managers to sample a number of polygons relatively quickly.

Plots that follow a regular pattern are easier to relocate, so systematic sampling is recommended for sites that will be sampled multiple times. There is one cautionary note about systematic sampling. Ecologists have noted that some ecological variables have a periodic nature, that is, they vary across the landscape with some predictability. If fire managers develop a systematic plot design that happens to correspond to the periodicity of the attribute being sampled, the sampling results will be biased. The chances of this situation happening are quite small, however, and the convenience of being able to easily relocate sampling plots far outweighs the potential for biased results.

With the statistical approach, the emphasis is on gaining a statistically sound estimate of the sampling entities. It is assumed that the random or systematic establishment of macroplots across a landscape

will adequately quantify the variability of sampled entities so that the entities can be compared using standard statistical tests. However, the only way to be certain that all characteristics a fire manager is interested in monitoring are sampled adequately is to design the sampling program with sufficient intensity to describe the variance of the most variable characteristic. Sampling at this intensity will probably lead to increased sampling effort.

A stratified approach describes how FIREMON plots are established across the landscape or sampling strata based on some land type stratification. The land stratification is based on the site characteristic or characteristics of interest. For instance, a fire manger may want to examine the effect of prescribed fire on exotic weed cover (one stratifying factor) and on sites with different fuel loads (the second stratifying factor). If there were three classes of weed cover and three of fuel load, the potential number of sampling strata would be $3^2$ or 9. Within the stratification, plots can be established using either a random or systematic approach. Both are well documented in the literature and have a proven track record, but most fire monitoring projects are designed using a stratified systematic plot approach for the reasons previously stated. Stratified sampling can reduce the overall cost of the project because stratification accounts for the within-stratification variability and that may reduce the total number of plots needed in the monitoring project. Stratification is especially useful if you are interested in examining treatment effects within the sampling strata.

**Sampling Design Keys**

There are three sampling design keys in FIREMON. The first, the **Sample Approach Classification Key**, is designed to help the FIREMON architect determine whether a relevé or statistical sampling approach should be used. The second key, the **Sampling Intensity Key**, is designed to identify the sampling intensity level that is most applicable to the monitoring project. Last, the **Methods Classification Key** is used to guide the FIREMON architect to determine the sampling methods that should be used in the project. Each key uses the sampling objectives and resources to determine the keys' outcome. The FIREMON architect must determine the scale of the monitoring project—landscape or polygon—before using the keys.

Again, the FIREMON keys are not meant to be used as strict criteria on designing a fire monitoring project. They are meant only as guidelines for developing a locally relevant sampling design that optimizes available resources with the quality and quantity of data required to successfully accomplish the project objectives.

**Step-by-Step Procedures**

If you are experienced with sampling methods and strategies or have previously implemented FIREMON fire monitoring projects you may not need to reread the detailed text in the next section. Instead, you can just refresh your memory on the steps needed to come up with a viable sampling design. In this case we have condensed the FIREMON ISS section into a series of step-by-step instructions to guide the design and implement your fire monitoring project (fig. ISS-1). These instructions are in the **Sampling Strategy Checklist** section and should be used as a quick reference for your monitoring project.

**PRELIMINARY SAMPLE DESIGN ACTIVITIES**

In this section, the FIREMON sampling architect performs some preliminary tasks and analyses that will help design an integrated monitoring project using FIREMON sampling design strategies, techniques, and field methods. This section includes the most important design elements and should be sufficient for most managers when they are setting up their monitoring program. There are many texts and Web sites that give an indepth view of sampling design theory—more thorough than what we are presenting here. If you are interested in learning more, a good place to start is: http://statistics.fs.fed.us/checklists/checklists.html.

[Figure]

**Figure ISS-1**—Flowchart showing the general process for designing your FIREMON Sampling project.

We suggest that the FIREMON architect use the Metadata table to store important information for each fire monitoring sampling project. The Metadata table should contain a detailed listing of the project objectives, the resources available to the project, and the logic and reasoning used to design the sampling strategy and data analysis for the project. Also, the outcome of the FIREMON keys should be recorded in the notebook. Take special care to ensure that the decision process at each step is explained. Figures can be included in the MD table using the Document Link field.

The first, and most critical, step in a FIREMON sampling effort is to succinctly state the objectives of the monitoring project. This step should include a definitive description of the sample population. In other words, when completed the objectives should not just identify *what* the project will be accomplishing but also *where*. For instance, will they be applied across a watershed or just in one treatment unit? The next step is to identify the amount of resources available to accomplish the sampling task. Sample size is determined using the objectives and resources. Note that this is different than most scientific studies where objectives and variance determine the sample size. When developing FIREMON we recognized that, for most fire managers, resources determine sample size, not variance. Usually, the fire manager does not have the funds, time, or personnel to undertake a rigorous sampling program. The result is that the FIREMON approach may not always provide data for statistical inference—especially when sampling at the Simple or Alternative levels—or may do so at lower precision or certainty than typically used in rigorous research studies. In lieu of determining statistical significance, the manager may examine monitoring data, note the changes, identify how well the data represent what was seen in the entire treatment area, and then determine the apparent effectiveness of the treatments.

**Stating Monitoring Goals and Objectives**

Succinct and comprehensive goals and objectives describing the purpose of the fire effects monitoring project set the tone for the remaining sample design and method decisions. It would be difficult to overemphasize the value of this step. To the person not dealing with them all of the time, goals and objectives can be difficult to differentiate. Briefly, goals are broad statements describing general intentions whereas objectives tend to be narrowly focused and precise. In terms of fire monitoring, goals generally explain the overall desired outcome of treatment while the objectives are the quantifiable measure used to evaluate the outcome.

Development of specific measurable objectives requires thoughtful reflection on what the FIREMON project manager wants from the monitoring effort. It may be intimidating to anticipate developing these objectives in light of the many diverse goals in fire management, but understanding exactly what questions the monitoring effort are supposed to answer will provide the context in which all other sampling design and implementation decisions are made. For instance, one manager may only want to qualitatively describe the general effects of a fire while another might want statistically valid estimates of change in vegetation and fuels across the landscape.

Many prescribed burns have a single goal of reducing fuel loading. Given that, a good objective statement would be: reduce dead and down woody debris biomass in the 3 inches and greater size class by at least 50 percent after the first burn. Or, if you have a specific desired future condition it could be: reduce dead and down debris in the 3 inches and greater size class to achieve an average fuel loading of 5 to 10 tons/acre. On the other end of the spectrum, if there were to be no actual measurements performed, the objective could be: complete a walk-through assessment to evaluate the reduction of dead and down debris within 6 weeks of the burn. As an example, a general goal coupled with specific objectives would be as follows:

Restore ecosystem processes and characteristics to pre-1900 conditions by:

1) Reducing fine woody fuel loadings by 80 percent or more after the first burn.
2) Reducing coarse woody debris by less than 50 percent after the first burn.
3) Killing 90 percent or more shade tolerant seedlings, saplings, and mature trees within 1 year of the second-entry prescribed burn.

4)  Providing for at least 50 percent or greater survival in seral, shade-intolerant mature trees within 1 year of the second-entry prescribed.

5)  Reducing duff depths by at least 10 percent for each prescribed burn entry.

6)  Opening tree canopy by at least 50 percent after the first burn.

The FIREMON architect will have to decide if all the objectives can be accomplished with one burn and avoid conflicting objectives.

There is a downside to specifying detailed objective statements in that the monitoring project may become complex and expensive in order to monitor all the important characteristics. Additionally, it may be difficult to achieve all objectives with just one burn. The ecosystem characteristics important to evaluating the success or impact of a burn should be explicitly stated in the objective statement to guide sample design with the recognition that some objectives may be met earlier in the monitoring sequence than others. In other words, all objectives might not be met with one prescribed burn. Try to make objectives broad enough to facilitate an efficient sample design while being specific about the most important ecosystem attributes that must be treated.

Do not think of objective statements as static contracts of purpose and need. Objective statements should be modified and refined as the project proceeds in design and implementation. In fact, objectives should be altered as new information and resources become available—this is a basic tenet of adaptive management. Sometimes environmental factors can influence the sampling that can be done. If snow comes early or stays late on a sampling location then a survey of down dead fuel cannot be accomplished and objectives relating to down dead woody fuel would need to be postponed or eliminated. It is more desirable to add objectives than eliminate them, but you have to recognize that some circumstances are beyond your control. Lastly, understand that there might be parts of the objective so important to the project that they absolutely must be evaluated at any cost. For instance, say you are treating a Research Natural Area where an important plant population resides and you have an objective relating to identifying and tracking changes in the plant community. It would be critical to have a botanist on the crew to accurately identify all of the plant species so that the fire effects on the community can be determined. If the botanist leaves for another job you cannot just drop this objective, as it is critical to the project. Critical objectives like these should be noted and remain unchanged in the objective statement.

If you are planning on implementing a statistically based monitoring plan, then as you write the objectives, you should also consider remarking on the minimum amount of change you want to be able to detect and the confidence level that will be used for the analysis of the monitoring data. Both values affect the sampling intensity and should be indicated for the attributes most important to the project objectives. The minimum detectable change (MDC) parameter is an absolute value calculated by multiplying the mean of the attribute of interest by the percent change of that attribute you want to be able to detect. If you want to be certain that you are detecting a 10 percent reduction in down woody debris on a site that has 25 tons/acre the MDC is equal to 2.5 tons/acre. The confidence level is a measure of the certainty in which you state your statistical results. For example, a 95 percent confidence level means that you are 95 percent certain the change you identified in your statistical analysis really happened. Or conversely, one of 20 statistical tests will note a significant change in an attribute when actually the change did not occur. Most research studies set the confidence level at 95 or 99 percent. However, monitoring studies, especially those with limited resources, might not need to be as restrictive. The confidence level should never be set lower than 80 percent. As MDC decreases and as confidence level increases, sample size will increase so you may need to balance MDC and confidence level against the sampling resources. There is a further discussion of confidence level, detectable change, and sample size in the Implementing the Statistical Approach section below.

**S.M.A.R.T. Objectives**

While objectives are critical to a well-written project plan, it is clear that writing "good" objectives can be difficult. You don't want a project to be determined a failure simple because the objectives were poorly

written. The acronym S.M.A.R.T. relates to five properties of well-written objectives. As you write your project objectives refer to this list to make sure they are S.M.A.R.T.:

1) Objectives must be *Specific*. They must provide a description of the precision required for the objective and link it to a rate, percentage, or some other value. See the list of six objectives listed above for examples.

2) Objectives must be *Measurable*. There must be a system in place that can measure attributes of interest. In FIREMON we have provided a number of sampling procedures. However, for some attributes, such as water quality, we do not provide a method. In such cases you must be able to determine your own sampling procedures and apply them appropriately.

3) Objectives must be *Achievable*. Make sure what you are proposing can be and should be accomplished. For example, an objective that states, "Eliminate 100 percent of the exotic, invasive plant species after 1 year of treatment," is not valid, realistically.

4) Objectives must be *Relevant*. There is no point in making an objective that your treatment will have little or no influence over. Say the agency you work for has a goal of improving air quality in the watershed where your treatment unit is placed. You may have the ability to burn on a day that will reduce the negative impacts on the air quality across the watershed, but an objective that states, "Reduce PM2.5 emissions across the watershed," is not relevant to your treatment because, through your treatment, you cannot effectively control the other sources of PM2.5 emissions across the watershed.

5) Objectives must be *Time Based*. This one is simple—you must have a date or timeframe for completion of the objective. The start time is usually intuitive because generally it begins with the application of a treatment; however, if it is not obvious, clearly state the start date or timeframe.

The subject of setting goals and objectives has been covered extensively in other texts. A quick search of the World Wide Web will help you locate them. The National Park Service and U.S. Fish and Wildlife Service fire monitoring guides are also available online and provide information for fire related projects.

**Determining the Sample Area and Spatial Stratification**

Perhaps the most important element of a monitoring program is *where* the treatments and subsequent monitoring project will be implemented. A detailed geographical description of the area to be sampled is an absolute necessity because it will also provide context for design descriptions. In statistical terms this description provides the *scope* of the treatments and, in most monitoring programs, the scope of inferences made by the statistical tests. Boundaries of the entire sample area should be explicitly stated and diagrammed on an appropriate map. In most cases, large scale maps (such as National Forest maps) will not provide the detail needed for a fire monitoring effort. Maps with a scale less than 1:30,000 will do a better job of accurately delineating the project area.

The entire sampling area must be spatially divided into sampling stratifications that match the sampling objective. Most resource managers delineate areas of homogeneous vegetation (stands), but fire monitoring can be stratified by other classifications such as aspect, slope, fuel condition, or land ownership. FIREMON presents procedures for mapping areas of similar fire severity from satellite imagery (see Landscape Assessment section), and the manager can also use severity as a stratifying factor.

As you define your strata be sure to match the mapping criteria with your sampling objectives. For example, if ponderosa pine restoration is a primary objective, then be sure the strata mapping guidelines delineate various successional stages of ponderosa pine communities. The sampling design can incorporate more than one stratification factor. For instance, a possible design might be to install FIREMON plots in all old-growth ponderosa pine stands that have slopes less than 50 percent and are on National Forest lands.

Figure ISS-2 shows three ecological characteristics mapped on a sample landscape: A) three levels of tree density, B) two levels of dead and down fuel load, and C) a corridor of exotic weed invasion along

[Figure]

[Figure]

[Figure]

[Figure]

**Figure ISS-2**—Overlay maps of strata defined by the stratifying factors in your monitoring project to identify the different polygons on the landscape. Once a sampling design has been determined, the polygons will be sampled with FIREMON plots. In this figure the strata of A) tree density, B) fuel load, and C) exotic weed invasion are overlaid to identify the sample polygons in D. Each shade and/or patten combination represents a specific sampling strata. There are 17 polygons grouped into 9 sampling strata.

the roads. The levels would be determined by the FIREMON architect based on project objectives. In D, characteristics A, B, and C are combined to identify nine sampling strata divided into 17 polygons. One stratum has low tree density and low fuel without exotic weeds, another has low tree density and low fuel with exotic weeds, another has moderate tree density and low fuel without exotic weeds, and so on. Potentially there could have been 12 strata in this example (3 x 2 x 2 = 12) but not all of the combinations occurred. Note how quickly adding ecological characteristics and levels increases the potential number of sampling polygons, which in turn increases the complexity of the monitoring project. This example landscape will be used for demonstration throughout the ISS.

The mapping of sampling entities across the landscape is greatly dependent on the type of fire: prescribed burns or wildfires (*postevent monitoring projects* versus *complete monitoring projects*). The difference is that for wildfires and wildland fire use, fire effects monitoring plots are installed after the fire, whereas prescribed fire monitoring plots are measured both before and after the burn. Most wildland fire use burns (previously called prescribed natural fires) fall into the postevent category because of the absence of preburn plots. In these cases, sample stands must be identified after the wildfires using remotely sensed images (aerial photos or satellite imagery) taken before the fire if fire effects measurements are to be summarized by vegetation type. If fire severity stratification is necessary the Landscape Assessment methodology can be used.

The mapping effort, and its integration with FIREMON sampling efforts, can be made much easier if the mapping and analysis are done within a Geographical Information System (GIS). A GIS allows complex queries on landscape and stand attributes that make design and subsequent implementation of a FIREMON sampling strategy efficient. A GIS can produce maps of the sample area for reference and navigation, and the sampled FIREMON field data can be linked to the GIS for many other applications (landscape pattern analysis, satellite imagery mapping).

Two statistics must be computed once the sample area has been mapped and the landscape divided into polygons. First, compute the total treatment area of the study site(s). Exclude all areas that will not be sampled (talus slopes, lakes, glaciers) from the estimate. Then, compute the number of polygons or stands within the area to be sampled. These statistics will be used to determine the resources needed to accomplish the sampling.

The sampling environment, like the project goal and objectives, provides the spatial and logistical sideboards for project planning. There are four attributes about the sampling area that must be known before sampling design can continue: 1) size of area, 2) topographic complexity, 3) transportation network, and 4) ecological characteristics. The size of the sampling project is often dictated by the boundary of the burn, and burn boundaries are notoriously coarse, so it is important that a precisely

developed burn map is provided for monitoring. Topography will dictate many aspects of the sampling effort. Steep, dissected landscapes will be difficult and dangerous to navigate, so the sampling project should be designed to accommodate or avoid these troublesome conditions. The network of roads, trails, and navigable terrain will provide the means of transporting crews to sampling areas. Remote areas with only trail access will require another level of planning because crews will probably need backcountry supplies along with the already extensive sampling gear, and this may require packstock support (mule and horse packing). Last, the ecological characteristics of the sample area will dictate the sampling design and methods. Forested environments will probably require time-intensive individual tree surveys, while rangeland types can be sampled using standard vegetation surveys. Areas with thick vegetation or high fuel loadings will be difficult to traverse. And areas with abundant threatened and endangered species will require a high resolution sampling design to properly evaluate fire's impact in small but highly valuable habitats.

**Determining Sampling Resources**

The details of the FIREMON monitoring project design are determined by striking a compromise between cost, personnel, time, logistics, and sampling environment within the context of the project goals and objectives. For example, say that monitoring on the Clear Creek burn is essential to determine tree mortality and subsequent potential for salvage logging. The project goal might read, "Determine tree mortality and salvage potential." This statement provides critical information to determine what and how to sample for monitoring and evaluation. Obviously, the project goals aren't related to weeds, grazing, or fuel consumption, so sampling techniques that measure plant cover, plant biomass, and fuel loadings are not needed. A tree population sampling method is most appropriate here. Next, say there is limited funding, and the only people available are the fire crew, and there are only 3 weeks to perform the monitoring tasks. This means that a stratified random sample across the entire burn is inappropriate because it would cost too much and take too much time. However, a relevé approach might be the right compromise between sampling resources and project desires. Because the data must be used for two purposes—to determine tree mortality and the amount of timber in those trees (salvage potential)—a detailed, individual tree sampling method is warranted.

The FIREMON architect should consult with the FIREMON project manager to determine the exact amount of resources available to conduct the fire monitoring project. There are four types of resources that should be evaluated: 1) funding, 2) personnel (number of people and their expertise), 3) logistics, and 4) time. All of these resources are somewhat related, but each resource should be carefully appraised to determine its contribution to the monitoring project in the context of the extent and complexity of the sampling area.

Funding is easily the most important sampling resource because it dictates the level of all other resources. It is critical that the FIREMON architect knows the exact amount of money dedicated to the monitoring effort. This will help determine the number of people to hire, the number of vehicles to acquire, and the quantity and number of supplies to purchase. In short, funding often determines sampling intensity.

The number and qualifications of people to use in the FIREMON project is an important resource for the monitoring project. It is essential that the skills of the FIREMON field crew match the level of detail of the data to be collected. For example, the monitoring of plant species cover change requires a field botanist who can consistently and comprehensively identify vascular and nonvascular plant species. It is also important that the field crews have sufficient training in FIREMON methods and techniques. A poorly trained crew will invariably spend excessive amounts of time and money collecting questionable data that will be useful to no one. As funding often dictates sampling intensity, the experience and capabilities of the field crew will determine the quality of sampled data.

It is important that the logistic capacity of a FIREMON project be identified prior to designing a sampling project. Critical elements are 1) the number of available vehicles, 2) the amount of sampling equipment, 3) the amount of camping gear (if needed), and 4) computer equipment. Often, available vehicles and equipment can limit the staffing of monitoring projects. Required sampling equipment

(compasses, clinometers, GPS units) must be available or rapidly and easily purchased. Maps of the sample area are absolutely essential for conducting a successful monitoring project. Laptop computers may also be used for data entry and reference to the FIREMON methods in the field. Logistical support determines the sampling ability.

The amount of time available to conduct the monitoring project can, in some circumstances, dictate the level of other resources. For example, it may be critical to establish monitoring plots across a large burn to determine appropriate levels of rehabilitation. To accomplish this objective, the sampling must take place directly after the burn and before the snow flies. This does not leave abundant time to mobilize extensive field crews and acquire new equipment and vehicles. Projects on fast timelines may need to forego extensive, statistically valid sampling designs in favor of relevé methods. The amount of time dictates the schedule of a sampling project.

The availability of funding, personnel, logistics, and time should be explicitly stated in the FIREMON field notebook. Obviously, the status of any of these resources can change; a good FIREMON architect will ensure there is plenty of flexibility in the sample design to accommodate changes in available resources, whether the changes are good or bad. There may be other resources or challenges to be included in the design of the sampling effort that are not mentioned here; for example, weather. Excessive rain or heat may hamper the sampling productivity of crews.

**Determining Sampling Design**

When the major sampling resources have been identified and described they will be summarized into the FIREMON statistics that are used in the sample design keys. Be sure to document the calculation of the sampling resource statistics in the monitoring notebook. A number of these parameters will be hard to estimate when you first start your monitoring activities but will be easier to determine as you gain experience. When possible we have provided some guidelines for your initial values. When making your own estimates use your best judgment and be realistic about the numbers you chose. Remember, monitoring almost always takes longer and costs more than you think it will.

**Calculating FIREMON sample statistics**

The first FIREMON sample design statistic that you will be calculating is the *Sampling Potential (SP)*, which is used to indicate the number of standard plots that can be installed during the sampling effort. This statistic integrates most sampling resources into one index that can describe the capacity to perform the monitoring project. SP is a function of project funds, crew costs, and plot production rate. Crew costs and plot production rate will probably need to be estimated.

You should first determine the amount of money available to conduct the entire monitoring effort *(Project Funds or PF)*. This amount should include salaries of existing personnel available to work on this project, including the FIREMON project leader and architect.

Next, estimate the *Crew Costs (CC)*. If necessary, include the cost of renting a vehicle for the period of sampling. Assume one vehicle will transport two people for the project. Calculate how much it will cost to outfit each crew with supplies, and if this figure is unknown, use $250.00. You will need to estimate the number of 8-hour workdays available to finish the monitoring project. Provide an estimate even if there appears to be plenty of time to finish the project. Use a target start and end date as a guide; try to identify a realistic day for starting the project and ending the project, then count the number of working days in between. Estimate the number of days that could be lost to inclement weather, if that is a possibility, and days lost to organizational, administrative, logistical, or personnel problems. In lieu of this information, add an additional 10 percent time to the project. Using your estimate of workdays, estimate the salary of one crew person for the duration of the monitoring project and multiply by the number people you plan to have on the crew. Finally, determine the CC by adding the transportation, equipment, wages, and any other expenses together and dividing by the number of workdays.

Last, *Plot Production Rate (PPR)* is an estimate of the number of plots that can be sampled per day by one crew. Early in the monitoring process PPR is probably unknown so use the rate of four plots per day

as an estimate. In our experience the major factor in PPR is crew transport to the sample sites, not the actual sampling.

When you have determined PF, CC, and PPR use the following formula to calculate SP:

Equation ISS-1
$$SP = \frac{(PF)(PPR)}{CC}$$

where SP is the sampling potential (plots per crew for the entire project), PF is project funds in dollars, CC is crew costs in dollars per day, and PPR is plot production rate in plots per day per crew.

Here is an example of estimating SP: Assume that a manager has $3,000.00 to spend on installing monitoring plots the first year of a project. He has identified four people that could work on the crew and, after reviewing the other tasks that need to be done during the season, apart from this monitoring project, he notes that there are 10 work days that can be spent on the project. A best guess from past experience tells him that the crew can get about six plots done per day. All equipment and transportation is on hand, so the only expense is the crews' wage. Adding all of the wages together he notes that the crew costs about $400.00 per day. Dividing $400.00 into $3,000.00 he finds he will get 7 days of sampling in before all of the available funds are spent. The final calculation of SP is:

$$\quad \frac{(3000)(6)}{400}$$

Thus, there are resources to sample about 45 plots. If there is a question about the ability to assess the effectiveness of treatments using data from 45 plots, then the revision of project objectives, sampling methods, crew size, and/or the scope of the area to be tested with statistical inference must be revisited to bring the sampling intensity in line with the project objectives. See the Considering Tradeoffs section for more information.

Note that in this example the number of available workdays was not the limiting factor in calculation of SP, so it was not taken into account (the manager potentially had 10 days for sampling but the project funds only allowed 7 days of sampling). If there were only five days available for sampling then the available workdays *would* enter into the calculation, further lowering the SP. Only about 30 plots could be sampled (5 days X 6 plots/day).

From this simple example it is clear that part of the art of monitoring is balancing all of the components in the monitoring program so that the data collected are useful for assessing the treatments. Almost every component, including the objectives, sampling crew, sampling approach, sample size, monitoring area, sample stratifications, and study area can be modified to bring the monitoring data in line with the project objectives.

When you start your fieldwork, divide the samplers into crews making sure that the number and expertise in each sampling group are appropriate to the monitoring tasks. For example, don't send out a crew and expect them to collect species level data accurately without a good botanist in the group.

**Calculating the number of polygons to be sampled**

The next set of statistics attempts to quantify the amount of sampling required for the area to be monitored. First, determine the total size of the *Sample Area (SA)* in acres. This is often all the area within the burn boundary. Next, compute the *Number of Polygons (NP)* that compose the sample area. If you have determined your polygons by overlaying the sampling strata as shown in figure ISS-2, then this is as simple as adding up the number of polygons. If you are interested in sampling by stands within a prescribed fire, they were probably mapped prior to the fire treatments being set up and just need to be summed across the sampling area. For wildfire situations, NP can be taken from stand maps, satellite-derived vegetation cover type maps, or burn severity maps created prior to the fire. However, this type of spatial data will not be known for many monitoring projects. If NP is not known for your project, you can estimate it using average stand size. In the Northern Rockies we have commonly

estimated the average stand to be 25 acres (10 ha) and used that value to calculate NP (NP = SA/25). This method assumes that the sampling will be stratified in space by stand characteristics (tree species, diameter, height, and so on). If this is not the case for your project, divide the landscape into the appropriate homogeneous sampling units. For example, burn severity polygons mapped from satellite imagery may be your spatial stratification. Or it could be treatment blocks within stands. The NP statistic is the number of spatially explicit sampling polygons in your project.

Sometimes there is no need to sample all of the stands or polygons in the sampling area. For example, monitoring plots might only be needed on steep areas where rehabilitation efforts will be prevalent. Or perhaps only forested areas need to be sampled to monitor tree establishment after wildfire. In these cases, calculate SA or NP only for those areas that are targeted for monitoring.

The NP and SP statistics are used to determine the suggested sampling approach and sampling intensity level. However, these statistics, and the statistics used to calculate them (CC, PPR, SA), can be modified to refine sampling strategies to fit the monitoring objectives or to generate several sampling alternatives for strategy design. For example, the sampling rate (PPR) can be doubled or halved to produce best and worst case sampling scenarios. Or NP or SA can be reduced or increased to match the sampling potential.

**Determining the sampling approach**

The next important step in designing a FIREMON sampling strategy is to decide on a sampling approach to collect fire effects data. The two basic FIREMON approaches are *relevé* and *statistical*. Both these approaches can be stratified by any landform, ecological attribute, or disturbance characteristic. The selection of the appropriate approach will dictate nearly all other sampling details.

The **Sample Approach Classification Key** provides the guidance to select the approach to match your project. Read down the list of statements for each approach answering "yes" or "no" to each item that is important to the sampling project with special reference to your monitoring objectives, sample area, and available resources as mentioned above. Simply count the number of yes answers and no answers in each list. The approach with the most yes answers is the approach that probably should be used in the monitoring project. However, this checklist does not include all the subtle advantages and disadvantages of each approach with respect to your unique sample area. There may be other important elements that will influence your final decision. Be sure to document these special conditions in the FIREMON notebook for future reference.

In general, the *relevé* approach is used when time, money, or personnel limitations require the sampling to be done quickly but without compromising the temporal aspects of monitoring at the stand level. However, it is important to recognize the limitations of the relevé approach. First, the within-polygon variation is not quantified, so a statistical comparison across polygons (comparing one polygon to another) is not valid. Next, the variation of ecosystem elements, such as fuels, trees, or plants, is not measured across space, so a statistically valid landscape comparison is not possible. The only possible statistically valid comparison using the relevé method would be the comparison of one single plot measured at two time periods using FIREMON methods that captured within-plot variation. The independence of microplot samples is suspect, however, so temporal inference may not be appropriate. There may be an inclination for managers to use relevé plots because of the difficulties in estimating the variance of one or a number attributes, but if statistical inferences are to be made, estimates of variance for the most important entities (at least) need to be made so that the statistical approach can be applied.

The *statistical* approach is used when the answers obtained must be compared using statistically valid procedures across space and time. This is the most useful and most commonly applied approach for monitoring projects.

**Selecting the level of sampling intensity**

The next step in sample design is to determine the level of sampling intensity for the monitoring project. Sampling intensity is usually described by the number of plots located across the project area and is

related to the amount of variance to be explained. The more plots established across the landscape, the more likely that the range of response and variance has been captured in measurements of fire effects. This translates to more accurate and defensible comparisons and evaluations.

FIREMON has three levels of sampling intensity integrated into the sampling strategy to facilitate sampling design. These levels are intended only as guides for the inexperienced designers and not as recommendations. The FIREMON architect can design a monitoring project at any intensity level, not just the three mentioned here. Refer to the **Sampling Intensity Key** to decide which intensity level best fits your situation.

**Choosing sampling methods**

This section describes how the FIREMON architect selects the sampling methods to employ at each plot during the project. Many people think this is one of the most difficult parts of a monitoring study, but in fact, choosing sampling methods is straightforward because you simply match the monitoring objectives to the attributes that need to be measured. Many managers get confused in sampling methods selection because of the complexity and diversity of sampling procedures available, but the selection process becomes simple when the decision is put in context of the objectives.

Methods for measuring fire effects are selected from the **Methods Classification Key** provided in FIREMON. Read each bullet in the Methods Key then refer to each of the project objectives to see if the bullet is true, and if so, employ the suggested method. Again, this key is intended as a guide and not a prescription. Use your own intuition and experience to modify results from the key to fit your special circumstances. FIREMON has been developed using established methods. Occasionally you may find that there is not a method that will assess the success of some objective. For example, there is no water quality sampling method in FIREMON. Thus, methods may need to be developed to monitor some attributes. These should be explicitly described in the MD table so that the exact procedure can be applied at the next sampling visit. Optionally, you might be able to add fields to existing methods to meet the objectives. For instance, if the Wildlife Biologist is interested in the presence of snag cavities, a field could be added in the Tree Data (TD) table of the database. (Use caution when adding fields to the FIREMON database, as it will make it difficult to merge your data with other FIREMON data.)

You will find that most of the time and money spent on field campaigns are in transporting crews to sampling areas and not on actual sampling. Therefore, it is often prudent to sample additional attributes at the FIREMON plot to strengthen monitoring analyses and to widen the scope of the monitoring project. This is especially true if the FIREMON architect is wondering whether or not to sample a particular attribute. It is much better to spend an additional 10 to 20 minutes on the plot sampling another fire effect, than it is to be frustrated because some component wasn't measured at the end of the sampling effort. For example, measuring crown characteristics for every tree on the macroplot may seem excessive if the sampling objective is to assess tree mortality, but those crown characteristics (percent crown scorch, tree DBH, height) could be used to develop salvage guidelines from percent crown scorch or predict crown fire potential using NEXUS (Scott 1999).

The process described in this section provides the FIREMON user important information on the elements of a sampling project that can be modified to fit the monitoring objectives. It is best to first compute all sample statistics from real data and then key the sampling approach, intensity, and methods. Then compare the key results to the monitoring objectives again to evaluate if the key results are appropriate. If not, go back and modify one or more sample statistics to achieve a more realistic result. Of course, if you modify a statistic, you must then implement that modification in the sample design. For example, if you reduce NP then you must make sure that the correct number of polygons are mapped. Alternatively, the FIREMON architect may reconsider the scope of the sampling (sampling a different, usually smaller, area than originally proposed) or reconsider the project objectives. This may result in fewer plots and/or a smaller area being sampled, but doing so could improve the quality of the collected data. Experienced sampling crews will be able to determine the most sensitive and important statistics and ensure that these attributes are well represented in the sample design.

**Considering Tradeoffs**

The design of the sampling strategy is a constant tradeoff between statistical significance and logistical feasibility. The only way to have both is with sufficient funds, experienced personnel, and ample time. Unfortunately these three factors rarely coincide; therefore, a compromise in sampling rigor is usually necessary. The compromises of your FIREMON sampling design should be recorded in the FIREMON notebook or MD information to ensure the data are never used for inappropriate purposes.

Careful consideration should be given to assess whether or not the approach identified in the sampling approach key will actually accomplish the sampling objectives. For example, say one sampling objective is to quantify significant reductions in fuel loadings after a prescribed burn, which absolutely requires a statistical approach, but limited funds and personnel lead you to identify a relevé approach in the FIREMON sampling strategy design. The relevé approach will only provide qualitative descriptions of changes in fuel loadings and will not detect statistically significant differences in loading after treatment. So either the sampling objective must change to remove the statistical requirement or the statistical approach must be implemented.

The main limitation of the relevé approach is that it does not provide for any analysis of statistical significance, even if multiple relevé plots are established on a sample site. The subjective location of the relevé in a representative portion of the stand biases the sample and does not fulfill the assumption of randomness needed for classical statistics. Relevé plots are used only for qualitative reasons or descriptive purposes and should never be used to quantify changes in ecosystem characteristics. However, relevé methods allow for efficient collection of complex data over large land areas with limited resources.

The main limitation of the statistical approach is its high expense in time, funds, and personnel. A good statistical sample requires multiple plots in homogeneous areas (landscape stratification) and that often necessitates extensive resources, especially if many entities are being measured such as fuel loadings, tree populations, and vegetation cover. However, the statistical approach is *required* if changes in ecosystem characteristics must be quantified with some test of significance.

There are many ways to compromise sampling rigor with logistic restrictions and settle on a tradeoff between statistical validity and general description. If a statistical approach is necessary but time and funds are limited, it may be possible to reduce the number of entities being sampled. For example, modify the design to measure only fuel loadings and do not sample tree populations and plant cover. Or if a large landscape is treated, it might be possible to statistically sample a small representative area within the large area and sample the remaining areas with a relevé approach. Recognize that the statistical sample could only be used to quantitatively describe changes in the representative area and that extrapolation of those results to the large area would be highly questionable, especially if not supported by data collected from the relevé plots. Optionally, it may be possible to aggregate the sampling strata to minimize the number of sampling areas. And of course, there is always the possibility of optimizing sampling efficiency by cluster sampling around accessible locations. In any case, be sure to document the limitations of your tradeoffs so that others will not use the data for inappropriate analyses.

Be aware that all analyses using the FIREMON software package requires multiple plots within a sample strata to calculate an acceptable measure of variation for statistical tests of significance. If only one or two plots are established in a stratum, then the data can be used only for descriptive purposes, which may not be compatible with the statistical sampling objective. In short, the usefulness of your monitoring data increases with statistical validity. The more samples you collect in a treatment area, the higher the value of that data to other resource efforts. The FIREMON software does not generate statistical tests of significance for one plot across multiple monitoring visits, even if multiple microplot, transect, or belt techniques were used to quantify variation at the plot level. In FIREMON, the only way to calculate a variance is with multiple plots.

**Monitoring Prescribed Burn Projects**

Ideally, every sampling polygon will be monitored to sufficiently track treatment effects. Identifying the polygons should not be a difficult task for a number of reasons. First, conventional prescribed burn projects require intensive planning and public involvement, and as a result, there is plenty of

documentation and data on the area to be treated, including maps, stand delineations, treatment block delineations, and supporting stand and historical fire data. Also, because the objective of most prescribed burns is contingent on preburn conditions and the burn boundary is known prior to the fire, postburn mapping is generally not needed. Lastly, prescribed burns are usually small in area compared to wildfires or wildland fire use. Use the following sample approach guidelines to monitor your prescribed fire project.

**Relevé Approach**—Establish one FIREMON plot in each sample polygon in the prescribed burn area. Locate the relevé in an area that displays the typical ecological conditions within the stand. If you find unique features that you think should be sampled, such as seeps, dense thickets, and pockets of snags, you should locate another relevé for sampling those areas. In FIREMON, we generally do not consider the relevé approach appropriate for monitoring prescribed fire treatments.

**Statistical Approach**—Sample size equations are provided in **Sample Size Determination** section. If you have a variance estimate, use these equations to determine the appropriate number of samples. If the variance is unknown, plan on establishing at least five plots in each sample polygon in the prescribed burn area. Sampling fewer than five plots results in variance estimates that are suspect. If the sample polygon exceeds 50 acres, then establish another plot for every 5 acres in the polygon.

**Comparing Objectives and Sampling Design**

Make sure you haven't specified any inconsistencies in your sampling strategy criteria. For example, the use of the relevé method at the Detailed sampling intensity level (Level III) is not correct or logical. Why use a descriptive method when you have plenty of resources to quantify the range of response and variation in fire effects measures? If you specified Level III intensity, then use the statistical approach. As always, these recommendations are provided to help you work through your design process, not as strict rules. There may be an occasion where the project objectives lead you to intensively sample a landscape but with few measurements taken on every plot. This would lead you to a "low" level of appropriateness in table ISS-1. However, if lots of plots measuring few attributes let you assess your objectives and stay within budget, then you have picked the right sampling scheme.

Finally, take one last thorough look at the project objectives and compare them with the sampling design that you have developed. Ask yourself if you will really be able to assess the effectiveness of the treatments using the sampling design that has been developed. If possible, have others familiar with the project review the monitoring plan. They may identify some shortcoming that you have missed. When you have decided on the approach, methods, and intensity you believe are appropriate for your project, record that information in the FIREMON notebook or MD table.

**DESIGNING A FIRE MONITORING PROJECT**

This section is designed to help the FIREMON architect develop the integrated sampling strategy for a monitoring project using FIREMON sampling design, techniques, and field methods. This section is organized according to the design criteria that were determined in the previous section. If you haven't done so already, use the **Sample Approach Classification Key** and **Sampling Intensity Key** to identify the most appropriate approach and intensity for your monitoring project. After reading **Determining Polygons and Building a Summary Tabl**e proceed to the section below that best fits your strategy criteria selections—relevé or statistical.

Table ISS-1—Appropriateness of sampling intensity levels by sample strategy. High indicates the sampling scheme is highly appropriate; Moderate indicates moderately appropriate; Low means somewhat inappropriate; and Inappropriate means the sampling design is not suggested.

| Approach | Level I-Simple | Level II-Alternative | Level III-Detailed |
|---|---|---|---|
| Relevé | High | Moderate | Inappropriate |
| Statistical | Low | Moderate | High |

**Determining Polygon Locations and Building a Summary Table**

Regardless of the sampling approach you use, monitoring plots must be located across the project area so that the data collected will be useful for assessing the treatments applied. In FIREMON we propose using the project objectives to determine sampling strata then overlaying the strata to identify polygons. By ordering the sampling sequence of the polygons based on the most important project objectives, fire managers will have the best chance of collecting useful data for the project.

Each polygon on the sample landscape must be described by one or more ecological attributes for the prioritization method to work. *It is essential that the attributes used to describe the polygons and/or strata be consistent with the monitoring objectives.* Using the example ISS landscape, the attributes would be tree density, fuel load, and weed invasion. It is best if the attributes you use are stored in a GIS for digital map analysis, but it is possible to do the entire exercise using spreadsheets or simple pen-and-paper analyses.

Some stratification attributes may be secondary to the objectives but important to the project. Any site characteristic that influences fire behavior—for instance, slope—may be important to factor into the monitoring project because the fire could influence the monitored attributes in different ways depending on the behavior.

A good way to determine the stratification criteria is to create a summary table (table ISS-2). The columns in the table are the stratifying factors with polygons ranked by factor levels in the rows. The example summary table was developed by overlaying the strata, numbering the polygons, and grouping them by stratum (fig. ISS-3).

Next, determine the prioritization attributes so that the important polygons get sampled. The prioritization attributes will probably be related to the list of stratification attributes. You can make the prioritization as simple or as complicated as you like. For instance, you could decide that only stands with ponderosa pine cover types will be sampled, or you could decide to sample mature stands of shade-intolerant cover types on steep slopes for each habitat type. Be sure to keep your prioritization flexible enough so you can easily modify the selection criteria.

In the **Implementing the Relevé Approach** and **Implementing the Statistical Approach** sections below, tree density, fuels, and weeds are the prioritization attributes. The prioritizations are described further in each example.

**Table ISS-2**—Develop a summary table to identify the number of polygons in each stratum. Label the first columns using the stratifying factors and list each combination of levels in the rows below. Give each stratum a unique name or code and in the last column list all that polygons that belong in that stratum. Include a table like this in the FIREMON project folder for future reference.

| Tree density | Fuel load | Exotic weeds | Stratum code | Polygon numbers |
|---|---|---|---|---|
| Low | Low | No | LLN | 1, 3 |
| Low | Low | Yes | LLY | 2 |
| Low | Moderate | No | LMN | N.A. |
| Low | Moderate | Yes | LMY | N.A. |
| Moderate | Low | No | MLN | 11 |
| Moderate | Low | Yes | MLY | 4 |
| Moderate | Moderate | No | MMN | 5, 7, 9, 12 |
| Moderate | Moderate | Yes | MMY | 6, 13 |
| High | Low | No | HLN | 8, 10, 16 |
| High | Low | Yes | HLY | N.A. |
| High | Moderate | No | HMN | 15, 17 |
| High | Moderate | Yes | HMY | 14 |

[Figure]

[Figure]

[Figure]

[Figure]

[Figure]

**Figure ISS-3**—Build a summary table by overlaying the stratifying factors to develop polygons, then number each polygon and note the stratum it belongs in. You should include a figure like this one in your FIREMON notebook for future reference.

**Implementing the Relevé Approach**

**Background**

The relevé approach requires that plots be located in a representative portion of the sample stand or polygon without preconceived bias. Data measured on relevé plots are not used to quantify the variation across the stands or polygons, but rather to provide a general description of the polygon and provide a baseline measurement of monitoring ecosystem characteristics for that polygon. The assumption in relevé sampling is that the plot is representative of a larger area (stand or polygon), and therefore conditions measured at the plot can be used to describe the stand or polygon as a whole. Thus, any fire effects measured on a plot can be used to describe fire effects across the entire polygon or sampling stratum. Two drawbacks of the relevé method are 1) the measured effects cannot be statistically compared, spatially, between polygons on a landscape and 2) extrapolation of relevé data across a polygon or stratum is controversial due to the subjective placement of the relevé and inherently high variability of ecological attributes.

**Polygon selection**

Ideally, each stand or polygon on the landscape will have at least one relevé. This is typically not possible because some polygons may be inaccessible, or resources limit the ability to sample the entire landscape.

Therefore, a compromise must be struck between sample frequency and logistics so that the most important polygons can be sampled given the resources available. For relevé sampling the most important factor is ensuring that there is adequate representation in plot frequency so that important conditions within each sampling stratum can be summarized in reports or databases. Base the plot frequency on the area in each of the sampling stratum. For instance, if stratum A has twice as many acres as stratum B, then stratum A should have twice as much sampling. Sample the polygons that are the most representative of the sampling stratum and locate the relevé plots in the most representative area of each polygon. Remember that if you locate unique or unusual characteristics within a polygon, such as a site containing rare plants, they should also be sampled and noted as being not really representative of the entire polygon but used for monitoring unique attributes.

Figure ISS-4 shows how relevé plots might be distributed if put into the most representative portions of each polygon of the sample ISS landscape at the Level I-Simple sampling intensity. Note that not every polygon or even sampling stratum is being sampled. Each weed corridor is sampled with only one relevé, even though there are five possible strata that could have been sampled within the exotic weed stratification. Thus, this example assumes that the FIREMON architect determined that for some objective-based reason weed sampling was not a priority.

Sometimes it is more efficient to group sample locations close together, usually around an easily accessible point, rather than randomly selected throughout the landscape (fig. ISS-5). These "cluster plots" minimize transport time, thus they can be more efficient than using random or systematic sample selection of polygons, especially across large landscapes. The main disadvantage is the introduction of

[Figure]

Relevé Simple

**Figure ISS-4**—this illustration shows plots located using the relevé approach at the Level 1-Simple sampling intensity. There are 10 plots distributed across the sample area. The exotic weed strata are not well sampled.

[Figure]

Relevé Cluster: Simple

**Figure ISS-5**—These 10 relevé sampling sites are clustered into three groups to increase sampling efficiency while at the same time getting a good spatial representation across the sample area.

bias because you are using only a small portion of the polygon to find a representative location for the relevé—there might be a more representative location, but it is outside the of area where you want to locate you cluster. Beyond the benefit of reduced transport times, cluster plots allow you note juxtaposition relationships of neighboring polygons.

Make an attempt to distribute clusters geographically around the landscape using transportation routes (roads, trails, rivers) as cluster centers. Cluster plots do have an element of subjectivity in the placement of cluster centers, but when sampling resources are low, cluster selection is a valuable alternative to random or systematic selection of sample stands.

**Relevé establishment**

Relevé plots are established by navigating to a sample polygon and then visiting various parts of the polygon to find the range of vegetation and biophysical conditions. After examining the polygon the FIREMON crew leader will determine the location of the relevé. The vegetation and biophysical conditions inside the relevé must comprehensively describe conditions across the entire polygon. Representative conditions should be assessed from a wide range of ecological attributes. First and foremost, the relevé should represent the conditions of the polygon that are important to the project objectives. In the example illustrations (fig. ISS-4 and ISS-5) each of the relevé plots were placed in a spot that represents the tree density, fuel load, and weed conditions of the polygon (the stratifying factors presented in fig. ISS-2). Secondly, relevé plots should be located in areas of the polygon that reflect the characteristics of the entire polygon for attributes not related to the stratifying factors. For instance, if the majority of your polygon is gently sloped you would want to avoid locating your relevé in a steep draw because the fire behavior and fire effects there would not be typical of the polygon. Most of the secondary considerations, such as topography, fuel conditions, and disturbance history, will be related to fire. More specific examples are slope, slope position, aspect, elevation, fuel load by size class, fuel condition, fuel model, insect and disease damage, and past fire effects. Note that there may be more than one of location inside the polygon that is appropriate for sampling, and if so, the FIREMON crew leader needs to choose the one that will lead to the most representative sampling given the resources.

The procedure used to establish relevé plots has always been embroiled in controversy. Locating a plot in a representative portion of the polygon without preconceived bias is part fantasy, part science, and part guesswork. The result is that most plot locations will contain some element of sampler bias. However, in complex ecosystems with high spatial and temporal variability the relevé method is generally a simple, efficient, and tenable sampling approach. A good mitigation measure to minimize bias and subjectivity is to mark a plot location, then randomly choose a direction (you can use the second hand on your watch) and place the plot center 50 feet away along the randomly selected direction. Of course, this procedure could lead you to establish the plot outside the representative portion of the polygon you are interested in sampling, in which case you would need to try another random offset.

Crews may encounter a wide diversity of ecological conditions within one sample polygon, making it difficult to locate the relevé plot in an area of representative conditions. In these cases, if it is possible, divide the polygon and put a plot in each division. If the resources are available, crews should also establish plots in areas that are atypical of the polygon as a whole so that those unique sites an also be monitored over time. For example, small seeps, blowdowns, or benches may be included in a polygon because of the coarseness of stand mapping (the fine scale attributes were not discriminated out due to the scale of the mapped attributes). However, these special features should be sampled if there are enough resources. Crews should give sampling priority to features that are important to the project objectives.

The main concern with using relevé plots is to know their weaknesses, strengths, and applications. Relevé plots do not allow a statistical comparison across polygons or strata because of the lack of a spatial measure of variability for sampled ecosystem characteristics. Relevé plots are best used as descriptions of polygons that compose a landscape. Monitoring results from relevé plots cannot be extrapolated across space for statistically valid comparisons because of the missing variability measure.

However, relevé plots are appropriate for broad descriptions of ecological attributes within sampling strata and polygons.

**Using the Simple sampling intensity (Level I)**

This sampling level assumes the number of FIREMON plots that can potentially be established in the monitoring effort (SP) is one-half or less than the number of plots needed to sample the entire landscape (NP). This level is often used with the relevé approach when monitoring is needed but there are few resources to complete the project. Level II will give you information about more ecological attributes (you can sample more polygons), but commonly Level I is the most realistic due to resource constraints.

The goal of this sampling scheme is to sample those stands or polygons that are the most important to fire effects monitoring. This can be difficult because often there isn't enough time, personnel, or funds to sample all of the important polygons. The key to a successful monitoring effort for this scheme is to prioritize those stands on the landscape that need sampling and sample them in order of priority. This means that the FIREMON architect must balance distribution of the plots across the landscape and accessibility with importance to management—a difficult task. Detailed below is a method to select sample polygons using the polygons identified in table ISS-2 above. The architect can vary the theme or strategy to fit local circumstances. Again, this is not a rigid procedural step, but rather a flexible framework for sampling design modification. It is important that a stratification system be explicitly stated and recorded in the FIREMON notebook.

Create a list of polygons to sample, ordering your sampling polygons based on your stratification and prioritization criteria. This list can be generated from a GIS or from a spreadsheet. Remember that this list is spatial in nature and does not take into account proximity and adjacency to other stands unless specifically designed and sorted.

If we use the ISS example landscape stratification described in the **Determining Polygons and Building a Summary Table** section then select two prioritization attributes—1) fuel load, especially in moderately loaded areas, and 2) exotic weed invasion, in that order—we can make up an ordered list of polygons for sampling (table ISS-3). In this example we will assume that SP is equal to nine. Figure ISS-6 shows how the plots might be located across the project area. For clarity, in this example tree density has also been used to order the polygons. However, tree density is not a prioritization attribute, so polygons that have the last two letters of the stratum code in common ("MN" in "MMN") could be sampled in any order. Sampling the polygons that are closer together first, will lower the sampling time.

We recommend that you have a list or map of polygons by stratification attribute in order of priority for field crews so that they can alter sampling if obstacles are encountered in the field. Remember to allow enough flexibility in the prioritization and selection process so that the field crews can modify sampling

Table ISS-3—Sites with moderate fuel load and weeds were used as prioritization attributes, then the strata in table ISS-2 were ordered to identify the polygons for sampling.

| Prioritization attribute 1 | Prioritization attribute 2 | Stratum code | Polygon number | Order of priority |
|---|---|---|---|---|
| Fuel load— moderate to low | Exotic weeds | HMY | 14 | 1 |
| | | MMY | 6 | 2 |
| | | MMY | 13 | 3 |
| | | LMY | N.A. | |
| | | HLY | N.A. | |
| | | MLY | 4 | 4 |
| | | LLY | 2 | 5 |
| | | HMN | 15 | 6 |
| | | HMN | 17 | 7 |
| | | HLN | 8 | 8 |
| | | HLN | 10 | 9 |

[Figure]

Relevé, Level I

**Figure ISS-6**—Nine relevé plots have been established in the most representative portion of the highest priority polygons in this example of the Level 1-Simple sampling scheme.

if problems arise. Examples could be high elevation snowfall or dangerous snags falling in a sample polygon. A table like ISS-3 should be included in the FIREMON notebook.

The low intensity of this sampling approach begs a cluster selection tactic to minimize transport times and maximize efficiency. In other words, the next prioritization criteria will often involve stand proximity. Those who know how to use GIS analysis can use GIS software routines to assign proximity measures to each stand based on transportation and prioritization attributes. For others, a color or grayscale map detailing prioritization and stratification attributes for each polygon overlaid on road, trail, and transportation route layers can be used to determine cluster centers and sample polygons.

The spatial prioritization criteria may be as simply stated as "sample all stands within 1 km of a road junction" or as complexly detailed as, "Sample all pole and mature stands within $\frac{1}{2}$ mile of a road junction that contain oak and hickory on slopes greater than 10 percent." Again, the selection of sample stands based on accessibility, adjacency, and attributes involves a great deal of subjectivity and sampling bias. We recommended that you have a list or map of polygons by stratum in order of priority for field crews so that they can alter sampling if obstacles are encountered in the field.

Using the prioritized polygons from table ISS-3 and then clustering the plots only marginally improves the sampling efficiency because many of the polygons that need to be sampled are not close to one another (fig. ISS-7). This is a good example of the realities of sampling.

[Figure]

Relevé, Cluster, Simple

**Figure ISS-7**—Attempting to cluster the prioritized polygons from table ISS-3 does not substantially improve the sampling efficiency because the polygons that need to be sampled are widely distributed on the landscape.

Remember to allow enough flexibility in the prioritization and selection process so that the field crews can modify sampling if problems arise. Examples could be high elevation snowfall or dangerous snags falling in a sample polygon.

**Using the Alternative sampling intensity (Level II)**

This sampling level assumes SP is close to NP. This level is often used with the relevé approach when monitoring is needed but there are not enough funds to implement a statistical approach to complete the project. The primary goal of this scheme is to describe all of the important conditions that need to be monitored on the sample landscape. This sampling intensity is recommended if a statistical approach is not warranted for the monitoring objective.

Sampling with the Level II approach assumes that you have enough resources to sample most or all of the polygons with one relevé. However, it is still important to prioritize the sampling so that the most critical polygons get sampled. The key to a successful monitoring effort for this scheme is to prioritize those stands on the landscape that need sampling and sample the most important ones first. This can be difficult to do because the FIREMON architect must balance distribution of the plots across the landscape and accessibility with importance to management. Detailed below is a method to select sample polygons using the polygons identified in table ISS-3 above. The architect can vary the theme or strategy to fit local circumstances. Again, this is not a rigid procedural step but rather a flexible framework for sampling design modification. It is important that a stratification system be explicitly stated and recorded in the FIREMON notebook.

Create a list of polygons to sample, ordering your sampling polygons based on your stratification and prioritization criteria. This list can be generated from a GIS or from a spreadsheet. Remember that this list is spatial in nature and does not take into account proximity and adjacency to other stands unless specifically designed and sorted.

If we use the ISS example landscape stratification described in the **Determining Polygons and Building a Summary Table** section then select two prioritization attributes—1) fuel load, especially in moderately loaded areas, and 2) exotic weed invasion, in that order—we can make up an ordered list of polygons for sampling (table ISS-4). In this example we will assume that SP is equal to 17. Figure ISS-8 shows how the plots might be located across the project area. For clarity, in this example tree density has also been used to order the polygons. However, tree density is not a prioritization attribute, so polygons that

**Table ISS-4**—Sites with moderate fuel load and weeds were used as prioritization attributes, then the strata in table ISS-2 were ordered to identify the polygons for sampling.

| Prioritization attribute 1 | Prioritization attribute 2 | Stratum code | Polygon number | Order of priority |
|---|---|---|---|---|
| Fuel load— moderate to low | Exotic weeds | HMY | 14 | 1 |
| | | MMY | 6 | 2 |
| | | MMY | 13 | 3 |
| | | LMY | N.A. | |
| | | HLY | N.A. | |
| | | MLY | 4 | 4 |
| | | LLY | 2 | 5 |
| | | HMN | 15 | 6 |
| | | HMN | 17 | 7 |
| | | MMN | 5 | 8 |
| | | MMN | 7 | 9 |
| | | MMN | 9 | 10 |
| | | MMN | 12 | 11 |
| | | LMN | N.A. | 12 |
| | | HLN | 8 | 13 |
| | | HLN | 10 | 14 |
| | | HLN | 16 | 15 |
| | | LLN | 1 | 16 |
| | | LLN | 3 | 17 |

[Figure]

Relevé, Level II

**Figure ISS-8**—All 17 polygons on the example landscape are sampled with one releve when the Level II-Alternative sampling scheme is applied.

have the last two letters of the stratum code in common ("MN" in "MMN") could be sampled in any order. Sampling the polygons that are closer together first will lower the sampling time.

We recommend that you have a list or map of polygons by stratification attribute in order of priority for field crews so that they can alter sampling if obstacles are encountered in the field. Remember to allow enough flexibility in the prioritization and selection process so that the field crews can modify sampling if problems arise. Examples could be high elevation snowfall or dangerous snags falling in a sample polygon. A table like ISS-4 should be included in the FIREMON notebook.

**Using the Detailed sampling intensity (Level III)**

There is rarely a situation that would match the relevé approach with a Detailed sampling intensity because the two are incompatible (see table ISS-1). Usually, if you have the resources to intensively sample a landscape, then a statistical approach is more appropriate to get more power from your monitoring results. However, if you want to use the relevé sampling approach with Level III sampling, then we suggest you follow the methods described in the previous section (Using the Alternative Sampling Intensity—Level II) with a small change. In the relevé Level II sampling scheme, plots are put in representative portion of most polygons on the sample landscape. With Level III sampling, we suggest that all special features be sampled within each polygon. The description or classification of special features must be explicitly stated in the sample design. For example, you might want to use a field on the Plot Description (PD) form, such a Landform, to detect special features. The goal of this sampling scheme is to sample all the polygons as well as the atypical conditions within the polygons, that represent conditions most important to fire effects monitoring. In this case, prioritization and stratification are not important; the only subjective element is the determination of special features. Cluster sampling is not needed because all polygons will be sampled.

**Implementing the Statistical Approach**

Use the statistical approach when it is important to compare the differences across polygons or sampling strata using statistically valid techniques. The statistical approach in FIREMON attempts to quantify the variance in a wide variety of sampled entities within a sampled polygon. Two or more sampled polygons can be compared to ascertain whether they are significantly different before and after a treatment. Moreover, before and after measurements of fire effects can be compared using standard statistical techniques to obtain a measure of change for the entire polygon rather than for one representative plot within the polygon such as used in relevé sampling. The statistical approach has strong interpretative power, but it comes at a cost. It is often resource-intensive to implement a statistical approach in fire monitoring because multiple plots per polygon are needed to quantify the variance of the myriad of ecosystem characteristics that are measured to evaluate fire effects.

The complex challenges posed by field sampling, such as steep slopes, dense stands and wildlife (snakes, bears, and so on), coupled with the extensive challenges of statistical sampling mean designing a statistically valid fire effects sampling scheme can be an extremely difficult and complex task that requires extensive expertise in statistical sampling techniques, field sampling, and operational management. As a result, this section in FIREMON is only a starting point for statistically based sampling and is not intended as a complete reference on the subject. We beleive that the material presented here provides an adequate start for a statistically valid sampling effort; however, if you are doing Level III sampling, then we strongly recommend that you have your sampling scheme designed or, at least, reviewed by an agency statistician or sampling expert. This is especially true if you are setting up a monitoring project on a large area, such as a watershed.

The FIREMON statistical approach assumes the fire manager wants a statistically relevant estimate of ecosystem characteristics for each polygon sampled. Since multiple plots are needed to quantify the variance in ecosystem characteristics within a polygon, the most difficult task for the manager is determining which polygons will be sampled. The only difference between sampling intensities Level I, II, and III is the number of polygons that will be sampled with the resources available.

The statistical approach may involve sampling at two spatial scales, which can make the sampling design difficult. The monitoring objectives might call for a statistical test of significance in fire effects at the landscape scale and the polygon scale. For example, the purpose of a FIREMON project may be to test if the entire landscape experienced a 50 percent reduction in fuels. Here, every polygon on the landscape, or groups of polygons on the landscape, must be sampled to test for statistical significance. However, only polygon-level changes may be important in another FIREMON project. For example, did the sampled polygons achieve 50 percent duff reduction? The FIREMON architect must decide whether landscape level, polygon level, or both levels of statistical testing are relevant to the monitoring objective. Record this information in the FIREMON Metadata table.

**Sample size determination**

The first important step in the statistical approach is to determine the number of plots needed to adequately sample each polygon. Most statistical sampling techniques recommend that sample size be determined by the amount of variability in the characteristic being sampled using standard formulae as determined from a pilot study—a small-scale sample collected simply to identify attribute variability. The two most often used statistical measures of variability are variance and standard deviation. Both are related to the difference between observed values in a group of numbers and the mean of those values. Standard deviation is simply the square root of the variance. Although either measure is appropriate, standard deviation is used more often because the units are the same as the units of the mean, whereas the units of variance are squared. For example, the standard deviation of 20 coarse woody debris (CWD) estimates might be 10 tons/acre or, equivalently, the variance would be 100 (tons/acre)$^2$. Variance and standard deviation estimates are easily made in spreadsheets, statistical software programs, and even on some handheld calculators. If resources allow, a pilot study of the characteristics that are important to the monitoring project should be done for each sampling stratum/sampling characteristic combination in the sampling project.

**Coefficient of variation for assessing variability**

Coefficient of variation (CV) is a third measure of variability but one that is not used often. However, it is a good measure to use when comparing variability estimates. Its benefit lies in the fact that it relates the attribute's standard deviation to its mean and, since most ecological attributes exhibit increasing variability with increasing mean, CV provides a variability measure that is somewhat standardized for comparison among attributes. For example, assume that you have a study and note that the standard deviation of fine woody debris (FWD) load is 1.0 tons/acre and the standard deviation of CWD is 3.0 tons/acre. In absolute terms the variability of the CWD is higher, but if the mean load of FWD is 0.5 tons/acre and the mean load of CWD is 3.0 tons/acre then, relatively, FWD is more variable because the

standard deviation is twice the mean, while the standard deviation of CWD equals the mean. The coefficient of variation is expressed as a percentage and is calculated using the formula:

Equation ISS-2 $$CV = \frac{s}{\bar{x}}(100)$$

where, s is the standard deviation estimate of an attribute $\bar{x}$ and is the estimated mean of the same attribute.

When trying to identify the attribute with the greatest variability we recommend using the coefficient of variation.

**Determining the attribute variability used to calculate sample size**

Sample size determination can be confounding in FIREMON because many fire effects monitoring projects sample more than one characteristic. For example, it is common for changes in fuels, tree mortality, and vegetation cover to be monitored before and after fire. This is quite different from conventional forest inventory techniques that use only timber volume to compute the required number of plots. The question is, which attribute should be used to represent the variability to compute the requisite number of plots? In FIREMON, we recommend that you use either the standard deviation of the most important characteristic or the standard deviation of the characteristic that has the greatest coefficient of variation.

The selection of the characteristic or characteristics to use to determine the number of plots ultimately depends on the importance of the sampled characteristic in successfully completing the monitoring objectives. If fuel reduction is the highest priority, then select fuel loadings as the variable to compute number of plots. If there is more than one characteristic important to the sampling objectives, then select the variable with the largest variance to ensure adequate plot representation for the other characteristics. For instance, if fuel reduction, tree mortality, and plant succession share equal weight in the monitoring effort, then select the variable, probably fuel loading, that has the highest within-sampling strata variance and calculate the sample size using that characteristic.

If, after calculating the number of required plots, you find that the sampling intensity is too high compared to the sampling resources, you will need to reduce the number of plots to a manageable level. One way to reduce the sample size is to sample the most important characteristic rather than the one with the highest variability. If that doesn't sufficiently reduce the number of required plots, estimate the number of plots needed after removing the least important polygons from the study. The worst-case scenario is that you start eliminating sampling methods—for instance, in the previous example, eliminating the vegetation sampling—or need to replace some intensive methods with less intensive methods, such as substituting photos for vegetation measurements. Remember, you want to be collecting the best quality data you can, given the objectives and resources. It is better to have a few plots with useful data than have lots of plots with data that doesn't let you assess how well you met the project objectives.

This is usually the point where most sampling designs come to a dead halt. The FIREMON architect can easily choose the characteristics to sample, but then must obtain some measure of variation for that selected characteristic. Typically, fire managers do not have the field data to quantify variation in sampled characteristics although, occasionally, data collected from past sampling efforts in similar terrain and ecosystems can be used. If field or pilot data are available for your FIREMON project, analyze them to determine the variation of a particular characteristic across a stand or mapped classification category, and then use the standard deviation to compute the number of required plots using the equation presented below.

If you don't have an idea of variability, the easiest and fastest way to find a variability estimate is to contact a local expert knowledgeable about the location and variables you are interested in sampling, who may be able to provide a good estimate of variability. The local expert will be able to state the variability in terms of the mean, and this will help put the variability in perspective. For instance, "On

that site, the standard deviation of CWD is about one and a half times the mean." If local experts cannot help, then examine research studies and reports to see if measures of variation for your characteristic of interest, on a similar landscape, are available. This can be time consuming and thus not possible for some projects. You may also be able to analyze previously collected FIREMON data by polygon or sampling stratum and identify an estimate of variability from the information. As a last resort, use your own experience or a best guess from what information you have been able to locate, and pick the standard deviation level that seems most appropriate—0.5, 1.0, or 1.5 times the mean. If you simply have no other information, calculate the required number of plots using a standard deviation that is equal to the mean of the attribute of interest.

There should be a measure of variability for the characteristic of interest for each sampling stratum in the monitoring project. For example, if the landscape is divided into polygons that are named according to cover type (ponderosa pine, Douglas-fir, lodgepole pine) and the selected variable is 1,000-hour fuel loading, then a measure of variability must be obtained for each cover type stratum on the landscape. This is often difficult because the data required to quantify variability are typically not available; therefore, it may be necessary to use the same standard deviation for a larger aggregation of sampling strata (all pine cover types).

**Calculating sample size**

The number of required plots (NRP) per polygon per sampling stratum can be computed a number of different ways to meet different statistical objectives. Monitoring projects designed using the FIREMON protocol can use the following equation with reasonable assurance the sample size will be appropriate:

Equation ISS-3
$$NRP = \frac{s^2 \left( Z_{/2} + Z \right)^2}{MDC^2}$$

where: $s$ is the standard deviation of the difference of the first and second sampling visit.

$Z_{\alpha}$ is the Z-coefficient for the type I error rate from table ISS-5.

$Z_{\beta}$ is the Z-coefficient for the type II error rate from table ISS-5.

MDC is the Minimum Detectable Change of the difference of sampled values, in absolute terms.

The confidence level for your monitoring project should be indicated in the project plan or the project objectives. If so, then the false-change error rate is calculated from the confidence level ($\alpha$ = 1− (confidence level/100)). If not, choose an $\alpha$ level that you feel is appropriate given the project objectives. In most research level studies it is 0.05 or lower. However, fire monitoring projects may not have to be as restrictive. Never set the confidence level below 80 percent (error rate = 0.20). The higher the confidence level the greater the number of samples needed. Most monitoring studies are less concerned with making a Type II error during the statistical analysis so use a missed-change error rate of 0.20 unless you have a reason to use a lower rate. Calculate NRP using the $Z_{\alpha}$ and $Z_{\beta}$ values that correspond to the a and b error rates. When using this equation be sure to divide $\alpha$ and $\beta$ by 2 before selecting the

Table ISS-5—Determination of sample size is dependent on the acceptable error rate. Use the appropriate z-values in this table for your sampling project.

| False-change (Type I) error rate ($\alpha$) | $Z_{\alpha/2}$ | Missed-change (Type II) error rate ($\beta$) | $Z_{\beta}$ |
|---|---|---|---|
| 0.40 | 0.84 | 0.40 | 0.25 |
| 0.20 | 1.28 | 0.20 | 0.84 |
| 0.10 | 1.64 | 0.10 | 1.28 |
| 0.05 | 1.96 | 0.05 | 1.64 |
| 0.01 | 2.58 | 0.01 | 2.33 |

z-value. For example, if you are using the error rates $\alpha = 0.10$ and $\beta = 0.20$, then $Z_\alpha$ and $Z_\beta$ would be 1.96 and 1.28, respectively.

Determining the MDC parameter can be confusing because it assumes the mean value of the attribute of interest is known, and generally it is not. Usually, you will be able to make an estimate that is sufficiently accurate for use in the NRP equation. If not, use either a pilot study or get information from an expert or the literature. Once the mean is known (or estimated) calculate MDC by multiplying the mean by the percent change you want to be able to detect. For instance, if you want to be able to detect a 20 percent change in a down woody debris load and you estimate the mean at 25 tons/acres, then MDC = 0.20(25) or 5 tons/acre. The lower the amount of change you want to detect the greater the number of plots you will need. Well-written objectives will give you some feeling for the detection level that is required in the study.

The NRP calculation must be completed for each polygon or sampling stratum in the project, then the NRP is summed across all polygons or sampling stratum on the landscape to compute the total NRP for the FIREMON sampling effort.

A good source for more information about determining NRP is: Measuring and Monitoring plant populations (Elzinga and others 1998 or Elzinga and others 2001). The entire 1998 publication is available as a PDF on the BLM library Web site: http://www.blm.gov/nstc/library/techref.htm. Select T.R. number 1730-1. Appendix Seven has a complete discussion about the determination of NRP.

**Using the Simple sampling intensity (Level I)**

The Simple sampling intensity level is used when the number of required plots (NRP) is much greater (more than two times) than the sampling potential (SP). It is inappropriate to match the statistical approach with the Level I-Simple sampling intensity level because they are in conflict. If the number of plots to establish is limited, then it will be difficult to achieve a statistically valid measure of variability at the polygon and landscape scale as required by the statistical approach. If the statistical approach is most important for the monitoring objectives, then the landscape level statistical validity must be given up to achieve statistical validity at the polygon level. If landscape statistical validity is important, then it will be difficult to sample the required number of polygons. If statistical validity is important reevaluate the project objectives to reduce the number of polygons that need to be sampled, so that NRP is roughly the same as SP, then use a polygon prioritization process.

There are several methods for reducing the number of sample polygons under the statistical approach. As stated in the previous **Considering Tradeoffs** section, one method is to revisit the project objectives to see if there are some polygons that can be sampled less intensively in terms of the number of plots or the methods applied. Another method is to prioritize the sampling polygons. Base the prioritization on the project objectives so that the most important attributes are sampled, then eliminate sampling in the least important polygons. Using a GIS can simplify the prioritization task. It is recommended that the priority list include some measure of accessibility to optimize the number of sampled polygons with the number of plots possible. For example, stand selection might include the following criteria: 1) within a mile of a road, 2) contain ponderosa pine, and 3) on slopes less than 50 percent.

If we use the ISS example landscape stratification described in the **Determining Polygons** and **Building a Summary Table** section (table ISS-2), then select two prioritization attributes—1) fuel load, especially in moderately loaded areas, and 2) exotic weed invasion—in that order, we can make up an ordered list of polygons for sampling (table ISS-6). For clarity, in this example tree density has also been used to order the polygons. However, tree density is not a prioritization attribute, so polygons that have the last two letters of the stratum code in common ("MN" in "MMN") could be sampled in any order. Sampling the polygons that are closer together first will lower the sampling time. We recommended that you have a list or map of polygons by stratification attribute in order of priority for field crews so that they can alter sampling if obstacles are encountered in the field. After selecting the polygons for sampling, reassess the your decision to use the Level I sampling scheme with the statistical approach. If the sampling intensity does not match the information required for the project objectives,

**Table ISS-6**—Sites with moderate fuel load and weeds were used as prioritization attributes, then the strata in table ISS-2 were ordered to identify the polygons for sampling.

| Prioritization attribute 1 | Prioritization attribute 2 | Stratum code | Polygon number | Order of priority |
|---|---|---|---|---|
| Fuel load— moderate to low | Exotic weeds | HMY | 14 | 1 |
| | | MMY | 6 | 2 |
| | | MMY | 13 | 3 |
| | | LMY | N.A. | |
| | | HLY | N.A. | |
| | | MLY | 4 | 4 |
| | | LLY | 2 | 5 |
| | | HMN | 15 | 6 |
| | | HMN | 17 | 7 |
| | | MMN | 5 | 8 |
| | | MMN | 7 | 9 |
| | | MMN | 9 | 10 |
| | | MMN | 12 | 11 |
| | | LMN | N.A. | 12 |
| | | HLN | 8 | 13 |
| | | HLN | 10 | 14 |
| | | HLN | 16 | 15 |
| | | LLN | 1 | 16 |
| | | LLN | 3 | 17 |

then adjust the methods and intensity you plan to use on each plot, move to the Level II sampling intensity, or use the relevé approach.

We recommended that you have a list or map of polygons by stratification attribute in order of priority for field crews so that they can alter sampling if obstacles are encountered in the field. Remember to allow enough flexibility in the prioritization and selection process so that the field crews can modify sampling if problems arise. Examples could be high elevation snowfall or dangerous snags falling in a sample polygon. A table like ISS-6 should be included in the FIREMON notebook.

**Using the Alternative sampling intensity (Level II)**

The Level II-Alternative sampling intensity level is used when the number of required plots (NRP) is between one and two times greater than the sampling potential (SP). If NRP is greater than the number of polygons (NP) to be measured, then NP must be reduced to make NRP roughly equal to SP. The reduction will probably not compromise the statistical validity of the sample design if a landscape level measure of variability is not as important to the monitoring objective as a polygon level measure of variability.

There are several methods for reducing the number of sample polygons under the statistical approach. As stated in the previous **Considering Tradeoffs** section, one method is to revisit the project objectives to see if there are some polygons that can be sampled less intensively in terms of the number of plots or the methods applied. Another method is to prioritize the sampling polygons. Base the prioritization on the project objectives so that the most important attributes are sampled, then eliminate sampling in the least important polygons. Using a GIS can simplify the prioritization task. It is recommended that the priority list include some measure of accessibility to optimize the number of sampled polygons with the number of plots possible. For example, stand selection might include the following criteria: 1) within a mile of a road, 2) contain ponderosa pine, and 3) on slopes less than 50 percent.

If we use the ISS example landscape stratification described in the **Determining Polygons** and **Building a Summary Table** sections (table ISS-2), then select two prioritization attributes—1) fuel load, especially in moderately loaded areas, and 2) exotic weed invasion—in that order, we can make up an ordered list of polygons for sampling (table ISS-7). For clarity, in this example tree density has also been used to order the polygons. However, tree density is not a prioritization attribute so polygons

**Table ISS-7**—Sites with moderate fuel load and weeds were used as prioritization attributes, then the strata in table ISS-2 were ordered to identify the polygons for sampling.

| Prioritization attribute 1 | Prioritization attribute 2 | Stratum code | Polygon number | Order of priority |
|---|---|---|---|---|
| Fuel load— moderate to low | Exotic weeds | HMY | 14 | 1 |
| | | HLY | N.A. | |
| | | MMY | 6 | 2 |
| | | MMY | 13 | 3 |
| | | MLY | 4 | 4 |
| | | LMY | N.A. | |
| | | LLY | 2 | 5 |
| | | HMN | 15 | 6 |
| | | HMN | 17 | 7 |
| | | HLN | 8 | 8 |
| | | HLN | 10 | 9 |
| | | HLN | 16 | 10 |
| | | MMN | 5 | 11 |
| | | MMN | 7 | 12 |
| | | MMN | 9 | 13 |
| | | MMN | 12 | 14 |
| | | MLN | 11 | 15 |
| | | LMN | N.A. | |
| | | LLN | 1 | 16 |
| | | LLN | 3 | 17 |

that have the last two letters of the stratum code in common ( "MN" in "MMN") could be sampled in any order. Sampling the polygons that are closer together first will lower the sampling time.

We recommended that you have a list or map of polygons by stratification attribute in order of priority for field crews so that they can alter sampling if obstacles are encountered in the field. Remember to allow enough flexibility in the prioritization and selection process so that the field crews can modify sampling if problems arise. Examples could be high elevation snowfall or dangerous snags falling in a sample polygon. A table like ISS-7 should be included in the FIREMON notebook.

If it is important to test for statistical significance at the polygon and landscape scale, then a random selection of polygons is warranted. This random selection should be weighted by some factor important to the monitoring objective. For example, it may be important to sample the greatest area, so selection should be weighted by area. This can be accomplished in a GIS by using a random selection algorithm linked to stand weights, or it can be done outside a GIS using a random number generator or list. Be sure enough stands will be sampled across the landscape to obtain a statistically valid estimate of variability.

**Using the Detailed sampling intensity (Level III)**

The Detailed sampling intensity level is used when the number of required plots (NRP) is much less than the number of plots that are possible with the available resources (SP).

Developing a sampling design at this level can be quite complex and beyond the scope of the FIREMON ISS. We recommend that you employ an expert in statistical sampling to design your sampling project. It will cost a small fraction of what will be spent sampling and will reap great rewards. Contact a statistical expert at your local research institution or university for assistance.

**Plot distribution when using the statistical approach**

In FIREMON we suggest plots be distributed using either systematic or random placement. When properly applied each will give you a statistically valid sample. The number of plots in each polygon can be weighted by area of each polygon in a sampling stratum or divided evenly among all of the polygons in the sampling stratum. For example, say NRP = 100 plots and there are two polygons in one sampling stratum with sizes of 20 and 80 acres. You could either weight the number of plots per polygon by area,

resulting in 20 plots in the first and 80 plots in the second polygon. Or, you could divide the plots evenly and have 50 plots in each polygon. Unless all the polygons are close to the same size the first option is probably the most appropriate.

When using the systematic approach, plots are distributed by developing a grid spacing that will give you the appropriate number of plots within each of your polygons in each sampling stratum. The base corner of the grid should be located at random. Choose randomly distributed plot locations by developing a system to locate plots within a polygon with a list of random numbers. Depending on the variability of the attributes you are monitoring, one sampling stratum may need more or fewer plots per polygon than another stratum.

A third plot distribution technique is to locate the sampling plots in clusters. If you are establishing cluster plots, the location of plot clusters is dependent on landscape features such as roads and topography in order to reduce the sampling effort. The best cluster design allows the maximum number of plots to be established and sampled with the least bias in plot location. Cluster sampling may not result in a statistically valid sample if plots are placed with bias, are not independent samples, or do not allow sampling across the entire range of conditions in each stratum.

Figure ISS-9 shows how plots could be distributed on the ISS example landscape using each of the plot location methods at the Detailed level of sampling. Illustrations E, F, and G each show 66 sampling locations. You could easily make this the Alternative intensity (Level II) by limiting the methods on each plot to only those critical to the objectives. For example, at the Detailed level, you may plan to use cover/

[Figure]

[Figure]

[Figure]

[Figure]

[Figure]

Statistical, Systematic, Detailed

[Figure]

Statistical, Random, Detailed

[Figure]

Statistical, Cluster, Detailed

**Figure ISS-9**—The three stratificaitons of the example ISS landscape (A, B, and C) are combined to identify the sampling polygons (D). Then, using the statistical, detailed approach, 66 sampling plots are distributed on the landscape map. Illustration (E) shows the systematic approach, (F) the random approach, and (G) the cluster approach.

frequency (for weed sampling), fuel load, and tree data methods on each plot in your study. To apply the Alternative level you could eliminate the cover/frequency methods when a plot falls out of the weed corridor. You would still be establishing 66 plot locations, but the sampling time would be reduced because you would not be using all three sampling methods on all of the plots.

**Establishing Control Plots**

Control plots are established in areas outside the perimeter of your treatment unit in order to collect reference data that will be used to compare against your posttreatment data. Control plots are especially important in rangeland ecosystems where year-to-year variation in weather can mask changes caused by fires. For example, say a prescribed burn unit in a sagebrush-steppe ecosystem has 10 plots established within burn boundaries and five plots established outside burn boundaries. When the unit is burned, the postburn measurement of grasses shows a doubling of biomass. The inference is that the burn has increased grass production. However, remeasurement of the control plots (plots that were not burned) shows that the grass biomass on these plots has doubled as well. So the increase in grass biomass was actually a result of the some other factor, a wet spring, for instance, that occurred that year, not from the fire. Control plots allow you to assess the effects of factors other than those you applied with your treatments (such as weather or wildlife) on the characteristics you are monitoring.

While control plots are valuable they are not required in many monitoring efforts. The decision whether or not to establish controls depends on the ecosystem, monitoring objectives, and available resources. Control plots may not be necessary in ecosystems that are relatively unaffected by annual or decadal weather variations, such as forest and alpine environments. The establishment of controls is usually warranted if the monitoring objective is to statistically determine significant changes in ecosystem characteristics in environments where vegetation is substantially affected by annual fluctuations in weather. Examples are grasslands, shrublands, and some forest understories. If project resources are limiting, then, depending on objectives, control plot establishment is probably the first task to be trimmed.

As when determining the sampling intensity for monitoring plots, the number of control plots to establish outside the treatment boundary depends on the environment, objectives, available resources, and the availability of appropriate sites. At a minimum, at least one control plot should be established if you are using the relevé approach. At least three plots should be established when using the statistical approach. A sampling objective that specifies statistical significance will probably require more sampling units to adequately capture the variance of sampled entities and, thus, a greater number of control plots. The most complete sampling design will have at least one control plot for every sampling stratum in the study. Some areas have diverse mosaics of stand conditions within burn boundaries so more than one control area may be warranted to adequately capture the within-burn prefire heterogeneity. This is especially true of landscapes with highly diverse patch characteristics.

Controls should be established adjacent to or near the treatment area. They should be located in an area that represents the characteristics important to the monitoring objective, found inside the treatment area. Controls should be established at the same time as the pretreatment measurements, and they should be installed using the same methods as you used on the monitoring plots. The only difference between control plots and monitoring plots is their location. When using the relevé approach, establish control plots in an area outside the treatment area that best represents the area within the treatment area. Use random or systematic control plot location, when using the statistical approach. If there are no areas outside treatment boundaries that are suitable for control establishment, establish control plots in the next best area with similar aspect, slope, and elevation. Sometimes more than one control area will be needed because of diverse characteristics (slope, aspect, vegetation, and so on) within the burn. Remember, control plots are used to determine temporal variation in ecosystem characteristics caused by factors other than the treatments. They do not need to be placed on locations that have identical characteristics to the treatment area, but they should be similar. The use of potential vegetation types or site types can be helpful to stratify control sample areas.

Monitoring plots established inside treatment boundaries that do not burn or are not impacted by other treatment activities have intrinsic value as fire effects control plots. For instance, a prescribed fire, for some reason such as high fuel moisture, may not actually visit all the monitoring plots laid out inside a burn perimeter, and these unburned plots could supplement your other control plots. Care must be taken, however, because the reason that the plot didn't burn may be related to the attribute you are trying to control. For instance, if a plot didn't burn because the fuel moistures were high, it might mean that a seep is subirrigating the area around the unburned plot, which in turn would increase vegetation cover and height. Using this unburned plot as a control plot to compare vegetation would be inappropriate. If there are monitoring plots that will be used as control plots, sample them at the same time and with the same methods as your monitoring plots.

Controls are sometimes useful for monitoring fire effects on plots when you were not able to install monitoring plots before an event, such as a wildfire. Control plots can approximate preburn conditions and these pseudo-preburn conditions can be used as reference. Simply establish the controls outside of the burn boundary in areas that best approximate the characteristics found prior to the burn as observed inside the fire perimeter. Use snags and downed logs to help visualize what the stand looked like before the burn. Many people do not establish controls after a fire because the subsequent information is somewhat suspect, as the preburn stand conditions within the fire can never be truly determined. However, when they are appropriately used, postfire control plots can provide useful information. You may need to establish controls on many site types to cover the wide range of environmental conditions within the burned area.

Ideally, each control plot should be remeasured at the same time as the burned monitoring plots. It is important that the controls be measured at least twice—once to establish the controls and once at the end of the monitoring effort.

**FIREMON GUIDES AND KEYS**

**Sampling Strategy Checklist**

*Suggested step-by-sep procedure for designing a firemon monitoring sampling effort*

*See figure ISS-1*

1. **State the monitoring objective(s)**. Describe, in detail, the reasons why this monitoring effort is being implemented.
2. **Determine the sample area**. Create a map that explicitly identifies the boundaries of the landscape to be sampled.
3. **Determine the sample stratification**. Create a map that delineates each polygon on the sample landscape, and then describe each polygon by one or more attributes that will be used to stratify monitoring results.
4. **Determine the sampling resources**. Record all resources that can be used in this sampling effort including personnel, monies, time, and vehicles.
5. **Calculate sampling resource statistics**. Compute the sampling statistics that are used throughout the FIREMON sampling design process (SP, NRP, and so on).
6. **Determine the sampling approach**. Select the most appropriate sampling approach, statistical or relevé, using the keys and text presented in FIREMON.
7. **Determine the sampling intensity level**. Select the most appropriate level of sampling intensity from the keys and text presented in FIREMON.
8. **Choose the most appropriate sampling methods**. Select the measurement methods that will be used to sample ecological characteristics using the keys and the text presented in FIREMON.

9. **Compare monitoring design and project objectives**. If the design is compatible with the objectives, distribute plots and begin sampling. Otherwise, review the project and go through the checklist again.

10. **Determine plot locations**. Use stratification factors to determine polygons, then distribute plots over the sampling area.

**Suggested step-by-step procedure for implementing a FIREMON monitoring sampling effort**

1. **Locate a FIREMON sampling polygon**. Use the sample design to select, then navigate to a stand on the sample landscape.

2. **Locate a FIREMON sampling plot**. Use the directions in How to Locate a Plot to go to an area within a stand that will be sampled.

3. **Establish a FIREMON sampling plot**. Use the directions in How to Establish a Plot to permanently or semipermanently mark the area to sample.

4. **Follow the procedures for each selected sampling method**. Refer to the Sample Methods discussions for the sampling methods and protocols selected for the project and follow those instructions for the appropriate sampling strategy and intensity level.

5. **Record measured field information on FIREMON plot forms**. Write down all information and measured entities on the plot forms provided in FIREMON. You may also record the measurements directly onto a field laptop computer.

6. **Check recorded information**. Double check your entries on the plot forms and make sure all fields are completed and appear correct.

7. **Enter data on plot form into FIREMON databases**. Enter the recorded field data into the Microsoft Access Databases provided by FIREMON.

**Field Assessment**

The field assessment portion of FIREMON contains an extensive set of procedures for sampling important ecosystem characteristics before and after a prescribed or natural fire for terrestrial ecosystems. The field assessment is composed of 1) field methods, 2) plot forms and cheat sheets, and 3) equipment lists. FIREMON has been designed so that the fire manager can tailor the field measurement procedures to match burn objectives or wildland fire use concerns. Additionally, the fire manager can scale the intensity of measurement to match resource and funding constraints. For example, to document tree mortality, the fire manager would choose one of three hierarchically nested sampling procedures, where the first procedure would provide general descriptions of tree mortality quickly at low cost (photopoints, walk-through), while the third procedure would document, in detail, individual tree health and vigor, to generate comprehensive data applicable to many analyses but costly to collect. A key has been provided help the fire manager decide the appropriate methods and sampling intensity.

**Sampling protocols**

FIREMON contains the following sampling procedures for monitoring many ecosystem characteristics:

**Plot Description (PD)**—A generalized sampling scheme used to describe site characteristics on the FIREMON macroplot with biophysically based measurements.

**Species Composition (SC)**—Used for making ocular estimates of vertically projected canopy cover for all or a subset of vascular and nonvascular species by DBH and height classes using a wide variety of sampling frames and intensities. This procedure is more appropriate for inventory than monitoring.

**Cover/Frequency (CF)**—A microplot sampling scheme to estimate vertically projected canopy cover and nested rooted frequency for all or a subset of vascular and nonvascular species.

**Point Intercept (PO)**—A microplot sampling scheme to estimate vertically projected canopy cover for all or a subset of vascular and nonvascular species. Allows more precise estimation of cover than the CF methods because it removes sampler error.

**Line Intercept (LI)**—Primarily used when the fire manager wants to monitor changes in plant species cover and height of plant species with solid crowns or large basal areas where the plants are about 3 feet tall or taller.

**Density (DE)**—Primarily used when the fire manager wants to monitor changes in plant species numbers. This method is best suited for grasses, forbs, shrubs, and small trees, which are easily separated into individual plants or counting units, such as stems. For trees and shrubs over 6 feet tall the TD method may be more appropriate.

**Rare Species (RS)**—Used specifically for monitoring rare plants such as threatened and endangered species.

**Tree Data (TD)**—Trees and large shrubs are sampled on a fixed-area plot. Trees and shrubs less than 4.5 feet tall are counted on a subplot. Live and dead trees greater than 4.5 feet tall are measured on a larger plot.

**Fuel Load (FL)**—The planar intercept (or line transect) technique is used to sample dead and down woody debris in the 1-hour, 10-hour, 100-hour, and 1,000-hour and greater size classes. Litter and duff depths are measured at two points along the base of each sampling plane. Cover and height of live and dead, woody, and nonwoody vegetation is estimated at two points along each sampling plane.

**Landscape Assessment (LA)**—Useful for mapping fire severity over large areas by combining satellite-derived Normalized Burn Ratio (BR) with a ground-based indicator of fire severity, Composite Burn Index (BI). The LA methodology will assist in determining landscape-level management actions where fire severity is a determining factor.

**Composite Burn Index (BI)**—The BI methodology is a subset of the LA methods. It provides users with a ground-based fire severity index derived from a number of plot measurements.

**Normalized Burn Ratio (BR)**—The BR method is the subset of the LA methods. It describes how to derive remotely sensed spatial information on burn severity, using Landsat satellite data.

We used the Western Region Fire Monitoring Handbook (FMH) (National Park Service 2001, see http://www.nps.gov/fire/fire/fir_eco_firemonitoring.html), ECODATA handbook (Hann and others 1988; Jensen and others 1993; Keane and others 1990) and the USDA Forest Service Natural Resources Information System (NRIS) protocols as the framework for selecting and designing FIREMON sampling methods. We modified these protocols so that there are now nested levels of sampling intensity coupled with nested levels of sampling flexibility.

**Sample Approach Classification Key**

Answer each bullet with a "yes" or "no" and add up the answers. More "yes" answers suggest using the statistical approach, and more "no" answers suggest using the relevé approach.

1. *Sufficient sampling resources are available*. There is plenty of funding, ample time, and sufficient personal (with necessary skills) to complete a detailed monitoring effort.
2. *NP < SP*. There are sufficient resources to sample the entire landscape. Sampling resources allow more than one plot in each stand.
3. *An estimate of across and within stand variation is important*. The project objectives require an estimate of variability in ecosystem characteristics or the statistical comparisons of sampled attributes.
4. *A statistician or statistics expert is available for consultation*. Someone can easily be contacted to answer questions about your sampling design. There is sufficient expertise for designing a valid statistical sampling scheme.

5. ***Navigation across the sample landscape is relatively easy***. Steep, dangerous terrain, long travel distances, or other features that limit plot access are not present on major portions of the landscape.

6. ***Few ecosystem components are being measured for assessing fire effects***. The monitoring objectives are concerned with just one or two ecosystem attributes whose variation must be quantified.

**Sampling Intensity Key**

Answer each bullet under each intensity level with a "yes" or "no." Count up the number of "yes" votes for each intensity level, and the level with the most "yes" votes would suggest that this may be the most likely intensity level for your monitoring study. The bullets are listed in order of importance with the first few bullets most important in sample design.

**Simple sampling intensity (Level I)**

1. ***Little funding is available.*** This project has little financial support and must be done with existing personnel and equipment.

2. ***SP << NP***. There are many more polygons to be sampled than there are potential plots to sample them. A good rule-of-thumb is that NP is more than twice the number of potential plots, then a simple sampling approach is appropriate.

3. ***There is little time to conduct the project***. There are only a few weeks or months to sample, and the project must be completed as quickly as possible.

4. ***Description is more important than comparison or evaluation***. The monitoring objectives can be accomplished by establishing enough plots to generally describe fire effects without quantifying the variability of the sampled attributes.

5. ***There are few people available***. It will be difficult to hire or obtain a crew of experienced field technicians, or it will be difficult to train inexperienced crews to collect the fire effects data

6. ***Travel across the landscape is difficult and restrictive***. The polygons being sampled are so difficult to traverse that establishing multiple plots in each stand would be laborious and time consuming. The landscape may be steep and dangerous, composed of thick vegetation and deep fuels, or contain many dangerous obstacles such as deep rivers, cliffs, ice, and so forth.

**Alternative sampling intensity (Level II)**

1. ***Sampling resources are available but limited***. Funding and other resources (people, vehicles) are available but not abundant. One or more categories of sampling resources is limiting such as few people, money, or time, but overall, there appear to be resources available.

2. ***SP = > NP***. There are about the same number of sample polygons as there are potential plots to sample them. There is a possibility that some polygons will be sampled with more than one plot. Or it is possible that some polygons may not be important to the objectives and could be removed from the list of polygons to be sampled.

3. ***An estimate of across and within stand variation is important but not essential***. An estimate of error in comparing polygon conditions is desired but not essential. It is more important that conditions within each polygon on the landscape be described so that management can proceed.

4. ***Many ecosystem components are being measured for assessing fire effects***. The monitoring objective is concerned with describing fire effects for many ecosystem elements (fuels, trees, plant species) so that an integrated stand-level evaluation of fire effects is possible.

5. ***A compromise is desired between the simple and detail methods***. Level I intensity is not enough to accomplish monitoring objectives and there are not enough sampling resources to implement a project at Level III intensity.

6. ***The sample area is complex, rugged, or remote***. Having to sample at a high intensity is not desirable. The size, topography, and limited roads/trails within the sampling area may limit the possibility of establishing enough plots to quantify polygon conditions across the entire area.

**Detailed sampling intensity (Level III)**

1. ***Sampling resources are abundant***. There are sufficient resources to conduct the monitoring project. None of the sampling resource categories is limiting.

2. ***SP >> NP***. The sampling potential is high enough that multiple plots can be established in each polygon and should allow *at least* two plots per polygon.

3. ***An estimate of stand variation is important***. An estimate of error across the sampling polygons is essential for describing fire effects. It is important that the error in fire effects be quantified to make management decisions.

4. ***A statistically defensible comparison is important***. It is important that results from this study capture sampled variations so statistical comparisons can be made with error estimates. This implies that the sampling approach is required to get the most statistically defensible results to back up any management action.

5. ***Only a few ecosystem components are being measured for assessing fire effects***. The monitoring objective is only concerned with one or two ecosystem attributes whose variability can be easily quantified for computing the number of plots to establish.

6. ***The sampling environment allows an intensive sampling effort***. There are no foreseeable dangers or restrictions in the area to be sampled; the area is safely and easily accessible. Crews are sufficiently trained and available to conduct this extensive sampling.

**Methods Classification Key**

Use this key to help you decide what methods should be used in your monitoring project. Start at the top of the key and, for every statement that is true for your situation, record the suggested methods or fields in your FIREMON notebook. Once finished, review the methods you have identified. Compare them with the sampling resources available and project objectives to determine if they are right for your project.

1. A description of the ***physical environment*** is important for providing context to monitoring results or for providing stratification for the data analyses. Physical descriptions include elevation, aspect, slope, soils, landform, and slope position. These measurements provide a general description of biophysical processes that might influence fuel, fire, and vegetation dynamics. It is strongly recommended that these variables be recorded at each plot.

   1.1. **Complete Biophysical Settings Fields in PD method**

2. A general description of ***vegetation characteristics*** is important for understanding monitoring results, stratifying analyses, or validating satellite imagery. This description includes lifeform (trees, shrubs, and grasses) cover by size class and an estimation of cover type and potential vegetation type. These measurements are especially helpful in describing general vegetation conditions and for relating plot-level stand-related vegetation characteristics to satellite imagery analysis and mapping.

   2.1. **Complete Vegetation Fields in PD method**

3. A general description of ***stand fuel characteristics*** is important for summarizing monitoring and interpreting monitoring results, stratifying analyses, or correlating with satellite imagery. These descriptions include ground cover (ash, bare soil, rock), fuel model, and crown fuel characteristics. These measurements are especially helpful in describing general fire-related fuel conditions and for relating plot-level fuels characteristics to satellite imagery analysis and mapping.

   3.1. **Complete the Fuels Fields in PD method**

4. ***Documentation of plot conditions and location*** is important to the monitoring objective. This includes photo-documentation, written notes, maps, and so on. This method is used to strengthen the

documentation of the location of the plot and to document spatial characteristics of the plot using visual tools.

4.1. **Complete the Common and Comments Fields in the PD method**

5. A general description of *fire behavior and effects* at the plot level is important for describing fire conditions in the interpretation of monitoring results, stratification of analyses, or presentations with the public. Descriptions include photo-documentation of fire behavior and effects, and quantification of overall fire effects. These methods are used to generate a pictorial assessment of fire behavior and effects, and are especially effective for relating fire effects to those unfamiliar with fire ecology.

5.1. **Complete the Fire Behavior and Effects Fields in the PD method**

6. A general description of *ambient weather conditions, fuel moistures, and fire behavior* for a prescribed fire is important for assessing fire effects. This information is collected for the entire fire event and is more complete than the plot level descriptive information collected using the PD methods.

6.1. **Complete the Fire Behavior (FB) methods**

7. *Changes in plant species cover and/or height* is important in assessing fire effects. Objectives state monitoring of individual species presence, cover, and/or height is important to the project. Possible applications include changes in species cover and height of threatened and endangered species, important forage species, and reductions in tree understory. This method is used mostly to track succession development in vegetation over time and to quantify changes in species cover due to disturbance.

7.1. A statistically valid comparison of changes in species cover over time is not important. Simple Sampling Intensity Level I. *Descriptive changes in species cover will fulfill monitoring objectives*. Species measurements include cover and height estimates without error estimates.

7.1.1. **Complete the Species Composition (SC) method**s. (See the Vegetation Sampling Overview for more information.)

7.2. *A statistical comparison of changes in species occurrence is important* for the successful completion of objective. Species measurements include plant frequency, cover, and height with error estimates.

7.2.1. *Height of >50 percent vegetation cover less than 3 feet*. Majority of plot is composed of plants that are less than 6 feet tall.

7.2.1.1. *A quadrat based examination of plant frequency or subjectively determined plant cover* is important to the project.

7.2.1.1.1. **Complete Cover/Frequency (CF) methods**. (See the Vegetation Sampling Overview for more information.)

7.2.1.2. *Objective determination of plant cover* is important for the project. Mostly fine-textured vegetation.

7.2.1.2.1. **Complete the Point Cover (PO) methods.** (See the Vegetation Sampling Overview for more information.)

7.2.1.3. *It is important to monitor changes in the number of plant species and/or the number of individual plants*.

7.2.1.3.1. **Complete the Density (DE) methods**. (See the Vegetation Sampling Overview for more information.)

7.2.2. *Height of >50 percent vegetation greater than 3 feet*.

7.2.2.1. **Complete Line Intercept (LI) method**. (See the **Vegetation Sampling Overview** for more information.)

8. Changes in *fuel loadings* are important to successfully completing monitoring objectives. Fuel measurements include loadings of all woody fuel size classes, duff, litter, live and dead shrub and herbaceous; duff and litter depths; and coarse woody debris description with rot classes. This method

is used mostly for estimating fuel consumption and smoke generation, and to describe stand-level fuel characteristics.

8.1. **Complete the Fuel Load (FL) method**

9. Changes in ***tree or stand characteristics*** (mortality, survival, damage) are important for describing fire effects for monitoring objective. Tree measurements include health, insect and disease evidence and damage, crown characteristics, diameter, height, and fire damage. This method is used mostly to quantify tree and stand mortality and to describe stand-level tree characteristics.

9.1. **Complete the Tree Data (TD) method**

10. Documentation of ***aggregate fire effects within strata*** is important to determine cumulative burn severity on the community. Coverage of large areas and diverse conditions is important, with minimal time spent per plot. Also, a means to calibrate or validate remote sensing data is sought for moderate resolution applications. This method integrates independent ratings of severity by strata to determine understory, overstory, and overall severity.

10.1. **Read the Landscape Assessment (LA) section, and complete the Composite Burn Index (BI) method**.

11. Spatial representation of ***fire effects is desired over large areas using GIS***, targeting burns exceeding about 100 ha (250 ac). Analysis of historic burns (back to about 1983) is desired. A need exists to monitor burns over long periods, as in a regional fire atlas, or to relate fire effects to environmental variables continuously across a landscape. Also, mapping is important, to display or analyze burn results at a landscape resolution of 30 meters.

11.1. **Read the Landscape Assessment (LA) section, and complete the Normalized Burn Ratio (BR) method**.

**Vegetation Sampling Overview**

The FIREMON system uses five vegetation sampling procedures that are useful for sampling vegetation for most monitoring situations. Each method has its strengths and weaknesses, and the descriptions below are provided to help you determine which sampling method is best for your project.

**Species Composition (SC) Method**: This method is primarily used to acquire inventory data over large areas using few examiners. The SC method is useful for documenting important changes in plant species cover and composition over time. However, this method is not designed to monitor statistically significant changes in vegetation over time. The SC sampling method primarily addresses individual plant species canopy cover and height for vascular and nonvascular plants. Canopy cover and average height may be recorded by size classes for plant species. Size class data can provide important structural information about the stand such as the vertical distribution of plant species cover.

[remaining 976,629 characters of this post omitted]

---

## Author Response (AR3)

RESPONSE TO EDITOR'S COMMENTS

Comments to the Author:
- I agree with the reviewer, and think that the last three sentences of the abstract are too speculative and misleading, please remove them: "However, the very large fraction of the emissions deposited on the Greenland Ice Sheet makes these fires very efficient climate forcers on a per unit emission basis. If the expected future warming of the Arctic produces more severe fires in Greenland, this could indeed cause albedo changes and thus contribute to accelerated melting of the Greenland Ice Sheet. The fires burning in 2017 may be a harbinger of such future events."
This is a local effect, and saying that they are very efficient climate forcers is misleading even if these fires become two orders of magnitude larger.
Maybe something along these lines is more appropriate:
However, the very large fraction of the emissions deposited on the Greenland Ice Sheet from these fires could contribute to accelerated melting of the Greenland Ice Sheet if these fires become several (1-2??) orders of magnitude larger under future climate.

RESPONSE: We agree and have corrected it as the Editor suggested.

Please address this issue in the conclusion too.

RESPONSE: Conclusions were corrected it as the Editor suggested.

- In section 4.1 please use "to evaluate" rather than "to validate".

RESPONSE: Corrected in the text and title of section 4.1 (see track changes).

- L938. There are two "on" please correct that.

RESPONSE: Corrected (see track changes on Conclusions section).

[revised manuscript text omitted]